# PD-1/CD80$^+$ small extracellular vesicles from immunocytes induce cold tumours featured with enhanced adaptive immunosuppression

Lin-Zhou Zhang[1,10], Jie-Gang Yang[1,2,10], Gai-Li Chen[3], Qi-Hui Xie[1,2], Qiu-Yun Fu[1], Hou-Fu Xia[1,2], Yi-Cun Li[4], Jue Huang[1], Ye Li[1], Min Wu[1], Hai-Ming Liu[1], Fu-Bing Wang[5], Ke-Zhen Yi[6], Huan-Gang Jiang[3], Fu-Xiang Zhou[3], Wei Wang[7], Zi-Li Yu[1,2], Wei Zhang[1,2], Ya-Hua Zhong[3], Zhuan Bian[1], Hong-Yu Yang[4], Bing Liu[1,2] & Gang Chen [1,2,8,9] ✉

Only a minority of cancer patients benefit from immune checkpoint blockade therapy. Sophisticated cross-talk among different immune checkpoint pathways as well as interaction pattern of immune checkpoint molecules carried on circulating small extracellular vesicles (sEV) might contribute to the low response rate. Here we demonstrate that PD-1 and CD80 carried on immunocyte-derived sEVs (I-sEV) induce an adaptive redistribution of PD-L1 in tumour cells. The resulting decreased cell membrane PD-L1 expression and increased sEV PD-L1 secretion into the circulation contribute to systemic immunosuppression. PD-1/CD80$^+$ I-sEVs also induce downregulation of adhesion- and antigen presentation-related molecules on tumour cells and impaired immune cell infiltration, thereby converting tumours to an immunologically cold phenotype. Moreover, synchronous analysis of multiple checkpoint molecules, including PD-1, CD80 and PD-L1, on circulating sEVs distinguishes clinical responders from those patients who poorly respond to anti-PD-1 treatment. Altogether, our study shows that sEVs carry multiple inhibitory immune checkpoints proteins, which form a potentially targetable adaptive loop to suppress antitumour immunity.

Immune checkpoints are hard-wired pathways in the immune system that are responsible for maintaining immune homoeostasis and preventing autoimmunity[1]. Cancer cells exploit inhibitory immune checkpoints to evade the immune system and promote malignancy.

Thus, immune checkpoint blockade therapy using monoclonal antibodies is a rational therapeutic approach[2–4]. Immunotherapy has demonstrated remarkable promise in treating various malignancies[5–7]. However, only a minority of patients respond to this treatment and the

[1]State Key Laboratory of Oral & Maxillofacial Reconstruction and Regeneration, Key Laboratory of Oral Biomedicine Ministry of Education, Hubei Key Laboratory of Stomatology, School & Hospital of Stomatology, Wuhan University, Wuhan 430079, China. [2]Department of Oral and Maxillofacial Surgery, School and Hospital of Stomatology, Wuhan University, Wuhan 430079, China. [3]Department of Radiation and Medical Oncology, Hubei Key Laboratory of Tumour Biological Behaviors, Hubei Cancer Clinical Study Center, Zhongnan Hospital of Wuhan University, Wuhan 430071, China. [4]Department of Oral and Maxillofacial Surgery, Peking University Shenzhen Hospital, Shenzhen 518036, China. [5]Department of Laboratory Medicine and Center for Single-Cell Omics and Tumour Liquid Biopsy, Zhongnan Hospital of Wuhan University, Wuhan 430071, China. [6]Department of Laboratory Medicine, Zhongnan Hospital of Wuhan University, Wuhan 430071, China. [7]Department of thoracic surgery, Renmin Hospital of Wuhan University, Wuhan 430060, China. [8]TaiKang Center for Life and Medical Sciences, Wuhan University, Wuhan 430071, China. [9]Frontier Science Center for Immunology and Metabolism, Wuhan University, Wuhan 430071, China. [10]These authors contributed equally: Lin-Zhou Zhang, Jie-Gang Yang. ✉e-mail: geraldchan@whu.edu.cn

responses are often partial or short-lived, limiting its widespread clinical implementation[8]. Thus, a comprehensive understanding of tumour resistance mechanisms is required to improve the patient response rate.

Among the currently identified inhibitory immune checkpoints, cytotoxic T-lymphocyte-associated antigen 4 (CTLA-4) and programmed cell death protein 1 (PD-1) are the most well-known checkpoints that negatively regulate T-cell function at different phases and by different mechanisms[9]. CTLA-4 is exclusively expressed on T cells, where it primarily regulates the amplitude of T-cell activation at the early phase[10]. It can suppress the activation of T cells by competing with CD28 for ligand binding of CD80 and CD86[11]. In contrast, PD-1 limits activated T-cell function at a later phase in peripheral tissues or tumour sites[12–14]. PD-1 ligands, particularly PD-L1, are abundantly expressed on various human tumour cells. The binding of PD-L1 to PD-1 on activated T cells attenuates antitumour immune surveillance[15]. Moreover, PD-L1 can also interact with CD80 on T cells to deliver inhibitory signals, leading to impaired activation and increased apoptosis[16]. Further research on these intricate binding interactions and cross-talk is crucial for gaining a deeper understanding of immune checkpoint biology and ultimately enhancing the clinical outcomes of immune checkpoint blockade therapy.

In addition to complicated immune checkpoint pathways, our knowledge of the interaction and functional patterns of immune checkpoints is also constantly evolving. The interactions between immune checkpoints have long been considered in a physical cell-to-cell manner[17–20]. However, recent studies including our own, have revealed that PD-L1 is enriched on tumour cell-secreted small extracellular vesicles (sEV) also suppresses T cell responses by disrupting their proliferation, cytokine production and cytotoxicity[17,18,21]. With sizes ranging from 40 to 180 nm, sEVs are cell-derived lipid bilayer-delimited particles that contain the constituents of a cell (i.e. nucleic acids, lipids and proteins)[22]. They can mediate long-distance intercellular communication through the circulatory system[20,22]. Studies have confirmed that sEVs carrying PD-L1 in circulation suppress antitumour immunity independent of cell-to-cell interactions[23], revealing a previously unknown mechanism for immune checkpoint function. An elevated level of circulating PD-L1+ sEVs contributes to intensified and systemic immunosuppression, limiting the effectiveness of immunotherapy[24–27]. However, the mechanism behind the elevated secretion of PD-L1+ sEVs, particularly in anti-PD-1-resistant patients, remains unclear.

Apart from PD-L1, previous study from Whiteside et al. has also discovered the presence of other immune checkpoints on circulating sEVs, including PD-1, CD80 and CTLA-4[28]. However, compared to PD-L1, the functions of other sEV immune checkpoints, especially under the context of immunotherapy, are either unknown or controversial. For instances, a recent study suggests that PD-1+ sEVs alleviate immune suppression[29], while the other proposes that reduce the effectiveness of immunotherapy[30]. A deeper investigation of the function of immune checkpoints on sEVs will improve our understanding of the crosstalk between tumour cells and the immune system, thereby offering valuable insights for predicting and enhancing the clinical outcomes of immune checkpoint blockade therapy. In our present study, we identified a range of inhibitory immune checkpoints in human circulation. It was found that the levels of PD-1/CD80+ sEVs, derived mainly from immunocytes, simultaneously increase in cancer patients' circulation and are closely associated with the poor response to immunotherapy. Mechanistically, PD-1/CD80 carried on immunocyte-derived sEVs (I-sEV) increases secretion of PD-L1+ sEVs into circulation and decreases membrane expression of PD-L1, as well as antigen presentation and intercellular adhesion molecules, on tumour cells, leading to an immunologically 'cold' phenotype. Moreover, we found out that the combination of pretreatment sEV PD-L1, PD-1 and CD80 serves as a superior predictor of patient response to immunotherapy than any

individual checkpoint on circulating sEVs[17,31]. Altogether, the present study reveals that PD-1/CD80 on sEV function via an adaptive loop, contributing to a robust yet previously unknown mechanism that conceals the immunogenicity and suppresses the antitumour immunity. This study also provides important insights for the accurate screening of immunotherapy-sensitive patients and the development of new immunotherapeutic approaches based on checkpoint blockade.

## Results

### Pretreatment levels of PD-1/CD80 on circulating sEVs associate with the response to anti-PD-1 immunotherapy in cancer patients

To gain better insight into the profile of immune checkpoint proteins on small extracellular vesicles (sEV), we first purified sEVs from the peripheral blood samples of 36 healthy donors and 46 patients with head and neck squamous cell carcinoma (HNSCC). The sEVs were characterised by transmission electron microscopy (TEM) as spherical membrane particles (Fig. 1a) and determined by nanoparticle tracking analysis (NTA) to have a mean diameter smaller than 180 nm (Fig. 1b). With a high-resolution nanoparticle flow cytometry platform that can detect nanosized particles (Supplementary Fig. 1), we quantitatively analysed the levels of immune checkpoint proteins on the surface of purified circulating sEVs. The results were visualised in a heatmap and then analysed using random forest machine learning (Fig. 1c). Among all the immune checkpoints tested, PD-1 and CD80 on circulating sEVs were proven to be critically associated with HNSCC by their high variable importance scores (Fig. 1c), in additional to sEV PD-L1 as demonstrated in our previous research[17].

We next investigated the potential association between the immunotherapy response and the levels of circulating sEV PD-1 or CD80. With blood samples from 23 HNSCC patients who underwent anti-PD-1 immunotherapy upon tumour recurrence, we found that nonresponders to anti-PD-1 immunotherapy exhibited a higher baseline level of circulating sEV PD-L1 than responders (Fig. 1d). Moreover, we observed elevated levels of circulating sEV PD-1 or CD80 in the nonresponders (Fig. 1e, f). Correspondingly, the results also revealed that the overall survival rate of anti-PD-1-treated patients with lower levels of PD-1 and CD80 on circulating sEVs was greater than that of those with higher levels (Fig. 1g, h), further suggesting that high circulating sEV PD-1 and CD80 levels were indicative of poor outcomes in patients receiving anti-PD-1 immunotherapy.

To further elucidate the distribution of PD-1 and CD80 carried by sEVs in circulation, we simultaneously labelled surface PD-1 and CD80 on circulating sEVs and identified PD-1+ sEVs, CD80+ sEVs and PD-1+CD80+ sEVs (Fig. 1i). Interestingly, with nanoparticle flow cytometry, we found that more than half of circulating PD-1+ or CD80+ sEVs cocarried both PD-1 and CD80 (Fig. 1j), revealing a synchronised increase in sEV PD-1 and CD80 (hereafter PD-1/CD80) levels in the circulation of cancer patients.

### Activated T cell-derived PD-1/CD80+ sEVs impair the response to anti-PD-1 immunotherapy in vivo

For a better understanding of the fundamental functions, we then investigated the potential cell sources of PD-1/CD80+ sEVs in cancer patients. Considering that the main cell sources of circulating sEVs are haematopoietic cells, endothelial cells and epithelial cells, circulating PD-1+ or CD80+ sEVs were costained with the corresponding cell markers and detected with a nanoparticle flow cytometer. The results showed that the majority of the circulating PD-1+ and CD80+ sEVs in both healthy donors and cancer patients were positive for CD45, suggesting that immunocytes were the main source of PD-1/CD80+ sEVs regardless of tumour development (Supplementary Fig. 2a). We next examined common markers of a series of immunocytes and

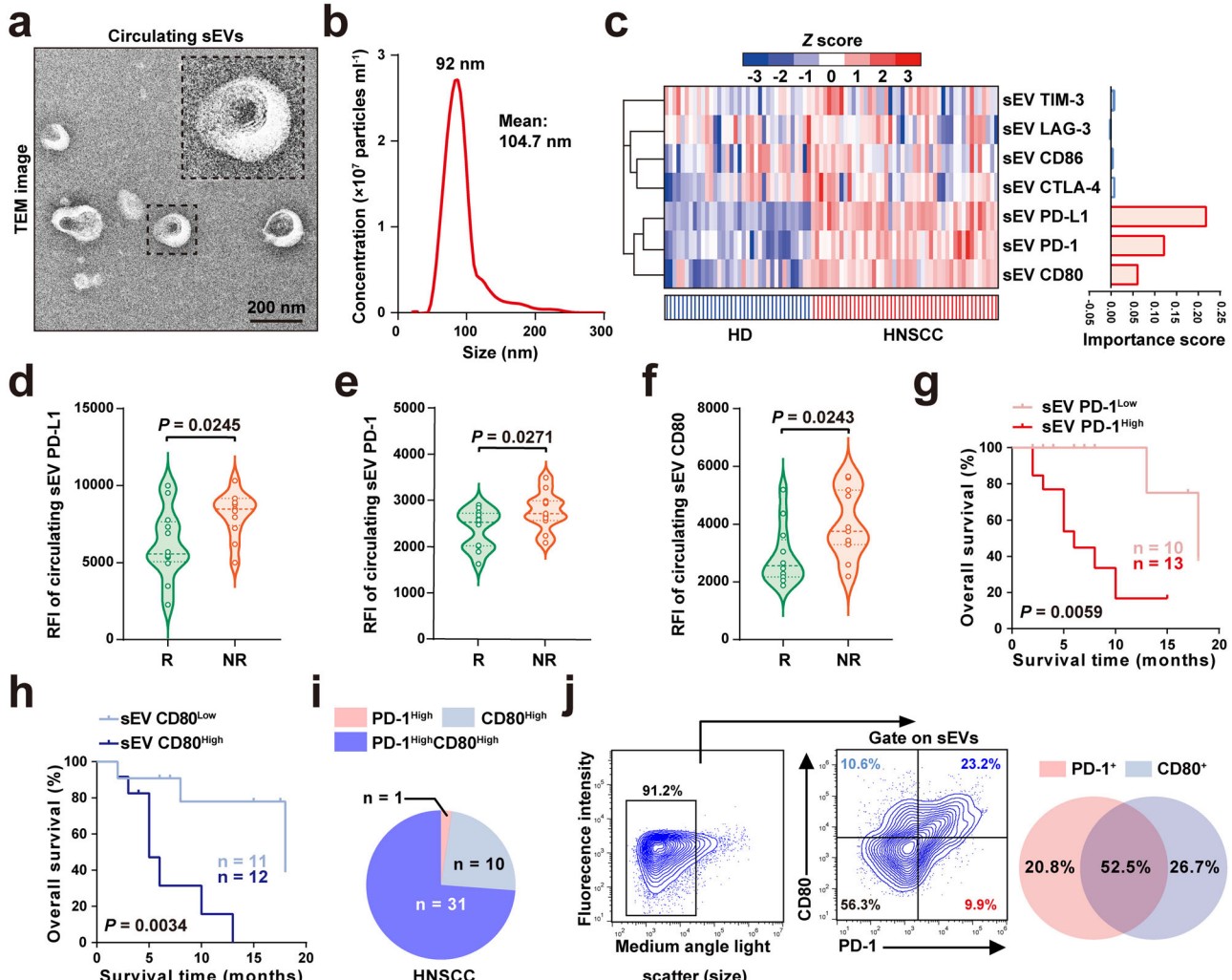

**Fig. 1 | Pretreatment levels of PD-1/CD80 on circulating sEVs associate with the response to anti-PD-1 immunotherapy in cancer patients. a** A TEM image of circulating sEVs from HNSCC patients, showing the cup-shape spherical morphology of sEVs. n = 3 biologically independent samples. Scale bar, 200 nm. sEVs small extracellular vesicles, TEM transmission electron microscope, HNSCC head and neck squamous cell carcinoma. **b** Determination of particle size distribution in purified sEVs using nanoparticle tracking analysis. **c** Heat map illustrating the levels of immune checkpoint proteins (CD80, PD-1, PD-L1, CTLA-4, CD86, LAG-3 and TIM-3) on circulating sEVs from healthy donors (HDs, n = 36) and HNSCC patients (n = 46) were shown on the left. Blue and red hatches indicate HD and HNSCC patients, respectively. Bar plots show bootstrap importance scores based on random forest model that distinguish HDs and HNSCC patients, higher values (red)

represent stronger association with HNSCC (right). Nanoparticle flow cytometry analysis of pretreatment levels of immune checkpoint proteins PD-L1 (**d**), PD-1 (**e**), and CD80 (**f**) on circulating sEVs from responders (R, n = 12) and non-responders (NR, n = 11). Overall survival for HNSCC patients with high and low levels of circulating sEV PD-1 (**g**) and CD80 (**h**). Log-rank test. **i** Pie Chart showing the proportion of circulating sEV PD-1$^{High}$, sEV CD80$^{High}$, and sEV PD-1$^{High}$CD80$^{High}$ in HNSCC patients (n = 46). **j** Nanoparticle flow cytometry analysis of PD-1 and CD80 expression in sEVs after purification. Left, the gating strategy. Right, the Venn diagram illustrating the percentages of PD-1$^+$ sEVs, CD80$^+$ sEVs and PD-1$^+$CD80$^+$ sEVs in HNSCC patients. For (**d**, **e**, **f**) Data were presented as mean ± s.d.; Two-sided *t*-test. The relevant raw data are provided as a Source Data file.

analysed them with random forest modelling. The results suggested that PD-1/CD80$^+$ sEVs secreted by T cells most significantly distinguished HNSCC patients from healthy donors (Supplementary Fig. 2b). The results indicated that T cells are the main source of increased PD-1/CD80$^+$ sEVs levels upon the occurrence and development of cancers. To verify the results, we further performed in vitro assays using sEVs derived from human primary T cells. The levels of sEV PD-1/CD80 in T cells and tumour CAL27 cells were analysed by a nanoparticle flow cytometer and standardised by the Quantum$^{TM}$ MESF (Molecules of Equivalent Soluble Fluorochrome) microsphere kit as previously reported[32–34]. The levels of PD-1/CD80 were markedly enhanced on activated T-cell-derived sEVs (aT-sEV) when compared to sEVs derived from nonactivated T cells, but PD-1/CD80 was barely detected in sEVs from epithelial tumour cells (E-sEV) (Supplementary Fig. 2c). We then performed iodixanol density gradient centrifugation

with aT-sEVs (Supplementary Fig. 2d), proving the accumulation of PD-1/CD80 together with the marker proteins of sEVs. Consistently, TEM also verified the extrafacial presence of PD-1/CD80 on aT-sEVs (Supplementary Fig. 2e).

Next, we set out to verify the potential role of PD-1/CD80$^+$ aT-sEVs in anti-PD-1 therapy in vivo. C57BL/6 mice were challenged with M38 murine colorectal cancer cells or B16F10 murine malignant melanoma cells 12 days before the administration of sEVs (Fig. 2a). The results showed that the tail-vein injection of aT-sEVs almost doubled tumour growth and shortened the lifespan of mice in vivo, which was rescued by preblocking PD-1/CD80 on aT-sEVs (Fig. 2b–d and Supplementary Fig. 3a, b). To further explore the impact of aT-sEV PD-1/CD80 on the efficacy of anti-PD-1 immunotherapy, we utilised mice bearing MC38 tumours as an appropriate cancer model due to their ability to elicit a moderate immune response[35–37]. Mice were treated with anti-PD-1

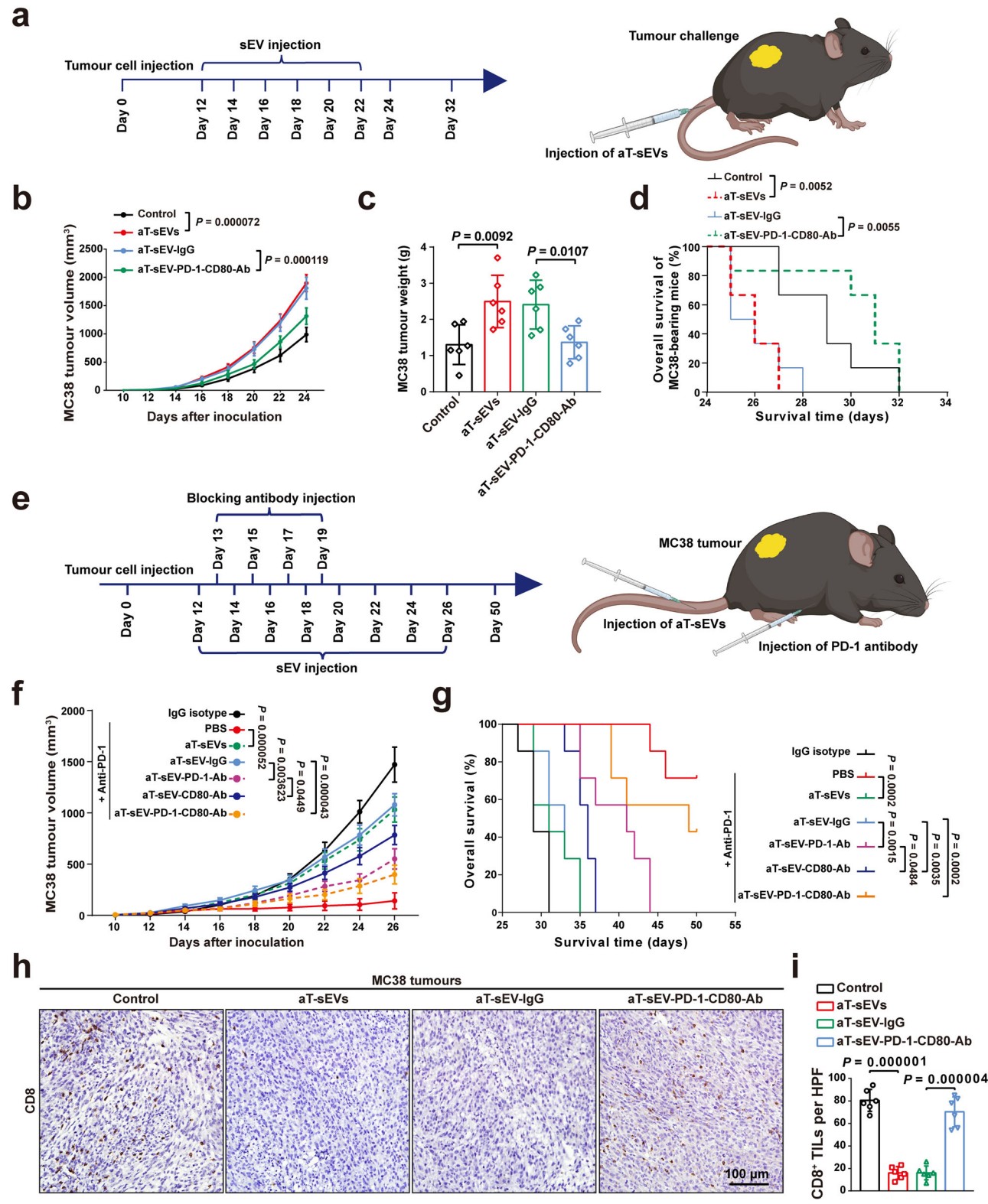

antibodies in the presence or absence of sEV injections (Fig. 2e). The results proved that treatment with anti-PD-1 antibodies significantly inhibited the growth of MC38 tumours (Fig. 2f) and prolonged the lifespan of mice (Fig. 2g). However, tail-vein injections of aT-sEVs obviously attenuated the immunotherapeutic effects of anti-PD-1 antibodies (Fig. 2f, g). Importantly, injection of aT-sEVs with their surface PD-1/CD80 blocked did not significantly altered the effects of anti-PD-1 treatment (Fig. 2f, g), suggesting the functional importance

of PD-1/CD80 on the sEVs. Given the fact that PD-1/CD80+ aT-sEVs impaired anti-PD-1 blockade therapy in vivo, we then investigated the association between PD-1/CD80+ aT-sEVs and tumour-infiltrating CD8 T lymphocytes (TILs), which are an important factor of the antitumour immunity. We revealed that the number of CD8+ TILs in MC38 xenografts was reduced by treatment with aT-sEVs, while blockade of PD-1 and CD80 on aT-sEVs attenuated the reduction (Fig. 2h, i). Similar results were obtained in mice bearing B16F10 tumours (Supplementary

**Fig. 2 | Activated T cell-derived PD-1/CD80⁺ sEVs impairs the response to anti-PD-1 immunotherapy in vivo. a** Scheme for the experimental setup of tumour xenograft mice model. $5 \times 10^5$ MC38 cells or $1.5 \times 10^5$ B16F10 cells were subcutaneously injected into flanks of 8-week-old female C57BL/6 mice. Tail vein injections of sEVs (100 μg) were performed every 2 days from days 12 to 22. For blocking PD-1 or CD80, sEVs were pretreated with corresponding blocking antibodies. IgG isotype was used as control. **b** Growth curve of MC38 tumours in C57BL/6 mice with indicated treatments (n = 6 mice per group). **c** Weights of MC38 tumours in C57BL/6 mice with indicated treatments (n = 6 mice per group). **d** Overall survival for C57BL/6 mice bearing MC38 tumours with indicated treatments (n = 6 mice per group). Log-rank test. **e** Scheme for the experimental setup of the MC38 tumour xenograft mice model with anti-PD-1 blockade antibodies

treatment. Anti-PD-L1 was intraperitoneally administered (200 μg) every 2 days from days 13 to 19. An isotype IgG antibody was used as the control. **f** Growth curve of MC38 tumours in C57BL/6 mice with indicated treatments (n = 7 mice per group). **g** Overall survival of C57BL/6 mice bearing MC38 tumours with indicated treatments. Log-rank test. **h, i** Immunohistochemistry analysis of CD8⁺ T cell infiltration in MC38 tumour sections from C57BL/6 mice with indicated treatments. Representative images (**h**) of CD8 staining in tumour sections. Scale bar, 100 μm. Quantification analysis (**i**) of infiltrated CD8⁺ T cells in MC38 tumours (n = 6 mice per group). TILs, tumour infiltrating lymphocytes. For (**b, f**) data were presented as mean ± s.d.; Two-way ANOVA. For (**c, i**) data were presented as mean ± s.d.; Two-sided *t*-test. Source data are provided as a Source Data file.

Fig. 3c). Altogether, the above results suggest that PD-1/CD80⁺ aT-sEVs derived from activated T cells may impair the response to anti-PD-1 immunotherapy.

### The levels of circulating PD-1/CD80⁺ sEVs were positively correlated with levels of PD-L1⁺ sEVs in patients with PD-L1-positive tumour cells

Our previous study in metastatic melanoma and the present findings in HNSCC have showed that elevated levels of circulating PD-L1⁺ sEVs, the interaction protein for both PD-1 and CD80, are also associated with the anti-PD-1 response and survival rate (Fig. 1d and Supplementary Fig. 4a). Therefore, we investigated the correlation between the levels of circulating PD-L1⁺ sEVs and PD-1/CD80⁺ sEVs in all patients but found no significant link (Supplementary Fig. 4b). However, interestingly, when we divided the patients into two groups according to the positivity of tumour cell surface PD-L1 (Fig. 3a), the results showed that the levels of circulating PD-1/CD80⁺ sEVs positively correlated with those of PD-L1⁺ sEVs in patients with PD-L1-positive tumour cells (Fig. 3b). The correlation was not statistically significant in patients who were negative for tumour cell PD-L1 (Fig. 3c). Consistently, we also found that treatment with aT-sEVs stimulated the level of circulating PD-L1⁺ sEVs in both MC38- (Fig. 3d) and B16F10-bearing mice (Supplementary Fig. 4c). However, the increased circulating PD-L1⁺ sEVs levels induced by aT-sEVs was suppressed by the preincubation of aT-sEVs with anti-PD-1 and anti-CD80 antibodies (Fig. 3d and Supplementary Fig. 4c). Additionally, aT-sEV treatment failed to increase the circulating PD-L1⁺ sEV levels of mice bearing B16F10 tumours with *Pd-l1* knockout (Supplementary Fig. 4d).

Next, we investigated the association between aT-sEV PD-1/CD80 and tumour cell PD-L1 expression. The results from animal studies revealed that aT-sEVs significantly decreased the expression level of membrane PD-L1 in both MC38 and B16F10 tumours. The tissue sections from MC38 xenografts were analysed with immunohistochemistry after application of aT-sEVs, revealing that tumour cell membrane PD-L1 expression was downregulated (Fig. 3e, f). For B16F10-bearing mice, tumour cells were harvested for flow cytometry analysis of PD-L1, demonstrating that treatment with aT-sEVs significantly reduced the expression of cell membrane PD-L1 (Supplementary Fig. 4e). However, the cell membrane PD-L1 expression reduced by aT-sEVs could be rescued by preblocking PD-1 and anti-CD80 on aT-sEVs (Fig. 3e, f and Supplementary Fig. 4e). These results suggest that PD-1/CD80⁺ sEVs potentially contribute to the increased circulating PD-L1⁺ sEV levels and the decreased tumour cell PD-L1 expression levels, which may be closely associated with resistance to anti-PD-1 therapy.

### PD-1/CD80 on aT-sEVs transligated and internalised tumour cell surface PD-L1 for sEV secretion

The above results suggest a close association between tumour cell PD-L1 and its binding partners PD-1/CD80 on aT-sEVs. Therefore, with Förster resonance energy transfer (FRET), we first determined whether sEV PD-1/CD80 was transligated to tumour PD-L1 by

evaluating their molecular proximity. To this end, tumour cells were incubated with anti-PD-L1 antibody labelled with tyramide594 (energy acceptor), while aT-sEVs were incubated with anti-PD-1 or anti-CD80 antibodies labelled with AF488 (energy donor). After coculture of the sEVs and tumour cells, immunofluorescence images were captured by exciting PD-1-AF488 or CD80-AF488 at 488 nm and PD-L1-tyramide594 at 594 nm, respectively. When FRET occurs, the excitement energy is transferred from the donor to the acceptor. Moreover, attenuating the transfer of energy by bleaching the acceptor could enhance the donor signal. Here, the photobleaching of PD-L1-tyramide594 increased the fluorescence of PD-1-AF488-labelled aT-sEVs (Fig. 4a), which was indicative of FRET. Similarly, the fluorescence of CD80-AF488-labelled aT-sEVs was also increased after the photobleaching of PD-L1-tyramide594 (Fig. 4b). These results suggest the transligation of aT-sEV PD-1/CD80 to tumour cell membrane PD-L1 in vitro.

To assess the downstream effect of transligation between aT-sEV PD-1/CD80 and tumour cell PD-L1, we then evaluated the expression level of tumour PD-L1 following aT-sEV treatment in a panel of tumour cell lines, including human oral cancer, melanoma, breast cancer and lung cancer cell lines. The flow cytometry results revealed that in all the tested cell lines, the level of membrane surface PD-L1 was significantly reduced by the treatment with aT-sEVs, which was rescued by preblocking PD-1 and CD80 on aT-sEVs (Fig. 4c and Supplementary Fig. 5a–c). This was consistent with the findings in animal studies. To elucidate the underlying mechanism, we first excluded the regulation of membrane PD-L1 at the transcriptional level, since no significant change in the mRNA level of PD-L1 was found after the treatment with aT-sEVs (Supplementary Fig. 5d), suggesting potential regulation at the protein level. Then, we evaluated the distribution of PD-L1 in aT-sEV-treated tumour cells and revealed significantly decreased membrane PD-L1 levels and increased cytoplasmic PD-L1 levels, indicating the enhanced internalisation of membrane PD-L1 as early as 30 min after treatment with aT-sEVs (Fig. 4d, e). Intriguingly, the internalisation of aT-sEVs was not comparable to that of membrane PD-L1, as evidenced by the observation that the number of membrane-bound CFSE-labelled aT-sEVs was not obviously decreased at the same time (Fig. 4d, e and Supplementary Fig. 5e). Additionally, we further demonstrated the increased colocalization of internalised PD-L1 with EEA1 (Early Endosome Antigen 1) and RAB 7, which are markers for early and late endosomes, respectively, in aT-sEV-treated tumour cells (Fig. 4f). This finding suggested that aT-sEVs enhanced PD-L1 trafficking through the endosomal pathway.

Subsequently, we also revealed the colocalization of PD-L1 with CD63, a marker protein of sEVs, by immunofluorescence staining in tumour cells and found that it was promoted after the treatment with aT-sEVs (Fig. 4g), suggesting the enhanced secretion of PD-L1 via sEVs. This finding is highly consistent with the close correlation between PD-1/CD80⁺ sEVs and PD-L1⁺ sEVs, as revealed in patient samples and animal models (Fig. 3b, d). Therefore, we asked whether aT-sEVs contributed to the downregulation of tumour cell surface PD-L1

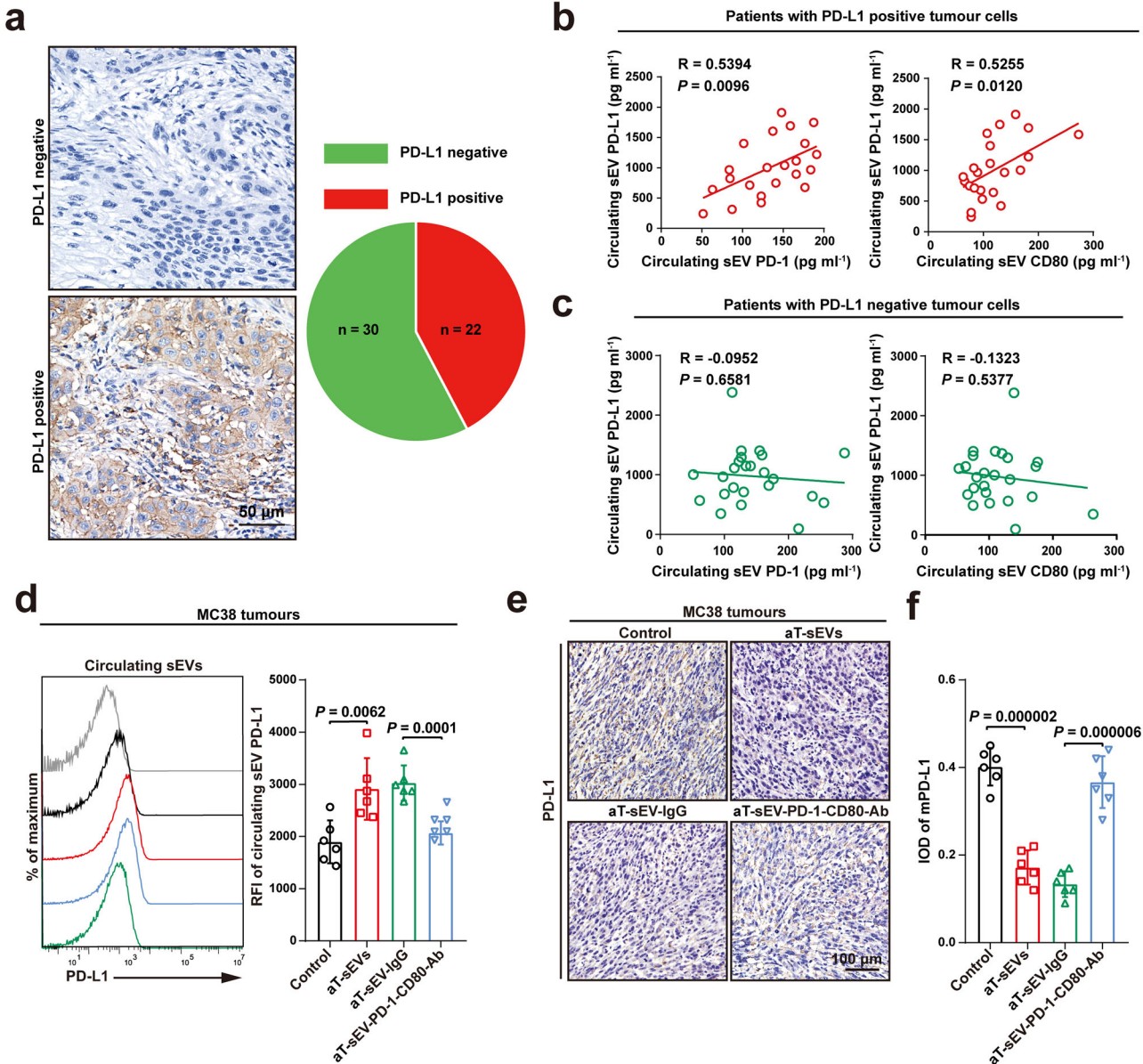

**Fig. 3 | The levels of circulating PD-1/CD80⁺ sEVs were positively correlated with that of PD-L1⁺ sEVs in patients with PD-L1-positive tumour cells.** **a** Immunohistochemistry analysis of PD-L1 expression in HNSCC patient with PD-L1 positive or negative tumours. Left, representative images of PD-L1 staining in biopsies of HNSCC patients. Scale bar, 50 μm. Right, pie chart showed the proportion of patients with either PD-L1 negative tumour cells (green, n = 30) and patients with PD-L1 positive tumour cells (red, n = 22). Pearson correlation analysis of the level of circulating PD-L1⁺ sEVs with circulating sEV PD-1 and CD80 in patients with PD-L1 positive (n = 22) (**b**) and PD-L1 negative (**c**) tumour cells (n = 24). **d** Flow cytometry profiles (left) and quantification of relative fluorescence intensity (RFI) (right) of circulating sEV PD-L1 levels in mice MC38-bearing C57BL/6 mice with indicated treatment (n = 6 mice per group). **e, f** Immunohistochemistry analysis of PD-L1 expression in MC38 tumour sections from C57BL/6 mice with indicated treatments. Representative images (**e**) of PD-L1 staining in tumour cells. Scale bar, 100 μm. Quantification analysis (**f**) of membrane expression of PD-L1 in MC38 tumour cells (n = 6 mice per group). For (**d, f**) data were presented as mean ± s.d.; Two-sided *t*-test. Source data are provided as a Source Data file.

expression by promoting secretion of PD-L1⁺ sEVs. To this end, the PD-L1⁺ sEV secretion was tested in a panel of tumour cell lines treated with aT-sEVs. The results from high-resolution nanoparticle flow cytometry revealed that treatment with aT-sEVs significantly increased the secretion level of PD-L1⁺ sEVs in human oral cancer, melanoma, breast cancer and lung cancer cell lines (Fig. 4h and Supplementary Fig. 5f–h). The increased sEV PD-L1 secretion in aT-sEV-treated tumour cells was further confirmed by western blotting (Fig. 4i). Taken together, these results together suggest that PD-1/CD80 carried on aT-sEVs might transligate and internalise tumour cell surface PD-L1 to enhance its secretion via sEVs.

## PD-1/CD80 on aT-sEVs led to adaptive immunosuppression and cooled down the tumours to an immunologically cold phenotype

Our previous study revealed that the secretion of PD-L1⁺ sEVs by tumour cells could also be triggered by IFN-γ[17]. Therefore, we investigated the potential difference between the effects of IFN-γ and PD-1/CD80⁺ sEVs, both of which are mainly produced by stimulated immunocytes (e.g. CD8⁺ T cells). The results revealed that IFN-γ and aT-sEVs promoted the secretion of PD-L1⁺ sEVs by tumour cells at similar rates (Fig. 5a, b). To gain a more comprehensive understanding of the roles of IFN-γ and aT-sEV PD-1/CD80 in regulating tumour sEV

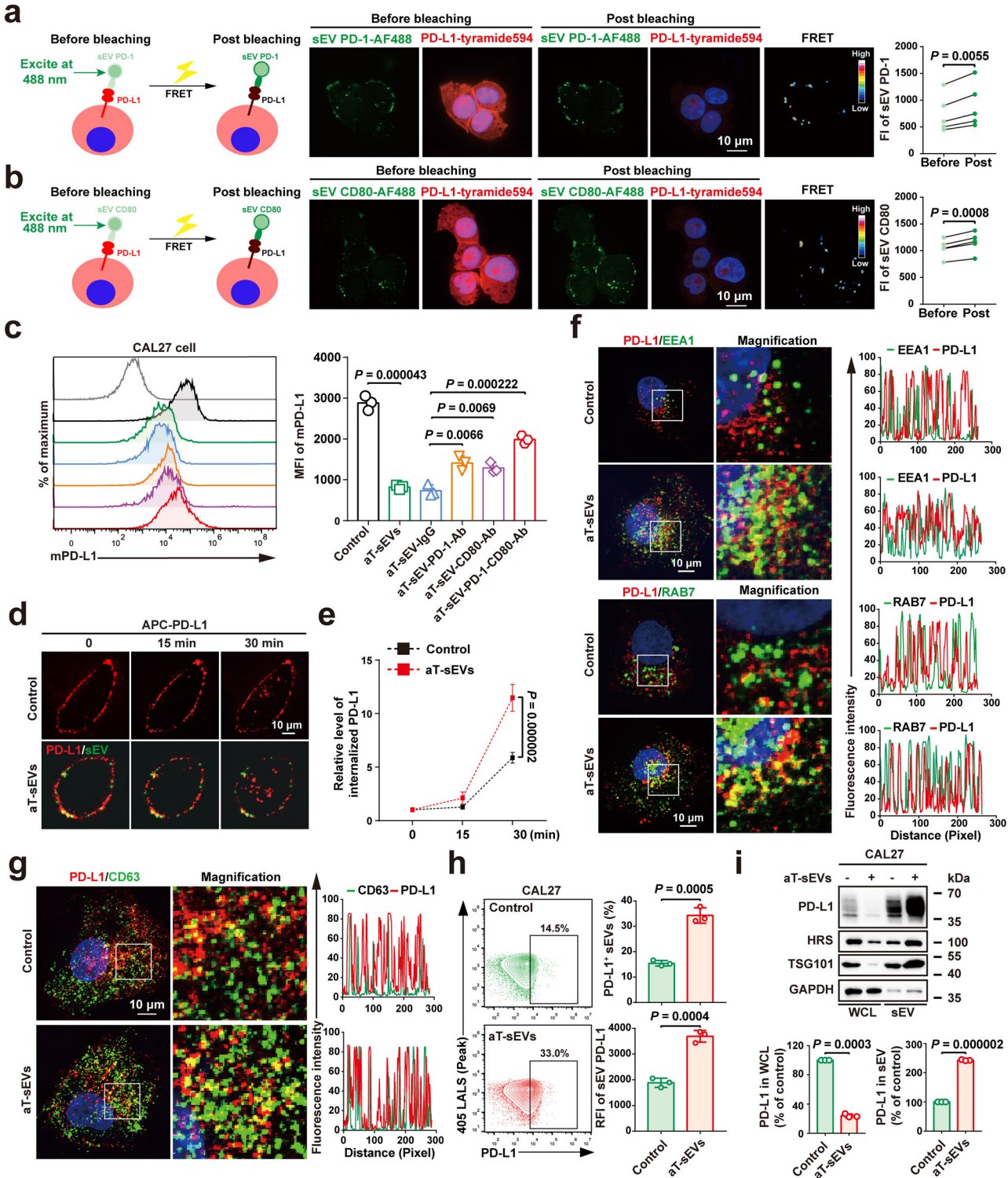

secretion, sEVs derived from tumour cells treated with IFN-γ or aT-sEVs were collected for mass spectrometry (Supplementary Fig. 6a). As the result of MaxQuant analysis, 280 proteins were upregulated and 203 were downregulated in sEVs from IFN-γ-treated tumour cells, while 263 proteins were upregulated and 105 were downregulated in sEVs from tumour cells treated with aT-sEVs (Supplementary Fig. 6b). Differentially expressed proteins were subjected to Gene ontology (GO) enrichment analysis, finding out that the upregulated proteins in sEVs secreted by IFN-γ-stimulated tumour cells were enriched in nucleic acid processing and translation (Supplementary Fig. 6c), while treatment with aT-sEVs led to enhanced EV biogenesis and trafficking in

tumour cells (Supplementary Fig. 6d). For the upregulated proteins in sEVs from both IFN-γ- and aT-sEV-treated CAL27 cells, it was found that the adhesion molecule ICAM-1 was also enriched by both IFN-γ and aT-sEVs treatment (Fig. 5c), as verified by nanoparticle flow cytometry (Fig. 5d). In addition, the mass spectrometry results also showed that both IFN-γ and aT-sEVs significantly increased the levels of MHC-I molecules in sEVs secreted by tumour cells (Fig. 5c). However, as the control, treatment with liposomes or cT-sEVs failed to initiate the substantial changes mentioned above (Supplementary Fig. 6e). The simultaneous upregulation of tumour sEV PD-L1, MHC-I and ICAM-1 by both IFN-γ and aT-sEVs indicates intensive immunosuppression since it

**Fig. 4 | PD-1/CD80 on aT-sEVs transligated and internalised tumour cell surface PD-L1 for sEV secretion.** An acceptor-photobleaching FRET assay showing the molecular nearness of PD-1 (**a**) or CD80 (**b**) on sEVs surface and PD-L1 on the tumour cell membrane. Left, scheme for experimental setup. Middle, Confocal microscopy images showing PD-1 (**a**) or CD80 (**b**) (green) on sEVs and PD-L1 (red) on CAL27 cells. The difference in fluorescence intensity of the energy donor before and after photobleaching was evaluated as the level of FRET efficiency. Right, the quantification analysis of the difference in fluorescence intensity (FI). Scale bar, 10 μm. **c** Flow cytometry profiles (left) and quantification of mean fluorescence intensity (MFI) (right) of membrane PD-L1 levels in CAL27 cells with or without aT-sEV treatment. **d**, **e** Immunofluorescence staining of PD-L1 in CAL27 cells after incubation with CFSE-labelled aT-sEVs for 0, 15 and 30 min (**d**). Scale bar, 10 μm. Quantification analysis of the relative levels of cytoplasmic PD-L1 in CAL27 cells after incubation with CFSE-labelled aT-sEVs (**e**). **f** Immunofluorescence images of CAL27 cells staining with PD-L1 (red) and EEA or RAB7 (green) treated with or without aT-sEVs. The fluorescence intensity profiles are plotted on the right. The nuclei were counter-stained with DAPI (blue). n = 3 biologically independent samples. Scale bar, 10 μm. **g** Confocal images of CAL27 cells staining with PD-L1 (red) and CD63 (green) treated with or without aT-sEVs. The nuclei were counter-stained with DAPI (blue). The fluorescence intensity profiles are plotted on the right. n = 3 biologically independent samples. Scale bar, 10 μm. **h** Flow cytometry profiles (left) and quantification of proportion (right, top) and relative fluorescence intensity (RFI) (right, bottom) of sEV PD-L1 in supernatants from CAL27 cells with or without aT-sEV treatment. **i** Western blot analysis of PD-L1 expression level in WCL and sEVs from CAL27 cells after aT-sEV treatment (top). Quantification analysis of relative PD-L1 levels in WCL and sEVs (bottom). WCL whole cells lysate. The samples derive from the same experiment and that gels/blots were processed in parallel. For (**a**, **b**, **c**, **e**, **h**, **i**) data were presented as mean ± s.d.; n = 3 biologically independent samples; Two-sided *t*-test. For (**e**) data were presented as mean ± s.d.; Two-way ANOVA. The relevant raw data and uncropped blots are provided as a Source Data file.

has been proven that coexpressed MHC-I and ICAM-1 would cooperate with sEV PD-L1 to strongly suppress CD8[+] T cells[19,38]. Interestingly, we noticed a more significant increase in the level of the 'do not eat me signal' CD47 in tumour cell-secreted sEVs after treatment with aT-sEVs compared to those in tumour cell-secreted sEVs after treatment with IFN-γ (Fig. 5c and Supplementary Fig. 6f), suggesting that aT-sEVs might result in low clearance rate of tumour sEVs by macrophages. Neither IFN-γ nor aT-sEVs affected the secretion of EGFR[+] sEVs in tumour cells (Fig. 5c and Supplementary Fig. 6g), indicating that PD-1/CD80[+] sEVs possibly participated in the specific mediation of immune-related sEV secretion in tumours.

After studying the influence of IFN-γ and aT-sEVs on tumour cell-secreted sEVs, we then tested the effects these factors on the tumour cells themselves. The results revealed that, unlike the dramatically decreased level of tumour cell surface PD-L1 induced by aT-sEVs, IFN-γ significantly increased the level of tumour cell surface PD-L1 (Supplementary Fig. 6h), suggesting that IFN-γ and aT-sEVs might function in different ways. Thus, we then investigated the potential effects of IFN-γ and aT-sEVs on tumour cells by mass spectrometry. Immune-related proteins were first analysed, and we revealed that IFN-γ increased not only tumour cell PD-L1 but also antigen peptide transporter 1 (TAP1), HLA-A, HLA-B, HLA-C and ICAM-1 (Fig. 5e), indicating strong adhesion and enhanced immunogenicity. In contrast, aT-sEVs downregulated the expression of PD-L1, TAP1, HLA-A, HLA-B, HLA-C and ICAM-1 in tumour cells (Fig. 5e). Neither liposomes nor cT-sEVs induced obvious changes in the abovementioned immune-related proteins (Supplementary Fig. 6i). The results were further verified by flow cytometry, and IFN-γ treatment, in accordance with previous reports, upregulated the membrane expression of both MHC-I and ICAM-1 on tumour cells (Fig. 5f, g). However, aT-sEVs downregulated the membrane expression of MHC-I and ICAM-1 (Fig. 5f, g). To investigate whether these effects induced by aT-sEVs were associated with PD-1/CD80 on sEVs, anti-PD-1 and anti-CD80 antibodies were used to block PD-1 and CD80 on aT-sEVs before treating tumour cells. As a result, blocking PD-1 and CD80 on aT-sEVs nearly abrogated their function in regulating tumour cell membrane PD-L1, MHC-I and ICAM-1 (Supplementary Fig. 7a). Importantly, we further revealed the negative correlation of circulating sEV PD-1/CD80 levels with ICAM-1, MHC-I and TAP1 levels in patients with HNSCC (Supplementary Fig. 7b). We then asked what would happen if tumour cells were treated with IFN-γ and aT-sEVs at the same time. The results revealed that, even in the presence of IFN-γ, treatment with aT-sEVs could still decrease the levels of membrane PD-L1, MHC-I and ICAM-1 on tumour cells (Fig. 5h). The above findings suggest a more prominent role of PD-1/CD80[+] sEVs in initialising adaptive immune resistance than IFN-γ, which is supported by data showing increased circulating PD-L1[+] sEVs that systematically threaten immune integrity and concealed membrane immunogenetic molecules that locally facilitate immune escape (Fig. 5i).

## The ESCRT machinery in tumour cells contributed to the enhanced PD-L1[+] sEV secretion initiated by PD-1/CD80 on aT-sEVs

To further understand the mechanisms underlying the protein trafficking of PD-L1 in tumour cell subsequent to PD-1/CD80[+] sEVs stimulation, we had differentially expressed proteins from Supplementary Fig. 8a, b subjected to Kyoto Encyclopedia of Genes and Genomes (KEGG) and GO analysis. As shown in Supplementary Fig. 8c, according to KEGG analysis, upregulated proteins in tumour cells treated with aT-sEVs were significantly enriched in the endocytic pathway, indicating activated endocytosis in tumour cells following the binding of aT-sEVs. However, the JAK-STAT signalling pathway was mainly enriched in IFN-γ-treated tumour cells (Supplementary Fig. 8d), consistent with previous studies[39]. In addition, based on enriched GO terms, the main downstream effects caused by IFN-γ were metabolic processes and immune responses (Supplementary Fig. 8e). It was further verified by real-time PCR that the mRNA levels of *HLA-A, HLA-B, HLA-C, ICAM-1 and PD-L1* were increased in CAL27 cells treated with IFN-γ, but they were not significantly changed after treatment with aT-sEVs (Supplementary Fig. 8f). In tumour cells treated with aT-sEVs, we revealed the potentially activated transportation of proteins and vesicles (Fig. 6a). Moreover, a panel of proteins in regulating vesicle transport was enriched and the expression levels of most of these proteins changed, although to a different extent, in tumour cells upon treatment with aT-sEVs (Fig. 6b). Among them, the most significant change was the downregulation of subunits belonging to the endosomal sorting complex required for transport 0 (ESCRT-0) and ESCRT-1, which are intensively involved in the cargo sorting of sEVs. The levels of EEA1 and some RAB family proteins, which are responsible for vesicle transport, were increased after treatment with aT-sEVs (Fig. 6b). Intriguingly, the results showed that IFN-γ did not significantly affect the expression of ESCRT members (Supplementary Fig. 8g).

Among the panel of ESCRT members that were significantly regulated by aT-sEVs, the ESCRT-0 subunit HRS played a pivotal role in the generation of multivesicular bodies and recruitment of target proteins to endosomes. More importantly, we previously demonstrated a direct role of HRS in regulating the secretion of PD-L1[+] sEVs in melanoma and HNSCC cells. In the present study, the role of HRS in mediating the secretion of PD-L1[+] sEVs was confirmed by western blotting, as aT-sEV treatment decreased the protein level of HRS in whole-cell lysates while increasing its counterpart in sEVs, which was similar to the transfer of PD-L1 from cells to secreted sEVs after aT-sEV treatment (Fig. 4i). Additionally, according to the immunoprecipitation and immunofluorescence experiments, treatment with aT-sEVs significantly induced the colocalization of PD-L1 with HRS and resulted in the sEV secretion of PD-L1 in tumour cells

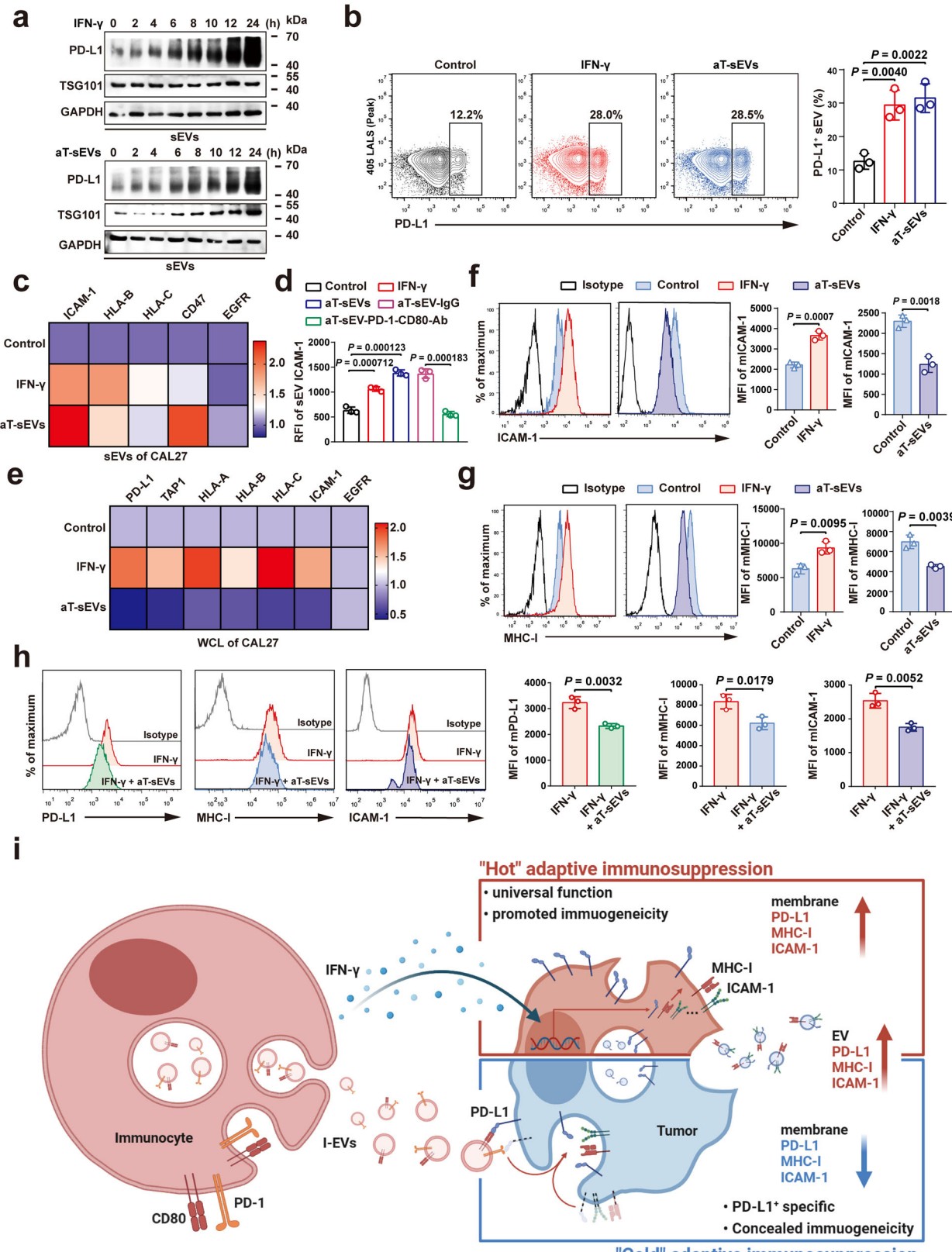

(Fig. 6c, d). Moreover, knockdown of *HRS* in tumour cells dramatically prevented the aT-sEV-stimulated secretion of PD-L1⁺ sEVs (Fig. 6e). Importantly, B16F10 cells with stable knockout of *Hrs* (Fig. 6f) were subcutaneously injected subcutaneously into the flanks of C57BL/6 mice. Notably, *Hrs* knockout in tumour cells

significantly decreased circulating PD-L1⁺ sEV levels (Fig. 6g) and alleviated tumour growth induced by treatment with aT-sEVs (Fig. 6h, i). These results suggest that the ESCRT machinery in tumour cells may contribute to the enhanced PD-L1⁺ sEV secretion induced by PD-1/CD80 on aT-sEVs.

**Fig. 5 | PD-1/CD80 on aT-sEVs led to adaptive immunosuppression and entirely cooled down the tumours to an immunologically cold phenotype. a** Western blot analysis of PD-L1 expression in sEVs from CAL27 cells treated with IFN-γ (top) or aT-sEV (bottom) for 0, 2, 4, 6, 8, 10, 12, and 24 h. **b** Flow cytometry profiles (left) and quantification of proportion (right) of PD-L1$^+$ sEVs in supernatants from CAL27 cells with IFN-γ or aT-sEVs. **c** Heat map illustrating relative levels of ICAM-1, HLA-B, HLA-C, CD47, and EGFR in CAL27 cell-derived sEVs after with indicated treatments. **d** Quantification analysis of flow cytometry showing the levels of ICAM-1 in CAL27 cell-derived sEVs (presented as RFI) after with indicated treatments. **e** Heat map illustrating relative levels of PD-L1, TAP1, HLA-A, HLA-B, HLA-C, ICAM-1, and EGFR in

CAL27 cells after indicated treatments. Flow cytometry profiles (left) and quantification of membrane MFI (right) of ICAM-1 (**f**) and MHC-I (**g**) in CAL27 cells with aT-sEV or IFN-γ treatment. mMHC-I membrane MHC-I, mICAM-1 membrane ICAM−1. **h** Representative flow cytometric histograms (left) and quantification analysis (right) of the membrane expression levels of PD-L1, ICAM−1 and MHC-I in CAL27 cells treated with IFN-γ and aT-sEV simultaneously. mPD-L1 membrane PD-L1. **i** Schematic of the immunogenicity immunosuppression patterns led by IFN-γ and aT-sEVs. For (**b**, **d**, **f**, **g**, **h**) data were presented as mean ± s.d.; n = 3 biologically independent samples; Two-sided *t*-test. The relevant raw data and uncropped blots are provided as a Source Data file.

## Synchronous analysis of multiple checkpoints on circulating sEVs is superior in predicting clinical responses to anti-PD-1 immunotherapy

The above results suggest an unprecedented mechanism by which multiple immune checkpoints on sEVs function via an adaptive loop to suppress the antitumour immunity and disturb immunotherapy responses. Therefore, we speculated that a strategy combining multiple immune checkpoints, such as PD-1/CD80 and PD-L1, might help to predict immunotherapy response. To this end, we studied 23 HNSCC patients who underwent anti-PD-1 immunotherapy. With the calculated cut-offs, sEV PD-L1, PD-1 and CD80 alone could to some extent stratify responders from nonresponders, but the sensitivity and specificity were not as high as expected (Fig. 7a, b). When combining the levels of PD-L1 and PD-1/CD80 on circulating sEVs, the patients were mainly divided into three groups: PD-1/CD80$^{low}$, PD-1/CD80$^{high}$-PD-L1$^{low}$ and PD-1/CD80$^{high}$-PD-L1$^{high}$ (Fig. 7c). All of the sEV PD-1/CD80$^{low}$ patients we studied showed responses to immunotherapy. Additionally, most of the patients with sEV PD-1/CD80$^{high}$-PD-L1$^{high}$ were nonresponders. For patients with sEV PD-1/CD80$^{high}$-PD-L1$^{low}$, the situation could be complicated (Fig. 7c). We hypothesised that other factors might be involved. To prove this hypothesis, factors including PD-L1 expression, IFN-γ concentration and tumour burden were fully studied in the 6 patients with PD-1/CD80$^{high}$-PD-L1$^{low}$ sEVs. With random forest classification, the results indicated that the tumour proportion score (TPS) of PD-L1 could distinguish nonresponders from responders in this patient subgroup (Fig. 7d). Using immunohistochemistry, we also found that all patients with a negative TPS of PD-L1 in tumour cells were nonresponders (Fig. 7e), further proving that downregulated PD-L1 expression contributes to immunotherapy resistance. Based on these findings, we proposed a workflow applicable to clinical practice for stratifying immunotherapy responders from nonresponders (Fig. 7f).

## Discussion

In the present study, we unveiled a pivotal role of small extracellular vesicle (sEV) immune checkpoints in antitumour immunity. Although the presence of PD-1 and CD80 in sEVs was discovered early in 2017, their precise functions in regulating the immune system are still unclear[28]. Recently, the once overlooked roles of PD-1$^+$ sEVs in antitumour immunity has received increasing attention from researchers. A previous study revealed that PD-1$^+$ sEVs secreted by activated T cells could interact with cell surface PD-L1, potentially interfering with PD-L1-induced suppression of T cells, which was similar to the effects of anti-PD-L1 blocking antibodies[29]. However, most of the conclusions in this study were drawn from an in vitro coculture system of tumour cells and cytotoxic T cells, neglecting the adaptive mechanisms in the context of an intact immune system. In contrast, a more recent study suggested a negative role of PD-1$^+$ sEVs in antitumour immunity, as a higher level of circulating PD-1$^+$ sEVs in melanoma patients was significantly correlated with a poorer clinical response to immunotherapy and survival rate, but the underlying mechanism was not elucidated[30]. Unlike the report that identified melanoma cells as one

of the main sources of PD-1$^+$ sEVs, we rarely observed membrane expression or sEV secretion of PD-1 in HNSCC cells. Moreover, PD-1 and CD80 levels can differ between various immunocytes[15,40,41], potentially mirroring their parental cell variations across sEVs derived from different immunocytes. In other words, certain immunocytes might release sEVs with elevated PD-1 levels, while others may carry higher CD80 levels. Consequently, sEV PD-1 and sEV CD80 might exert predominant effects in distinct situations. In any case, it's reasonable to speculate that the coexistence of PD-1 and CD80 on sEVs could lead to synergistic effects.

A previous study revealed that PD-1$^+$ sEVs from T and B cells and PD-L1$^+$ sEVs from melanoma cells were independent biomarkers for predicting immunotherapy response, according to multivariant Cox-hazard regression analyses for both overall and progression-free survivals[30]. Conversely, we proved that circulating PD-1/CD80$^+$ sEVs highly correlate with PD-L1$^+$ sEVs, highlighting their substantial role in promoting the secretion of PD-L1$^+$ sEVs. A recent study reported the endocytosis of membrane PD-L1 after treatment with PD-1$^+$ sEVs, hypothesising that the internalised PD-L1 would end in degradation, although this notion was not experimentally confirmed[29]. However, we revealed that the internalised tumour membrane PD-L1 induced by PD-1/CD80$^+$ sEVs did not ended in proteasome- or lysosome-mediated degradation (Supplementary Fig. 9), but mainly underwent secretion of PD-L1$^+$ sEVs instead. Clinically, we observed that patients with PD-L1$^+$ tumour cells and high levels of circulating PD-1/CD80$^+$ sEVs but low levels of PD-L1$^+$ sEVs responded well to immunotherapy. For this situation, we hypothesised that up-regulation of membrane PD-L1 in the patients with sEV PD-1/CD80$^{high}$-PD-L1$^{low}$ might reflect the insensitivity of tumour cells to I-sEVs, leading to potentially enhanced responses to immunotherapy. To our knowledge, no similar approach has been exploited in prognosis prediction for immunotherapy based on secretion of I-sEVs.

IFN-γ can increase the secretion of PD-L1$^+$ sEVs[17], thereby leading to adaptive and systemic immunosuppression. However, IFN-γ also enhanced the efficiency of antitumour immunity by increasing immunogenicity[39]. However, PD-1/CD80$^+$ sEVs secreted by activated T cells could induce immunosuppression while reducing immunogenicity of tumour cells. Thus, I-sEV PD-1/CD80 plays an intensive role than IFN-γ in muting antitumour immunity, even simultaneous stimulation of tumour cells with sEV PD-1/CD80 and IFN-γ (both at the pathophysiological concentrations in cancer patients), may still lead to cold tumour phenotype. Moreover, PD-1/CD80$^+$ sEVs might selectively interact with PD-L1$^+$ tumour cells leading to more specific and efficient regulation of antitumour immunity. In contrast, IFN-γ, regardless of the expression level of PD-L1 in tumours, induced increasingly universal effects on immune mediation. This finding might also partially explain the fact that in some of patients, a high level of IFN-γ was not always accompanied by high expression of tumour cell PD-L1 and a positive response to immunotherapy. Transcriptionally, IFN-γ upregulates the mRNA expression of various molecules, including PD-L1, ICAM-1 and MHC-I, by activating the JAK/STAT signalling pathway and stimulated the unspecific secretion of sEVs[39]. This process explained

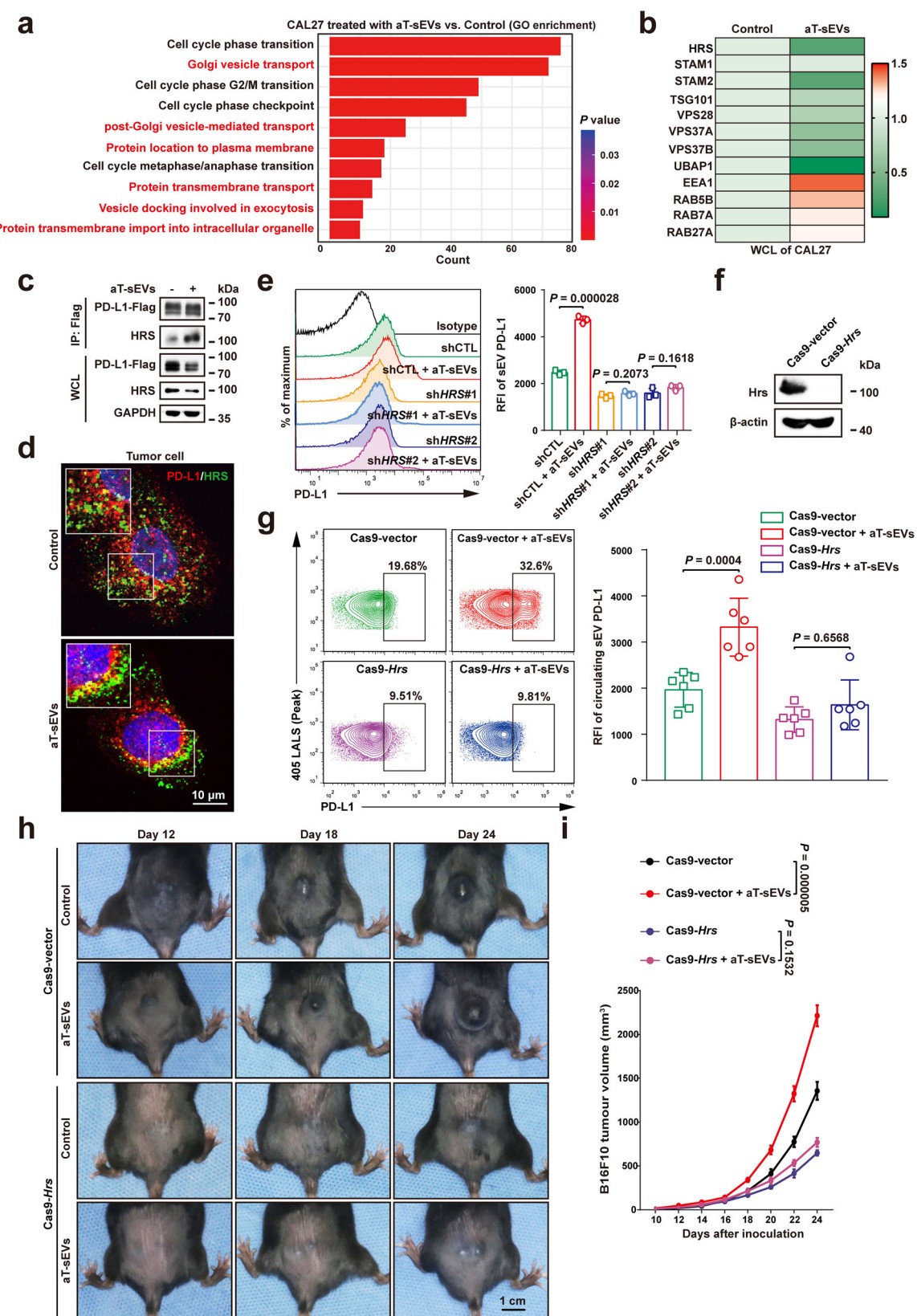

the reason why the protein level of PD-L1 was increased in both IFN-γ-treated tumour cells and secreted sEVs. In contrast to IFN-γ, sEV PD-1/CD80 mainly initiated transportation of vesicles, endocytosis and exocytosis, indicating its posttranslational regulation of membrane PD-L1 and subsequent secretion of PD-L1⁺ sEVs through intracellular trafficking, which was highly selective and active. Besides, the lipid

bilayer structure could ensure the stability and distant migration of sEV PD-1/CD80 in the circulation[22,42,43], while IFN-γ, is most likely eliminated within several minutes, limiting its systematic distribution and long-term effects[44,45]. These findings collectively suggest that, compared to IFN-γ, PD-1/CD80 carried on I-sEVs could give rise to a much stronger suppression of antitumour immunity in a distinct manner.

**Fig. 6 | The ESCRT machinery in tumour cells contributed to the adaptive immunosuppression initiated by PD−1/CD80 on aT-sEVs. a** GO analysis of key DEGs in CAL27 cells with aT-sEV treatment. DEGs, differentially expressed genes. **b** Heat map illustrating relative levels of HRS, STAM−1, STAM-2, TSG101, VPS28, VPS37A, VPS37B, UBAP1, EEA1, RAB5B, RAB7A and RAB27A in CAL27 cells with aT-sEV treatment. **c** Co-immunoprecipitation of PD-L1 and HRS in aT-sEV-treated CAL27 cells expressing exogenous Flag-tagged PD-L1 proteins. Immunoprecipitation was performed using an anti-Flag antibody. **d** Immunofluorescence staining of intracellular PD-L1 (red) and HRS (green) in CAL27 cells with or without aT-sEV treatment. Scale bar, 10 μm. **e** Flow cytometry profiles (left) and quantification of RFI (right) of secreted sEV PD-L1 in supernatants from CAL27 cells with indicated treatments. Vector plasmid was used as a control for knockdown of *HRS*. **f** Western blot analysis of the level of Hrs in B16F10 tumour cells with or without knockout of *Hrs*. Cas9-vector was used as a control for Cas9-*Hrs*. **g** Flow cytometry profiles (left) and quantification of RFI (right) of circulating sEV PD-L1 in C57BL/6 mice bearing B16F10-Cas9-vector and B16F10-Cas9-*Hrs* tumours with or without aT-sEV treatment (n = 6 mice per group). **h** Representative images of B16F10-Cas9-vector and B16F10-Cas9-*Hrs* tumours in C57BL/6 mice with indicated treatments at days 12, 18, 24. **i** The growth curve of B16F10-Cas9-vector and B16F10-Cas9-*Hrs* tumours in C57BL/6 mice after in indicated treatments (n = 6 mice per group). For (**i**) data were presented as mean ± s.d.; Two-way ANOVA. For (**e, g**) data were presented as mean ± s.d.; n = 3 biologically independent samples; Two-sided *t*-test. The relevant raw data and uncropped blots are provided as a Source Data file.

It can be speculated that sEVs carrying PD-1/CD80 might work as a functional unit with the assistance of other coexpressing molecules, thereby requiring a more sophisticated blockade than that provided by antibodies in circulation.

This is supported by our very recent finding that ICAM-1/LFA-1-mediated adhesion was essential for the interaction between tumour cell-derived PD-L1+ sEVs and T-cell PD-1[38]. Of interest, a previous study also revealed the interaction between PD-1+ and PD-L1+ sEVs and suggested that this interaction serves as a potential mechanism to reduce circulating immunosuppressive PD-L1+ sEV levels[29]. However, another recent report pointed out the binding of anti-PD-1 blocking antibody to sEV PD-1 would lead to a consumption of immunotherapy drugs[30]. It is reasonable to deduce that, the application of an anti-PD-1 blocking antibody could abrogate the binding of PD-1+ sEVs to PD-L1+ sEVs, indicating that I-sEVs excessively consume immune checkpoint blockade antibodies regardless of the existence of circulating PD-L1+ sEVs. Through competitive binding, the antibodies lead to the dissociation of PD-L1+ sEVs from PD-1+ sEVs, releasing it back into circulation. Regardless, the binding rate of blocking antibody to target proteins on sEVs was absolutely less than 100%, similar to a former report pointing out that PD-L1 presented on the sEVs was less responsive to the current antibodies and resulted in resistance of PD-L1+ sEVs[30]. More importantly, the application of conventional anti-PD-1 antibodies would not block sEV CD80. Therefore, I-sEV PD-1/CD80 could still interact with tumour PD-L1 and induce immunosuppression even in the presence of therapeutic antibodies. These important findings regarding the potential harmful effects of PD-1/CD80+ sEVs on antitumour immunity serves as a warning signal for the improbability of using biogenic or synthesised vesicles carrying PD-1/CD80 for cancer treatment.

In summary, we discovered that the ligation of PD-L1 on tumour cells by circulating PD-1/CD80+ I-sEVs resulted in an intense and adaptive immunologically 'cold' tumour phenotype (Supplementary Fig. 10). This process reprograms the landscape of adaptive immune resistance, which was thought to be mainly initiated by IFN-γ. A more comprehensive understanding of multiple checkpoints on circulating sEVs might be the key to overcoming immunosuppression and increasing the efficacy of antitumor treatment.

## Methods
### Cell culture
The EL4 mouse T cells were purchased from CTCC (China Centre for Type Culture Collection, CTCC). The CAL27 human oral cancer, MDA-MB-231 human breast cancer, H1264 human lung cancer, MC38 mouse colon cancer, A375 human melanoma and B16F10 mouse melanoma cells were purchased from ATCC. CAL27, MDA-MB-231, MC38 and A375 were cultured in DMEM (Sigma) supplemented with 10% fetal bovine serum (FBS) (Invitrogen). EL4 and H1264 were cultured in RPMI 1640 medium (Invitrogen) supplemented with 10% FBS. All cells were cultured at 37 °C in a 5% $CO_2$ incubator. For stimulation with IFN-γ, cells were incubated with 10 ng ml$^{-1}$ of recombinant human IFN-γ (Sino Biological Inc.) for 48 h.

### shRNA and CRISPR/Cas9 genome editing
Generation of stable *HRS* knockdown CAL27 cells. First, short hairpin RNAs (shRNA) against human *HRS* (also known as *HGS*) (NM_004712, GCACGTCTTTCCAGAATTCAA, GCATGAAGAGTAACCACAGC) or scrambled shRNA-control (Addgene) were packaged into lentiviral particles using 293 T cells co-transfected with the viral packaging plasmids. Second, we collected lentiviral supernatants at 48 h after transfection. Last, cells were infected with filtered lentivirus and then selected by 8 μg ml$^{-1}$ puromycin.

Generation of stable *Hrs or Pd-l1* knockout B16F10 cells. The procedure of CRISPR/Cas9 genome editing was done as previously study[46–48]. For *Pd-l1* gene disruption, mouse *Pd-l1* (guide 1: GTTTAC-TATCACGGCTCCAA, guide 2: GGGGAGAGCCTCGCTGCCAA). For *Hrs* gene disruption, mouse *Hrs* guide (TCCTGCTCCACAGAGGCAAGTGG) was transfected. Knockout clone was identified by western blot analysis.

### Isolation of human primary human T cells
Firstly, peripheral blood mononuclear cells (PBMC) were isolated using ficoll gradient and standard protocols[49,50]. After that, human T cells were purified from PBMCs using EasySep™ Direct Human T Cell Isolation Kit (STEMCELL Technologies Inc). Primary T cells (1 × 10$^6$ ml$^{-1}$ per well) were then seeded in 24 well plates in the presence of PHA (5 μg ml$^{-1}$) for 48 h. Culture supernatants were collected for following study.

### Patients and specimen collection
The study was conducted strictly based on the guidelines setting forth by the Medical Ethics Committee of Hospital of Stomatology Wuhan University. All patients, aged over 18 years, consented to participate in the study and provided signed informed consent forms. Blood samples from primary head and neck squamous cell carcinoma (HNSCC) patients used in this experiment were collected at the Department of Oral and Maxillofacial Surgery, School and Hospital of Stomatology Wuhan University. Clinical staging was performed in accordance with the guidelines of the International Union Against Cancer (UICC, 2018) on cancer tumour, node, metastasis system. Blood samples from healthy donors were collected at the Hospital of Stomatology Wuhan University after approval by the ethics committee. Written consent was obtained from each healthy donor before blood collection. All studies involving blood samples from healthy donors were performed according to relevant ethical regulations. Patients with recurrent or metastatic HNSCC treated with an anti-PD-1 antibody (anti-PD-1) and chemotherapy were enrolled in further study. These patients were treated with the same combination chemotherapy regimen of paclitaxel plus cisplatin. All patients had signed the informed consents and pretreatment peripheral blood was obtained in sodium heparin tubes. Clinical response was determined as best response based on immunerelated RECIST (irRECIST) using unidimensional measurements. The assessment of clinical responses for patients was performed independently in a double-blind fashion. All patients' information were shown in

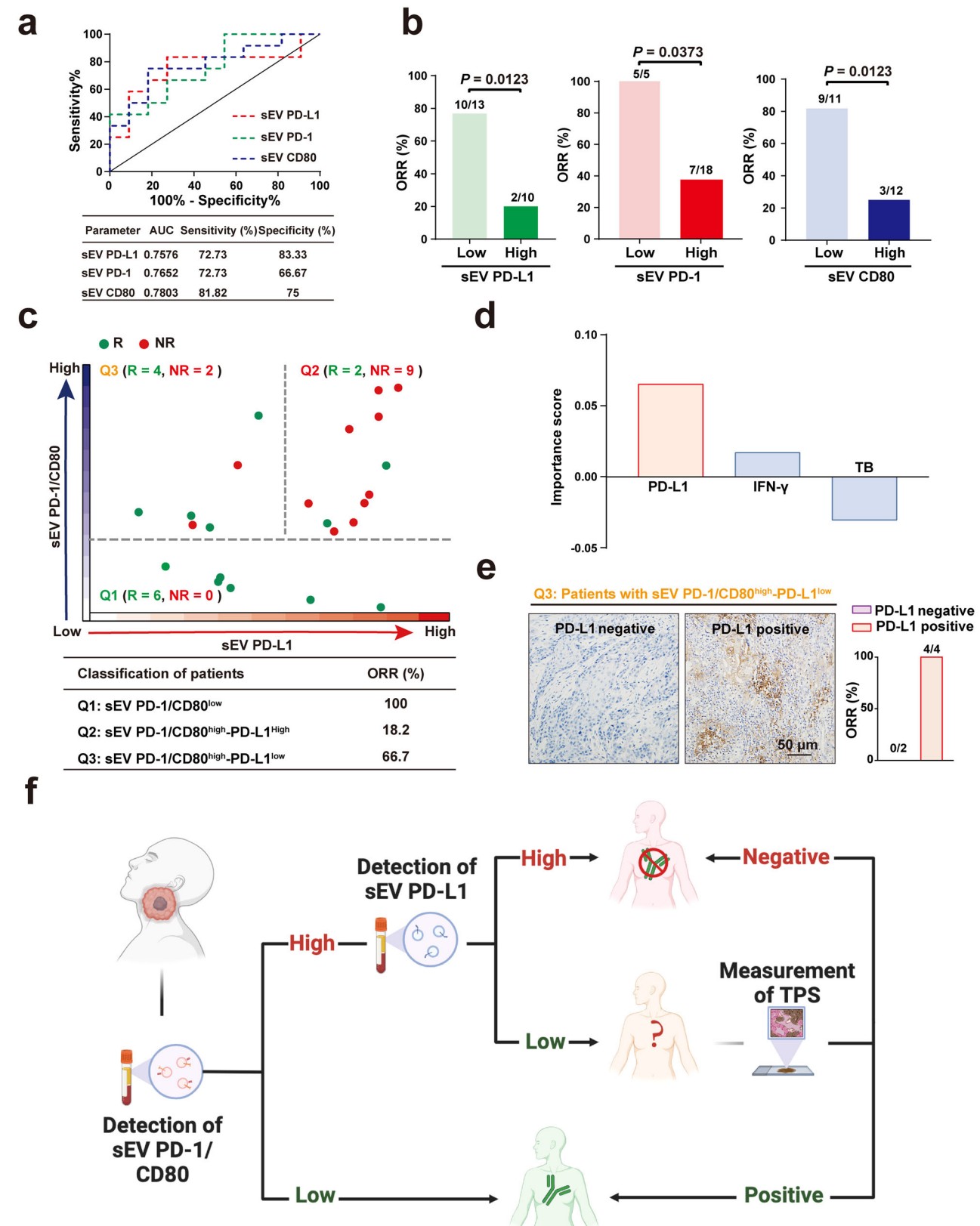

Supplementary Table 1. All participants received a compensation of transportation expenses.

## sEV separation

For sEV purification from cell culture supernatants, cells were cultured in media supplemented with 10% sEV-depleted FBS. Supernatants were collected from cell cultures and extracellular vesicles were purified by a standard differential centrifugation protocol[17,43,51–53]. In brief, culture supernatants were centrifuged at $3000\,g$ for 30 min to remove cell debris and dead cells. Then large extracellular vesicles were removed after centrifugation at $16,500\,g$ for 45 min (Beckman Optima XPN, USA). The supernatants were collected for subsequent purification.

**Fig. 7 | Synchronous analysis of multiple checkpoints on circulating sEVs is superior in predicting clinical responses to anti-PD-1 immunotherapy. a** ROC curve analysis of circulating sEV PD-L1, PD-1, and CD80 in clinical responders (n = 12) and non-responders (n = 11) (top). AUC area under curve. Bottom, detailed data associated with the ROC curve analysis. **b** ORR of immunotherapy in patients with high (n = 10) and low (n = 13) pretreatment levels of circulating sEV PD-L1 (left), high (n = 18) and low (n = 5) pretreatment levels of circulating sEV PD−1 (middle), high (n = 12) and low (n = 11) pre-treatment levels of circulating sEV CD80 (right). ORR, objective response rate. **c** Tracking the levels of circulating sEV PD-L1 and PD-1/CD80 stratified responders to anti-PD-1 therapy (green) from non-responders (red) (left). HNSCC patients were divided into three quadrants (Q1-Q3) based on the

levels of circulating sEV PD-L1 and sEV PD-1/CD80 (top). Detailed information of ORR in the three subtypes of patients (bottom). **d** Bar plots showing importance scores of different indexes based on random forest machine learning for distinguish responders to immunotherapy from nonresponders. TB, tumour burden. **e** Analysis of ORR in patients with PD-L1 negative tumour cells (n = 2) and patients with PD-L1 positive tumour cells (n = 6). All six patients were extracted from Q3 in (**c**), in which high levels of sEV PD-1/CD80 and low levels of sEV PD-L1 were detected in patients. Representative histochemistry images of PD-L1 staining in patient with PD-L1 positive or negative tumours (left). Right, ORR was plotted bar diagram. **f** Scheme of the strategy for predicting patients' immunotherapy response. For (**b**, **d**) Data were represented as mean ± s.d.; Two-sided Fisher's exact test.

For isolating sEVs for characterisation or in vitro experiments, iodixanol density gradient centrifugation was used. Briefly, sEVs harvested by differential centrifugation were loaded on top of a discontinuous iodixanol gradient (5%, 10%, 20% and 40%, made by diluting 60% Opti-Prep aqueous iodixanol with 0.25 M sucrose in 10 mM Tris) and centrifuged at 120,000 g for 18 h at 4 °C (Beckman Optima XPN, USA). Twelve fractions of equal volume were collected from the top of the gradients, with the sEVs distributed at the density range from 1.04 to 1.32 g ml⁻¹, as previously demonstrated[17,38,54,55]. For further testing, the small extracellular vesicles (sEV) were finally pelleted by ultracentrifugation at 120,000 g for 70 min at 4 °C. For collecting sEVs for animal experiments, sEVs were purified by a standard differential ultracentrifugation protocol[17,18,43]. In brief, the supernatants with cell debris and large extracellular vesicle removed were then centrifuged at 120,000 g for 70 min at 4 °C (Beckman Optima XPN, USA) to pellet the sEVs. Small EVs obtained through differential ultracentrifugation were routinely utilised for experimental treatments following a size quality check using a nanoparticle tracking system.

For isolating circulating sEVs from blood samples, we followed a previously established protocol[17]. Briefly, peripheral blood from mice, cancer patients or healthy donors was centrifuged at 1550 g for 30 min to obtain cell-free plasma. Then the obtained plasma was centrifuged at 16,500 g for 45 min. The collected supernatants were then centrifuged at 120,000 g for 70 min at 4 °C (Beckman Optima XPN, USA) to pellet circulating sEVs followed by resuspending them in PBS for subsequent analysis.

### Characterisation of purified sEVs

For characterising sEV, iodixanol density gradient centrifugation was used as mentioned above. The purified sEVs were verified using electron microscopy. Briefly, the purified sEVs suspended in PBS were dropped on formvar carbon-coated nickel grids. After staining with 2% uranyl acetate, grids were air-dried and visualised using a transmission electron microscope (Hitachi HT-7700, Japan). For immunogold labelling, purified sEVs suspended in PBS were placed on formvar carbon-coated nickel grids, blocked and incubated with monoclonal antibody that recognises the extracellular domain of PD-1 or CD80, followed by incubation with the secondary antibody conjugated with protein A-gold particles (5 nm). Each staining step was followed by five PBS washes and ten ddH₂O washes before contrast staining with 2% uranyl acetate.

The size and concentration of sEVs purified from cell culture supernatants and patients' plasma were evaluated using an individual particle tracking on a ZetaView (Merkel Technologies Ltd., Germany), which was equipped with fast video capture and particle-tracking software.

### ELISA

For detection of PD-L1, PD-1 and CD80 on sEVs from HDs' and patients' plasma, the procedure was conducted as described previously[17,56,57]. Briefly, ELISA plates (96 well) (Biolegend) were coated with 25 μg circulating sEVs per well (100 μl). The plates were then incubated with capture antibody against PD-L1 (Clone 5H1-A3, Millipore) and

biotinylated monoclonal PD-L1 antibody (Clone MIH1, eBioscience). sEV PD-1 and sEV CD80 were quantified using the human PD-1 and CD80 ELISA kit (for PD-1 catalogue number DY1086, for CD80 catalogue number DY140) from R&D Systems (Minneapolis, MN).

### Nanoparticle flow cytometry analysis

Purified sEVs of the blood and cell culture supernatants were tested by nanoparticle flow cytometry analysis. 110 nm orange FluoSpheres (standard beads with fluorescence provided by manufacturer) of known particle concentration (5000 particles/μl) were applied to calibrate the sample flow rate. Then, non-fluorescent standard beads with sizes of 180 nm, 240 nm, 300 nm, 590 nm, 880 nm and 1300 nm (provided by manufacturer) were employed for size gating, ensuring an appropriate working range for subsequent detection. The following reagents were used for nanoparticle flow cytometry analysis: antibody against CD45, CD144, EpCAM, CD4, CD8, CD11c, CD19, CD86, PD-1, CD80, PD-L1, TIM-3, LAG-3, CTLA-4, ICAM-1, CD47, EGFR, IgG1 or IgG2b was added. Each sample (0.5–1 μg, about 1–2 × 10⁷ particles) with 0.25 μg antibody were incubated at room temperature for 30 min and then washed twice with 1 ml PBS at 120,000 g for 70 min at 4 °C (Beckman Coulter MAX-XP centrifuge). The pellet was resuspended in 300 μL PBS for analysis under identical detection condition for standard beads (A50 micro plus Flow Cytometry, Apogee). All samples were measured for 2 min at a flow rate of 1.5 μL/min using SSC triggering (405-nm laser, 70 mW). The detection threshold was set at 20 a.u. (small angle light scatter [SALS]) and 25 a.u. (large angle light scatter [LALS]) to eliminate optical and electronic background noise without losing particles of interest. Positive events were defined as those exhibiting a fluorescent signal within the designated gate. Concentrations were determined by accounting for flow rate, measurement time and sample dilution to correct the number of detected.

### Mass spectrometry-based proteomics data analysis

The samples were resolved and processed as previously described[58,59]. Briefly, Dried peptides was redissolved in 10 μL of 0.1% formic acid (FA), injecting 2 μl of each into a nanoElute for the proteomic analysis. All peptides on a 25 cm in-house packed column (360 μm OD × 75 μm inner diameter (ID)) were separated by a 120 min gradient elution at a flow rate of 300 nl/min. The mobile phase buffer consisted of buffer A (0.1% FA in ultrapure water) and buffer B (0.1% FA in acetonitrile). The eluate was online electrosprayed and analysed using a timsTOF Pro mass spectrometer (Bruker). Next, these data were processed and quantitated. The raw files were searched directly against the uniprot database version downloaded November 2019 with no redundant entries, using PEAKS Studio X+ software (Bioinformatics Solutions Inc.).

### Quantitative PCR (qPCR)

Total RNA was isolated from CAL27 cells using TRIzol Reagent (Invitrogen), and reverse transcribed into first-strand complementary DNA (cDNA) with HiScript II Q RT SuperMix for qPCR (+gDNA wiper) (Vazyme). One-fifth of cDNA was then used for PCR by ChamQ SYBR qPCR Master Mix (Vazyme) in a CFX96 Real-Time PCR Detection

System (Bio-Rad Laboratories). The primer sequences used for RT−qPCR was as follows: *PD-L1*: 5′- GGCATTTGCTGAACGCAT-3′ and 5′- CAATTAGTGCAGCCAGGT-3′, *ICAM-1*: 5′- ATGCCCAGACATCT GTGTCC-3′ and 5′- GGGGTCTCTATGCCCAACAA-3′, *HLA*-A: 5′-AGGGTT TCTTGCTGAGGTACA-3′ and 5′- GGTCTCTCTGTCCCATGTCTTA-3′, *HLA*-B: 5′- CAGTTCGTGAGGTTCGACAG-3′ and 5′- CAGCCGTACATG CTCTGGA-3′, *HLA*-C: 5′- CCATGAGGTATTTGTGGACCG-3′ and 5′- TCTC GGACTCTCGTCGTCG-3′, *GAPDH*: 5′- GGAGCGAGATCCCTCCAAAAT-3′ and 5′- GGCTGTTGTCATACTTCTCATGG-3′. *GAPDH* was selected as the internal control for each experiment.

## Förster resonance energy transfer (FRET) assays

FRET assays for determining molecules interaction were performed as previously described[60,61]. Briefly, CAL27 cells were incubated with aT-sEVs for 6 h. And then CAL27 cells were incubated with anti-PD-L1 antibody detected by tyramide594-labelled biotin (energy acceptor), while aT-sEVs were incubated with anti-PD-1 or anti-CD80 antibodies labelled with AF488 (energy donor). If energy donor interacts with energy acceptor, excitation at 488 nm would result in subsequent energy transfer to tyramide594, resulting in the detection of tyramide594 fluorescence. Upon photobleaching of tyramide594, the fluorescence would be extinguished, preventing AF488 from further energy transfer and leading to an increased fluorescence signal from AF488. The difference in fluorescence intensity of the energy donor before and after photobleaching was evaluated as the level of FRET efficiency, indicating the close proximity and physical interaction between sEV PD-1/CD80 and tumour surface PD-L1. The fluorescence intensity and FRET efficiency of sEV PD-1 or CD80 were calculated by Image J.

## Treatment of tumour cells with the aT-sEVs

To block PD-1 or (and) CD80 on the sEVs surface, purified aT-sEVs were incubated with PD-1 or (and) CD80 blocking antibodies (10 µg ml$^{-1}$) or IgG isotype antibodies (10 µg ml$^{-1}$) in 100 µl PBS for 6 h, then washed with 1 ml PBS and pelleted by ultracentrifugation to remove the non-bound free antibodies. Then, tumour cells were cocultured with 25 µg ml$^{-1}$ of sEVs individually for 24 h. Finally, the cells were for further study. For evaluating the role of aT-sEVs in stimulating sEV secretion of tumour cells, tumour cells were treated with aT-sEVs for 6 h. And then we performed PBS washes to remove any residual aT-sEVs before an additional 24 h of cell culture in sEV-free medium. After that, the levels of sEV PD-L1 in the supernatant were measured. For some experiments, sEVs derived from non-activated T cells (cT-sEVs) were used as control.

## Flow cytometry analysis

To assess the expression levels of specific proteins on various cells, cells were divided into several tubes and stained, in parallel, with different antibodies. The stained cells were analysed on a flow cytometer FACS Verse using CytoFLEX (Beckman Coulter, Life Sciences). The detailed gating strategies of tumour cells including CAL27, A375, MDA-MB-231 and H1264 cells were shown in Supplementary Fig. 11.

## Immunohistochemical staining

Immunohistochemical staining were done as described previously[62,63]. Briefly, streptavidin-biotin complex (SABC) immunohistochemical kit (MXB Technology Ltd, China) was used. Serial sections of the HNSCC were immunohistochemically stained for PD-L1, ICAM-1, MHC-I and TAP1. In addition, serial sections of the MC38 tumours were then immunohistochemically stained for CD8 and PD-L1. For evaluation of tissue samples, Aperio ScanScope CS scanner was utilised for scanning slices.

## Assessment of PD-L1 staining

The stained slides were simultaneously assessed by a dedicated head and neck pathologist, certified for PD-L1 testing and a head and neck researcher; discrepancies were resolved by consensus. Staining was

assessed for tumour proportion score (TPS). The TPS was defined as the number of positive tumour cells divided by the total number of viable tumour cells multiplied by 100%. Clinically relevant cut-offs of ≥1 for TPS was used.

## Immunofluorescence staining

Immunofluorescence staining was performed on formaldehyde-fixed cells, formalin-fixed, paraffin-embedded (FFPE) sections or frozen sections. For cells plated on slides frozen sections, permeabilization with 0.3% Triton X-100 was performed before blocking with bovine serum albumin (BSA) for 1 h at room temperature. For FFPE sections, antigen retrieval by steaming in citrate buffer (pH = 6.0) was performed before blocking. The fixed cells or FFPE sections were incubated with primary antibodies overnight at 4 °C, followed by incubation with fluorophore-conjugated secondary antibodies for 1 h at 37 °C. Nuclei were counter-stained with DAPI. Samples were observed using a confocal microscope at 100× magnification (Ultra-VIEW VoX, PerkinElmer).

## Western blot analysis

Whole cell lysates or sEV proteins were separated using 10% SDS−PAGE and transferred onto PVDF membranes. The blots were blocked with 5% non-fat dry milk at room temperature for 1 h, and incubated overnight at 4 °C with the corresponding primary antibodies at dilutions recommended by the suppliers and followed by incubation with HRP-conjugated secondary antibodies (Cell Signalling Technology) at room temperature for 1 h. CD9, CD81, HRS, ALIX and TSG101 were used as sEV markers. GAPDH or β-actin was used as a loading control.

## In vivo mice experiment

All animal handling and procedures were approved by the Ethics Committee for Animal Research, Wuhan University, China. The methods were carried out in accordance with guidelines and regulations established for the care and use of laboratory animals. 8-week-old female C57BL/6 mice were housed in specific pathogen-free (SPF) animal facilities for all animal experiments. Experimental and control animals were housed in separate cages within the same room. MC38 or B16F10 tumours were established by subcutaneously injecting $5 \times 10^5$ MC38 or $1.5 \times 10^5$ B16F10 cells in 150 µl medium into flanks of mice. Tumours were measured using a digital calliper and the tumour volume was calculated using the following formula: (length x width$^2$)/2. Mice were euthanized if the longest dimension of the tumours reached 2.0 cm. Immediately following euthanasia (Inhalant anaesthetic overdose followed by decapitation), blood samples were harvested by cardiac puncture and sEVs were purified and detected by Nano-FCM analysis using the afore mentioned method. For establishing syngeneic mouse melanoma model in C57BL/6 mice, B16F10-Cas9-vector, B16F10-Cas-*Pd-l1* and B16F10-Cas9-*Hrs* cells ($1.5 \times 10^5$ cells in 150 µl medium) were subcutaneously injected into immunocompetent C57BL/6 mice. Tail vein injections of sEVs (100 µg in 150 µl PBS) were performed every 2 days. Blocking antibodies were given on days 13, 15, 17 and 19. Antibodies used for in vivo PD-1 blockade experiments were given intraperitoneally at a dose of 200 µg per mouse and include: PD-1 (RMP1-14) and rat IgG 2b isotype (LTF-2) (BioXCell). Differences in survival were determined for each group by the Kaplan−Meier method and the log-rank test. Survival analysis was performed with 'survival' package in GraphPad Prism version 9.3.1. Downstream analyses of mouse samples (immunohistochemical, immunofluorescence staining, flow cytometry and Nano-FCM analysis) were performed in a blinded fashion. For flow cytometry, the tumour samples were harvested and single cell suspensions were prepared, and red blood cells were lysed using ACK Lysis Buffer (Beyotime Biotechnology). B16F10 cell gating strategy in flow cytometry analysis was shown in Supplementary Fig. 12.

## Computational analyses

Computational analysis and predictions were performed using the R language and environment for statistical computing (version 3.01) and Bioconductor (version 2.221). For Gene Set Enrichment Analysis (GSEA), we used the entire proteomic expression dataset. Gene sets from Molecular signatures database (MSigDB, https://www.broadinstitute.org/gsea/msigdb/index.jsp) v5.1 were used for GSEA (H: 50 hallmark gene sets; CS: KEGG: 186 canonical pathways from Kyoto Encyclopedia of Genes and Genomes [KEGG] pathway database; C5: 825 gene sets based on Gene Ontology [GO] term). The default parameters were used to identify significantly enriched gene sets.

Random Forest is a machine learning method combined the output of an ensemble of regression trees to predict the value of a response variable[64,65]. Using this method reduces the risk of overfitting and makes the method robust to outliers and noise in the input data. Heatmaps based on random forest algorithm were generated to find highest predictive values.

## Statistics and reproducibility

Statistical analysis and number of biologically independent sample (n) were indicated in the figure legends. Data was processed with GraphPad Prism, Version 9.3.1 (GraphPad Software). Differences between groups were calculated using Student $t$ test. Log-rank test was used for survival analysis. Correlations were determined by Pearson's $r$ coefficient. Two-way ANOVA was performed to compare mouse tumour volume data among different groups. Error bars shown in graphical data represent mean ± s.d. $P$ value of 0.05 and below was regarded statistically significant.

## Reporting summary

Further information on research design is available in the Nature Portfolio Reporting Summary linked to this article.

## Data availability

All data are available in the main text, Supplementary Information, or source data file. The mass spectrometry proteomics data are available via ProteomeXchange with identifier PXD050744. Source data are provided with this paper.

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

## Acknowledgements

This work was supported by the National Natural Science Foundation of China (82341023, 81922038), The Innovative Research Team of High-level Local Universities in Shanghai (SHSMU-ZLCX20212300); Medical Science Advancement Program (Basic Medical Sciences) of Wuhan University (TFJC2018005), the Fundamental Research Funds for the Central Universities (2042022dx0003, 2042023kfyq02) and The Interdisciplinary Research Project of School of Stomatology Wuhan University. All schematic figures were created with BioRender.com.

## Author contributions

G.C., L.Z.Z. and J.G.Y. conceived the project and designed the experiments. L.Z.Z., J.G.Y., G.L.C., Q.H.X., Q.Y.F., H.F.X., Y.L. and J.H. purified and characterised small extracellular vesicles. L.Z.Z., J.G.Y., J.H., M.W., and H.M.L. performed the mice experiments. L.Z.Z. and J.G.Y. performed the immunoprecipitation and western blot analysis. L.Z.Z., J.G.Y., Q.H.X., Q.Y.F. and H.F.X. performed the Nano-FCM analysis and flow cytometry experiments. L.Z.Z., J.G.Y., Q.H.X., Q.Y.F. and H.F.X. performed the TEM, NTA, ELISA, qPCR, FRET assay and immunofluorescence staining experiments. Z.L.Y. and W.Z. performed pathological analyses. G.L.C., F.B.W., K.Z.Y., H.G.J., F.X.Z., W.W., H.Y.Y. and Y.H.Z. provided human samples and associated clinical data. L.Z.Z., J.G.Y., G.L.C., Q.H.X., Q.Y.F., H.F.X., Y.L., J.H., H.M.L. and Y.C.L. analysed and interpreted the data. G.C., L.Z.Z. and J.G.Y. wrote the paper. Z.B., H.Y.Y. and B.L. edited the paper. All authors have read and approved the final manuscript.

## Competing interests

The authors declare no competing interests.
