## [Peer Review File · Nature Communications]

PD-1/CD80⁺ small extracellular vesicles from immunocytes induce cold tumours featured with enhanced adaptive immunosuppressionREVIEWER COMMENTS

Reviewer #1 (Remarks to the Author):

In this extensive study of circulating and tumor-derived EVs in patients with HNSCC, the authors convey a message that activated T cell-derived sEV carrying PD-1/CD80 reprogram the tumor and the TME, converting it from "warm" (T-cell infiltrated) to "cold" (T-cell depleted) milieu. Further, the authors propose that PD-L1(+) sEV secreted by reprogrammed tumors are instrumental in this conversion, contributing to resistance of the tumor to IT. While this hypothesis, linking PD-L1(+) sEV to the emergence of "cold" TME and resistance of tumors to IT is intriguing, it is also provocative and thus requires convincing and demanding experimental evidence for multiple steps and mechanisms operating in the TME to bring about the conversion. The authors present a lengthy and complex set of what amounts to as two manuscripts merged into one, since 18 Supplemental Figures and Legends constitute a necessary component of the presented scenario. The result is extremely difficult to peruse, and this creates a major difficulty with appreciating and understanding the presented data. The text of this manuscript is verbose and very lengthy (i.e., the Introduction takes 4 pages, and the Discussion is 9 pages long). Combined with somewhat distorted use of English, the text is often difficult to follow and comprehend.

If the scenario of cellular-vesicular interactions the authors present is real, the significance of the manuscript would be very high. However, there are several methodology issues that are questionable, undermining the validity of some of the results. Specifically, isolation of tumor derived and circulating sEV is not adequately detailed and involves ultracentrifugation in all cases, which does not differentiate sEV from larger vesicles and yields non-purified sEV. It is not clear whether iodixanol density gradient centrifugation was used only for sEV characterization or routinely followed ultracentrifugation steps. Flow cytometry analysis of sEV on beads is not adequately described in Methods. In Sfigure 1, on bead flow data are presented as percentages but, as the numbers of vesicles sitting on beads is unknown, the appropriate designation is "relative flow intensity (RFI)" compared to isotype control and not percent positive. Thus, the nanoflow analyses in this work are not correctly interpreted and quantitation of proteins carried by circulating sEV may be suspect.

Innovation of the data is partly compromised by previously reported information about the participation of the checkpoint proteins, especially PD-L1 or PD-1/CD80 in cellular interactions, including the senior authors' previously published work. Release by immune cells of sEV bearing PD-1/CD80 in patients with HNSCC and other tumors and correlations with anti-tumor immunity and disease activity have been previously reported and are not referenced here (especially some the data presented as novel in Figures 1 and 2 are largely confirmatory). Also, the fact that injections of tumor-derived sEV (TEX) into tumor-bearing mice reduced the numbers of CD8+ T cells infiltrating the tumor has been previously reported in murine oral carcinogenesis model (Razzo, Ludwig et al, Carcinogenesis). On the other hand, the data showing that PD-1/CD80(+) aT-sEV promoted secretion of PD-L1 by tumor cells is novel and should be the major focus of this manuscript. FRET (energy transfer) assays in Figure 3 illustrating the co-localization of PD-1/CD80 and potential cis/trans interactions are interesting, but the presentation of these data is confusing and needs clarity. The mechanism responsible for reduction in PD-L1 on the tumor surface due to its internalization following ligation by CDPD-1/CD80 (+) is an expected effect of receptor/ligand interactions and was previously reported for PD-1/PD-L by G. Freeman among others. The data related to reprogramming of tumor cells by aT-sEV are not novel and the secretion by reprogrammed tumors of immunosuppressive sEV has been previously reported. The PD-1/PD-L1 pathway is not the only molecular pathway involved in these interactions, and its contribution to immunosuppression in the TME of HNSCCs may be trumped by TGF- β and adenosine-mediated pathways

Overall, this manuscript presents an overwhelming volume of data some of which are confirmatory and others new that are split between the main text and Supplemental Materials. The most important and potentially clinically relevant data on the effects PD-1/CD80 (+) sEV produced by activate T cells exert on the tumor in patients with HNSCC appear in Supplemental Materials. The Figures lack informative legends, and the reader is forced to navigate between the main text and Supplemental Figures. This makes for difficult reading and does not enhance our understanding of the interactions between immune sEVs and the tumor.

Reviewer #2 (Remarks to the Author):

Reviewer: Samantha Morrissey, M.D., Ph.D.
Departments of Internal Medicine and Pediatrics, UNC Health

Reviewer comments for Nature Communications Article

Overall, this is a very well designed and supported research paper. Understanding the role that exosomes from immune cells play in cancer progression is an exciting new frontier in the field. Previously, the focus in the field has been the effect of tumor derived exosomes on immune cells. Here, the authors investigate just the opposite. This paper addresses a specific mechanism by which a particular cohort of immune cell derived exosomes influence tumor immunogenicity. The authors did a very thorough job detailing the phenotype of these immune cell exosomes and exploring their effect on tumor cells. It is quite a clever adaptive mechanism they identified in that exosomes from activated T cells cause tumor cells to downregulate PD-L1 and become immunologically cold.

Understanding the mechanisms stratifying patients into responders versus non-responders has become the holy grail, if you will, in immuno-oncology. Teasing apart the biomarkers in a patient's blood to determine if a particular therapy will be effective is the pinnacle of clinical importance. This paper aims to suggest one new strategy for the prediction of immunotherapy response. Therefore, the novelty of this paper is twofold. The authors first identify a new mechanism by which tumor cells avoid immunotherapy as mentioned above. Second, they suggest sEV phenotyping using PD-1/CD80 and PD-L1 expression can help predict responsiveness to immunotherapy.

Major Revisions

- 1). Are the sEVs that express PD-L1 the same sEV that express PD-1 and CD80? PD-L1 expression seemed to be the most sensitive and specific marker to delineate HNSCC vs HD as demonstrated in supplemental figure 1b.
- 2). Is the conclusion that PD-L1+ sEV are the predominate type overall in the blood of HNSCC patients whereas the PD-1/CD80+ immune cell sEVs are the second most common and a novel population in cancer patients that contribute to tumor pathogenesis?

Minor Revisions

- 1). Patients in this study are currently on anti-PD-1 therapy and chemotherapy. Are they on the same chemotherapy regimen? Do they all have a similar starting point in terms of exosome composition after washout from previous treatments? Was there a difference at baseline in sEV composition in primary versus recurrent HNSCC?
- 2). Supplemental Figure 1E- do you have the flow plot for this Venn diagram?
- 3). Figure 1E and Supplemental Figure 2F- is it possible to plot tumor burden and survival against co-expression of PD-1/CD80 instead of just one surface marker alone?
- 4). I understand that the majority of PD-1/CD80 sEV are CD45+ but could you provide the flow plots and gating strategy for supplemental Figure 4A? I think it is an important concept to demonstrate within the overall sEV pool what percentage are CD45+ and within that particular pool the prevalence of PD-1/CD80+ sEV? Again, this goes back to the above question regarding PD-L1 vs PD-1/CD80. Are CD45+ sEV PD-L1 positive? What is the relative abundance of sEV PD-L1+ versus CD45+/CDPD-1/CD80+ in HNSCC?
- 5). Figure 1F- Could you include the CD3 staining in the heatmap?
- 6). Supplemental Figure 4E- based on the method section, it looks like the macrophages were

polarized to an M2 phenotype. Literature suggests that both M1 and M2 macrophages play a role in tumor pathogenesis. What kind of sEVs do M1 macrophages secrete?

7). Supplemental Figure 4 G/H- Could you provide CD3 and/or CD8 staining for both the WB and TEM figuring to confirm these are indeed T cell derived sEV?

8). Do patients with HNSCC have a higher number of immune cells compared to HD and that is why you see increased sEV with PD-1/CD80 markers?

9). Figure 1K- the legend graft colors are off and don't line up with the words

10). Supplementary Figure 9C- why do you think the T cell derived sEV localize to the tumor? Are there specific integrins driving this phenotype?

11). Supplementary Figure 9E- Will the tumor cells uptake any kind of EV vesicle or is this specific to the T cell derived sEV? If you pretreat the sEV with anti-PD-1 antibody, do you lose this association on the tumor cell surface?

12). Line 425- trans-ligation is misspelled.

13). Supplemental Figure 10D- those errors bars are wide, more replicates would be more convincing. It is an important point to note if PD-L1 mRNA is transferred from T cell derived EV or not.

14). Figure 3D- very nice experiment.

15). Figure 3D- I assume EEA and Rab7 are endosome associated markers, I would specify this in the manuscript.

16). Figure 4B-E- How are you measuring the secreted PD-L1? Are these sEV collected from the supernatants of the tumor cells before and after stimulation? Could the T cell derived sEV have PD-L1 on them which would be increasing overall PD-L1 expression instead of just secretion from the tumor cells?

17). Figure 4F- can you explain the results of the WCL WB? I am not sure I understand why TSG101 levels would go down when you add T cell derived sEV to the supernatant?

18). Supplemental figure 11B-Very nice western blots.

19). Supplementary figure 12 E&F- what are the control tEV? Are they from non-activated T cells?

20). Supplementary Figure 12G- do you have decreased phagocytosis of tumor cells with the increase in CD47 expression after sEV treatment?

21). Supplemental Figure 16E- PDL-1- patients have 0% ORR score whereas PD-L1+ patients have 100% response? Not sure I understand these results.

22). In line 873, you claim that less than 30% of sEVs were antibody-bound in HNSCC patients treated with blocking antibody. If the sEVs do not readily bind antibody, how can you explain the effect seen in the neutralizing experiments performed throughout the paper where anti-PD-1/anti-CD80 antibody was added to the sEV prior to culture or in vivo injection?

23). How much sEV were added for each of these in vitro experiments? Was it the same 100ug as the in vivo studies?

Reviewer #3 (Remarks to the Author):

In the manuscript titled Extracellular vesicle PD-1/CD80 from immunocytes induce cold tumors featured with enhanced adaptive immunosuppression, the author unveiled the mechanisms of sEV PD-1/CD80 via an adaptive loop to suppress the anti-tumor immunity. Also, the author proposed a novel strategy to predict the immunotherapy response by synchronously analysing the multiple checkpoints on circulating sEVs. This study provided some interesting findings and are valuable for further evolution of the immunotherapeutic approaches. However, considering the following problems, MINOR revision has to be done before the manuscript could be accepted for publication. Minor comments:

1. In figure 1a, one of the EVs you showed is more fusiform than spherical. Please explain the reason or provide a more typical photo.
2. The author needs to make an accurate definition of EV PD-1/CD80. Is EV PD-1/CD80 a mixture of EV PD-1 and EV CD80, or an EV containing both PD-1 and CD80, or a mixture of the three? This should also be reflected in your Graphical Abstract and figure 5h.
3. In the introduction, you should summarize the previous researches (reference 28-30) and point out your own innovation to highlight the advantages of your study.
4. Some sentences are too wordy and obscure (e.g. line 545-550). The current manuscript can be polished by a native English speaker or a professional language editing service.
5. The discussion section contains too many repetitions of the results. Sentences need to be more concise. In addition, there was no mention of the limitation of the study. This needs to be presented in the discussion.

Point-by-point response

Reviewer #1

Comments to the Authors

In this extensive study of circulating and tumour-derived EVs in patients with HNSCC, the authors convey a message that activated T cell-derived sEV carrying PD-1/CD80 reprogram the tumour and the TME, converting it from “warm” (T-cell infiltrated) to “cold” (T- cell depleted) milieu. Further, the authors propose that PD-L1(+) sEV secreted by reprogrammed tumours are instrumental in this conversion, contributing to resistance of the tumour to IT. While this hypothesis, linking PD-L1(+) sEV to the emergence of “cold” TME and resistance of tumours to IT is intriguing, it is also provocative and thus requires convincing and demanding experimental evidence for multiple steps and mechanisms operating in the TME to bring about the conversion. The authors present a lengthy and complex set of what amounts to as two manuscripts merged into one, since 18 Supplemental Figures and Legends constitute a necessary component of the presented scenario. The result is extremely difficult to peruse, and this creates a major difficulty with appreciating and understanding the presented data. The text of this manuscript is verbose and very lengthy (i.e., the Introduction takes 4 pages, and the Discussion is 9 pages long). Combined with somewhat distorted use of English, the text is often difficult to follow and comprehend.

Response: We appreciate the reviewer's thorough evaluation of our study on circulating sEV PD-1/CD80 from activated immunocyte and subsequent induction of sEV PD-L1 section in tumour cells. However, we would like to clarify that our manuscript primarily focuses on elucidating the function of activated immunocyte-derived PD-1/CD80⁺ sEVs in immunotherapy and the mechanism by which these sEV PD-1/CD80 contribute to the formation of an immune-desert tumour microenvironment. It appears that there may have been a misunderstanding regarding the role of PD-L1⁺ sEVs in creating "cold tumours," as pointed out by the reviewer. In fact, our results actually

demonstrate that sEV PD-1/CD80, mainly secreted by activated T cells, induce the downregulation of membrane MHC-I, ICAM-1, and PD-L1 in tumour cells, along with the stimulated secretion of sEV PD-L1 into the circulation. These combined factors ultimately contribute to the development of a "cold tumour" phenotype. We apologize for any confusion that we might bring in our manuscript.

Even though we totally agree with the reviewer's suggestion on the importance of providing convincing experimental evidence for our proposed hypothesis. Considering the methodology issues raised by the reviewer, we apologize for the lack of detailed descriptions, particularly regarding the methods of ultracentrifugation and nanoparticle flow cytometry, in the original manuscript. In the revised version, we have included comprehensive details and addressed these concerns in our point-by-point response.

We thank the reviewer's suggestion on the simplicity and the length of our manuscript. Indeed, to ensure the logical integrity of the paper, we have had included a substantial amount of data, including some relevant findings from previously reported research in the starting and the transitional part of our plot. We have now restructured the content to enhance clarity and readability by reducing redundancies and addressing the complexity of the data presentation. Briefly, to emphasize the novelty, we have eliminated redundant sections and paragraphs, and combine certain sections to make the manuscript more concise. The **Introduction** section is now condensed to **under 3 pages**, while the **Discussion** section takes **less than 5 pages**, encompassing a total of 17 figures (7 in the Main Text and 10 in the Supplementary Information). Meanwhile, we have diligently edited the text with the assistance of Springer Nature Author Services to enhance the language quality (SNAS verification code, 94C1-BD35-FF23-2554-DD1P). All the changes have been highlighted in red in the revised manuscript. We believe that the modification will enhance the reader's understanding and appreciation of the data presented.

Methodology issue

1. If the scenario of cellular-vesicular interactions the authors present is real, the significance of the manuscript would be very high. However, there are several methodology issues that are questionable, undermining the validity of some of the results. Specifically, isolation of tumour derived and circulating sEV is not adequately detailed and involves ultracentrifugation in all cases, which does not differentiate sEV from larger vesicles and yields non-purified sEV. It is not clear whether iodixanol density gradient centrifugation was used only for sEV characterization or routinely followed ultracentrifugation steps.

Response: We appreciate the reviewer's recognition of the significance of our manuscript. We apologize for the insufficient details provided regarding the isolation and subsequent utilization of sEVs in our manuscript. In response to this concern, we have thoroughly revised the manuscript to provide a comprehensive description of the methods employed for sEV isolation by sequential ultracentrifugation (**Page 26, Line 659**). In the present study, we employed iodixanol density gradient centrifugation for characterizing sEVs and isolating them for subsequent cell experiments. For sEVs used in animal experiments, we opted for sequential ultracentrifugation since large quantities of sEVs are required. For isolating circulating sEVs, we followed a standard differential centrifugation protocol⁷. We acknowledge the reviewer's concern regarding the purity of sEVs through ultracentrifugation. However, as stated in Minimal Information for Studies of Extracellular Vesicles 2018 (MISEV2018) guidelines, "*absolute purification, or complete isolation of EVs from other entities, is an unrealistic goal (as for many biological products).*" It is worth noting that sequential ultracentrifugation is still a widely accepted and most commonly used procedure among various techniques for sEV separation such as density gradients, precipitation, filtration, size exclusion chromatography, and immunoisolation¹. The intermediate step of sequential ultracentrifugation at 16,500 g is typically performed to remove large vesicles and protein aggregates, ensuring a more purified sEV population. More recent studies by

David Lyden et al. also supports the use of sequential ultracentrifugation for isolating small exosomes (Exo-S, 50-70 nm) and large exosomes (Exo-L, < 200 nm), collectively referred to as extracellular vesicles and particles (EVPs)^{2,3}. Besides, we notice that, for some researchers, size-exclusion chromatography (SEC) is an alternative method for sEV isolation, known for its ability to separate particles based on their size. According to the MISEV2018 guidelines, both SEC and differential ultracentrifugation are classified as methods with intermediate recovery and intermediate specificity for sEV isolation⁴. This indicates that these two methods are almost comparable for subsequent experiments. Moreover, before proceeding with any further treatments, we assessed the size distribution of the purified sEVs obtained through ultracentrifugation. As shown in **Figure R1a**, more than 90% of the purified sEVs were found to be less than 200 nm in size, excluding possibility of significant contamination from large vesicles. Another point of consideration is, combinations of methods will be used and may outperform single-method approaches. In our study, we have incorporated high-resolution nanoparticle flow cytometry to ensure the detection of vesicles with sizes less than 180 nm, employing a cytometry gate strategy based on standard bead samples (detailed descriptions have been provided in our response to the subsequent point and are included in the revised manuscript). We acknowledge that density gradient centrifugation contributes to much higher purity of separated sEVs when combine with other separation techniques. But it is worth noting that density gradient centrifugation is not simply a size-based separation method for sEVs, which means that larger-sized sEVs cannot be completely excluded either. Besides, it significantly compromises the yield rate, reducing it to approximately 10% (**Figure R1b**), which is insufficient to meet the demands of our subsequent animal experiments, considering the large quantity of sEVs required. This is in accordance with the suggestion raised in MISEV2018 that density gradient centrifugation offers high specificity but may have limitations in terms of recovery efficiency.

Figure R1 Characterization and yield rate of sEVs separated by ultracentrifugation and density gradient centrifugation. a, Size distribution of sEVs separated by ultracentrifugation (UC) and iodixanol density gradient centrifugation (DGC) using nanoparticle tracking analysis (NTA). b, yield rate of sEVs separated by UC and iodixanol DGC.

2. Flow cytometry analysis of sEV on beads is not adequately described in Methods. In Sfigure 1, on bead flow data are presented as percentages but, as the numbers of vesicles sitting on beads is unknown, the appropriate designation is “relative flow intensity (RFI)” compared to isotype control and not percent positive. Thus, the nanoflow analyses in this work are not correctly interpreted and quantitation of proteins carried by circulating sEV may be suspect.

Response: We apologize for any confusion caused by the inadequate description of nanoparticle flow cytometry in Methods. However, it seems that the reviewer misunderstands the technology we have used in the manuscript. Traditional methods for detecting specific protein-positive sEVs, such as the use of Dynabeads™ magnetic beads, often take advantage of beads coated with specific antibodies for capture. However, in our study, we did not employ this traditional method. In contrast, by utilizing nanoparticle flow cytometry, we are able to assess the membrane proteins carried on sEVs at the single-particle level without the need for bead-binding, thus preserving the physical characteristics of the sEVs. It is important to note that the beads we used in our study served as standard samples for size and fluorescence, rather than being

employed for sEV capture. Thus, there were no "vesicles sitting on beads" in our experimental setup. The presented data as a percentage reflects the proportion of sEVs that are positive for specific membrane proteins relative to the total number of sEVs analysed. In principle, both RFI and percentage can appropriately reflect the levels of PD-1 and CD80 carried on sEVs that we want to demonstrate (**Figure R2**).

We appreciate the reviewer's suggestion, and in the revised **Methods** section (**Page 28, Line 720**), we have provided a detailed description of nanoparticle flow cytometry and the utilization of standard beads for detection.

Figure R2 Quantification of PD-L1⁺, PD-1⁺ and CD80⁺ sEVs in healthy donors (HDs) and HNSCC patients. The data were presented in percentage (% , upper panel) and relative fluorescence intensity (RFI, lower panel).

Innovation issue

1. Innovation of the data is partly compromised by previously reported information about the participation of the checkpoint proteins, especially PD-L1 or PD-1/CD80 in cellular interactions, including the senior authors' previously published work. Release by immune cells of sEV bearing PD-1/CD80 in patients with HNSCC and other tumours and correlations with anti-tumour immunity and disease activity have been previously reported and are not referenced here (especially some the data presented as novel in Figures 1 and 2 are largely confirmatory).

Response: We agree that cell immune checkpoints, especially PD-L1 and PD-1, have always been a hot topic in the field of tumour immunology. However, generating new insights and understanding on this not-so-new topic remains crucial for driving innovation in cancer immunotherapy, as existing research has not fully unveiled the complexities involved. Over the past 5 years, works from a great many research group including our own have discovered that immune checkpoints carried on sEVs represent a novel form of interaction between immune checkpoints, being distinct from conventional cell-to-cell manners⁵⁻⁹. These, to some extent, explain the intense immune suppression and poor response rates to immunotherapy. However, most studies in this field have focused primarily on sEV PD-L1; while the presences, sources, and functions of sEV-carried immune checkpoints other than PD-L1 are still relatively limited and superficial.

Best to our knowledge, the detection of PD-1 and CD80 in sEVs of patients with head and neck cancers was first introduced by Whiteside et al. in 2017¹⁰. Thus, we respectfully appreciate their contribution on revealing their presence and referenced their work in the **Introduction** and **Discussion** section of the original manuscript (**original citation 28** and **31**; in **revised manuscript citation 27** and **28**). Additionally, we acknowledged more recent research by Mien-Chie Hung group (**original citation 29**; in **revised manuscript citation 29** and Amalia Azzariti group (**original citation 30**; in **revised manuscript citation 30**) regarding the correlation between sEV PD-1 and anti-tumour immunity as well as disease activity. However, as stated in our discussion, the former study primarily indicated a positive role of sEV PD-1 through consuming detrimental sEV PD-L1, based solely on *in vitro* results; while the latter one suggested a negative impact of sEV PD-1 in anti-tumour immunity without elucidating the underlying mechanism. As for sEV CD80, another protein with interaction with PD-L1, there has been limited research attention thus far, to the best of our knowledge. Therefore, the novelty of our study lies in elucidating the

mechanisms through which sEV PD-1/CD80, mainly released by immunocytes, impairs anti-tumour immunity.

We have carefully reviewed previous publications, and we would greatly appreciate it if the reviewer could kindly inform us of any important literature that we may have overlooked. Meanwhile, considering the existing literatures on the association of sEV PD-1/CD80 with HNSCC progression, as suggested by the reviewer, we are willing to reorganize **Fig. 1** and corresponding supplementary figures without compromising the logical flow of the manuscript. This reorganization aims to reduce the repetition of known results, minimize the length of the manuscript, and better emphasize the novelty of our study.

2. Also, the fact that injections of tumour-derived sEV (TEX) into tumour-bearing mice reduced the numbers of CD8⁺ T cells infiltrating the tumour has been previously reported in murine oral carcinogenesis model (Razzo, Ludwig et al, Carcinogenesis).

Response: We have carefully reviewed the entire manuscript and did not find any mention of using tumour-derived sEVs to treat tumour-bearing mouse models in our study. Instead, we used sEVs derived from activated T cells. Thus, it is possible that our study might have been misinterpreted by the reviewer.

We acknowledge that tumour-derived sEVs reduced infiltrated CD8⁺ T cells through PD-L1 and FasL in tumour microenvironment of mice model^{6,7,11-15}. In fact, it has been reported by tremendous amount of publications including our own⁵ that tumour-derived sEVs impaired immune system via inhibitory immune checkpoint PD-L1. However, there are few studies that explicitly elucidate the reasons behind the increased secretion of sEV PD-L1. Our study actually provides mechanistic insights into the increased secretion of PD-L1⁺ sEVs by tumour cells. We demonstrate that sEV PD-1/CD80, mainly secreted by activated immunocytes, can lead to the re-distribution of PD-L1 on tumour cells through an ESCRT-dependent manner, resulting in immune desert of the tumour microenvironment and ultimately leading to systemic

immunosuppression. Yet, to the best of our knowledge, we have thoroughly reviewed the existing literature and have not found any evidence from previous studies specifically addressing the regulation of CD8 infiltration by aT-sEVs.

3. On the other hand, the data showing that PD-1/CD80(+) aT-sEV promoted secretion of PD-L1 by tumour cells is novel and should be the major focus of this manuscript.

Response: Thank the reviewer for the suggestion on the novelty and significance of our findings regarding the promoted secretion of PD-L1 in tumour cells in response to aT-sEV PD-1/CD80. We agree that this aspect is a major focus of our manuscript, and we have re-organized the structure of the manuscript based on the reviewer's suggestions, trimming certain sections without compromising the overall logical coherence. This allows us to better focus on the main topic and highlight the novelty of our research.

4. The mechanism responsible for reduction in PD-L1 on the tumour surface due to its internalization following ligation by PD-1/CD80 (+) is an expected effect of receptor/ligand interactions and was previously reported for PD-1/PD-L by G. Freeman among others.

Response: The interaction between PD-1 and PD-L1 demonstrated by G. Freeman could be dated back to 2000, and we believe that there has been still enormous number of studies on PD-1 and PD-L1 after that. Although the outcomes of ligand binding on membrane protein fate can be relatively predictable, we did not find any literature documenting the regulation of cell surface PD-L1 levels by sEV PD-1, let alone reports on the role of sEV CD80 in this context.

We have indeed considered the valuable suggestions from the reviewer. As mentioned above, in the revised version, we have reduced potentially known contents to emphasize the novelty of our research while ensuring the integrity of the manuscript.

5. The data related to reprogramming of tumour cells by aT-sEV are not novel and the secretion by reprogrammed tumours of immunosuppressive sEV has been previously reported.

Response: We appreciate the reviewer's comment regarding the previously reported reprogramming of tumour cells by aT-sEV and the secretion of immunosuppressive sEV by reprogrammed tumours. We acknowledge that these aspects have been reported in some of the literatures. However, the concept of reprogramming itself is indeed ambiguous, encompassing both positive and negative effects. Based on our literature search, the specific regulatory effects of aT-sEVs on tumour cells and the underlying mechanisms are still under debate and require further investigation. In our study, we clearly propose that aT-sEVs stimulate several types of tumour cells, including HNSCC, to exhibit a negative regulatory phenotype that suppresses tumour immunity. Furthermore, we provide an explanation for the molecular mechanisms involved in this process.

We are certainly aware that tumour cells can secrete sEVs, leading to systemic immune suppression, as reported in many studies. However, it is evident that the highlight of our study is not focused on the immunoregulatory effects of tumour cell-derived sEVs. Instead, it lies in elucidating the reasons behind the aberrant elevation of immunosuppressive sEVs in tumour patients, with aT-sEV PD-1/CD80 being an important initiating factor. To the best of our knowledge, no previous research has proposed that PD-1/CD80 carried on immunocyte-derived sEVs conceal tumour immunogenicity and strengthen immune suppression. Accordingly, we have made revisions to the manuscript to provide appropriate context and references acknowledging these previous findings. Furthermore, we have placed greater emphasis on our insights into the underlying mechanisms of this reprogramming process and the specific role of sEV PD-1/CD80 in promoting PD-L1 secretion.

Other points

1. FRET (energy transfer) assays in Figure 3 illustrating the co-localization of PD-1/CD80 and potential cis /trans interactions are interesting, but the presentation of these data is confusing and needs clarity.

Response: Thank for the reviewer's positive comment regarding the FRET assays presented in **Fig. 3** (revised **Figure 4**). We apologize for any confusion caused by the presentation of these data. We have provided a more detailed description in **Methods** section (**Page 30, Line 772**) and revise the corresponding figure legends to enhance the understanding of these experiments.

In our study, we investigated the molecular ligation between PD-1 or CD80 on sEVs and PD-L1 on tumour cells using a Förster resonance energy transfer (FRET) assay. This assay is based on the principle of rapid energy transfer from a donor molecule to an acceptor molecule when they are in close proximity, leading to the emission of fluorescence. In our experimental setup, cells were incubated with a tyramide594-biotin-labelled anti-PD-L1 antibody, which served as the energy acceptor and emitted fluorescence upon energy transfer. As for energy donor, aT-sEV PD-1 was labelled with AF488. When PD-1 labelled with AF488 interacts with tyramide594-labeled PD-L1, excitation at 488 nm activates AF488, which then transfers energy to tyramide594, resulting in the detection of tyramide594 fluorescence. Therefore, when tyramide594 undergoes photobleaching, fluorescence is quenched, and AF488 cannot continue to transfer energy, resulting in enhanced AF488 fluorescence. The difference in fluorescence intensity of the energy donor before and after photobleaching is referred to as FRET efficiency, indicating that the two target molecules are in close proximity and have a physical interaction.

Thank the reviewer for bringing this to our attention, and we have addressed this concern in the revised manuscript.

2. The PD-1/PD-L1 pathway is not the only molecular pathway involved in these interactions, and its contribution to immunosuppression in the TME of HNSCCs may be trumped by TGF- β and adenosine-mediated pathways.

Response: We appreciate the reviewer's comment highlighting the complexity of the tumour microenvironment (TME) in HNSCCs and the involvement of multiple molecular pathways in immunosuppression. Indeed, in this study, we validated the interaction between PD-1/CD80 and PD-L1 both *in vitro* and *in vivo*. Through experimental approaches such as blocking antibodies against target proteins on sEVs or tumour cells, as well as key protein gene knockout, we demonstrated that the disruption of the aforementioned interaction resulted in a reduction of downstream immune suppression. Therefore, we proposed that aT-sEV PD-1/CD80 exerts its immunosuppressive function through the surface PD-L1 on tumour cells. This is logically sound and comprehensive. It's important to note that throughout our manuscript, we haven't asserted that this effect is the exclusive underlying mechanism. We acknowledge that other signalling pathways, as mentioned by the reviewer, such as TGF- β and adenosine-mediated pathways, might potentially play a role in the interaction between immune cell-derived sEVs and tumours. We are indeed open to exploring a comprehensive understanding of these potential mechanisms.

To address the concern, we tested the activation of these pathways through high-throughput data analysis and western blotting. As shown in **Figure R3**, mass spectrometry and western blot analysis was performed to evaluate the level of TGF- β , as well as CD39 and CD73, the ectonucleotides for adenosine production. However, the results showed that aT-sEVs did not alter the levels of proteins tested, indicating that these two might not contribute the downstream effects in the specific context of this study. We appreciate the reviewer for bringing this to our attention.

Figure R3 Detection of TGF-β and adenosine-mediated pathways in tumour cells. (a) Mass spectrometry analysis for TGF-β and adenosine-associated protein CD39 and CD73 in CAL27 cells. (b) Western blot analysis was performed to evaluate the level of TGF-β, as well as CD39 and CD73, the ectonucleotides for adenosine production.

Overall comments

Overall, this manuscript presents an overwhelming volume of data some of which are confirmatory and others new that are split between the main text and Supplemental Materials. The most important and potentially clinically relevant data on the effects PD-1/CD80 (+) sEV produced by activate T cells exert on the tumour in patients with HNSCC appear in Supplemental Materials. The Figures lack informative legends, and the reader is forced to navigate between the main text and Supplemental Figures. This makes for difficult reading and does not enhance our understanding of the interactions between immune sEVs and the tumour.

Response: We appreciate the reviewer's feedback regarding the organization and presentation of the data. In response to this concern, we have restructured the manuscript to ensure that the most important and clinically relevant data on the effects of sEV PD-1/CD80 produced by activated T cells in patients with HNSCC are included in the main text. We have also improved the clarity of the figure legends and made efforts to enhance the reader's understanding. We believe these revisions will improve the overall readability and comprehension of the manuscript. Meanwhile, we also extend our gratitude to the reviewer for its valuable suggestions. We sincerely apologize if any misinterpretation of our

research results or experimental designs has arisen due to our descriptions or other reasons. We hope that our revisions will help clarify any misunderstandings the reviewer might have had, and we are committed to addressing any concerns raised.

Reviewer #2

Comments to the Authors

Overall, this is a very well designed and supported research paper. Understanding the role that exosomes from immune cells play in cancer progression is an exciting new frontier in the field. Previously, the focus in the field has been the effect of tumour derived exosomes on immune cells. Here, the authors investigate just the opposite. This paper addresses a specific mechanism by which a particular cohort of immune cell derived exosomes influence tumour immunogenicity. The authors did a very thorough job detailing the phenotype of these immune cell exosomes and exploring their effect on tumour cells. It is quite a clever adaptive mechanism they identified in that exosomes from activated T cells cause tumour cells to downregulate PD-L1 and become immunologically cold.

Understanding the mechanisms stratifying patients into responders versus non-responders has become the holy grail, if you will, in immuno-oncology. Teasing apart the biomarkers in a patient's blood to determine if a particular therapy will be effective is the pinnacle of clinical importance. This paper aims to suggest one new strategy for the prediction of immunotherapy response. Therefore, the novelty of this paper is twofold. The authors first identify a new mechanism by which tumour cells avoid immunotherapy as mentioned above. Second, they suggest sEV phenotyping using PD-1/CD80 and PD-L1 expression can help predict responsiveness to immunotherapy.

Response: We thank the reviewer for the positive feedback on our manuscript. In the present study, we thoroughly characterized the phenotype of immunocyte-derived sEVs and their impact on tumour cells, unveiling a clever adaptive mechanism where sEVs from activated T cells induce immunologically cold tumour cells. Our study aims to contribute to the holy grail of immuno-oncology by suggesting a novel strategy for predicting immunotherapy response. Also, we sincerely appreciate the reviewer's constructive suggestions, which have significantly contributed to the refinement of our work.

With confidence, we believe that the revised manuscript and the point-by-point response would adequately address the concerns raised by the reviewer.

Major

1. Are the sEVs that express PD-L1 the same sEV that express PD-1 and CD80? PD-L1 expression seemed to be the most sensitive and specific marker to delineate HNSCC vs HD as demonstrated in supplemental figure 1b.

Response: We appreciate the reviewer's concern. First and foremost, it is important to clarify that based on the results from western blotting and nanoparticle flow cytometry, the majority of PD-1⁺ or CD80⁺ sEVs are not EpCAM-positive, indicating that tumours are not their primary source (**Supplemental Fig. 2a**). Accordingly, studies including ours have consistently shown that PD-1 and CD80 are rarely detectable on the surface of various tumours, including HNSCC. Therefore, it can be inferred that tumour cells can secrete PD-L1⁺ sEVs without co-expressing PD-1 and/or CD80 on their surface. Since PD-L1 can be also expressed in immunocytes, it is expected that immunocyte-derived sEVs would contain all three molecules (PD-1, CD80, and PD-L1). However, considering that tumour cells predominantly secrete PD-L1⁺ sEVs, we can speculate that only a small fraction of PD-L1⁺ sEVs in circulation would simultaneously express PD-1 and CD80. To verify, we conducted nanoparticle flow cytometry analysis on circulating sEVs, utilizing multiple labelling of PD-L1, PD-1, and CD80. As shown in **Figure R4**, triple positive sEVs only account for 2.5% of the total circulating sEVs. And the results in **Figure R4** have now been included in **Supplementary Fig. 1b** in the revised manuscript.

Although in **Fig. 1c**, it may appear that the difference in sEV PD-L1 levels between HNSCC and HD is more pronounced, the *p*-values are actually at the same significance level (**Fig. 1d-f**). Thus, we refrain from asserting that sEV PD-L1 is the most sensitive and specific marker for distinguishing HNSCCs from HDs. In our previous research, we demonstrate the on-treatment

dynamics of sEV PD-L1 could be utilized to predict response to immunotherapy⁵. However, the long working procedures and narrow time window still limited the application of sEV PD-L1 as a prediction biomarker. Consistently, as shown in **revised Figure 7c**, even with low level of circulating sEV PD-L1, some of HNSCC patients were still non-responders to anti-PD-1 immunotherapy. This is also the reason why we propose a clinical workflow for stratifying immunotherapy responders from non-responders, which incorporates multiple factors including sEV PD-1/CD80 and sEV PD-L1.

Figure R4 Nanoparticle flow cytometry analysis on circulating sEVs with multiple labelling of PD-L1, PD-1, and CD80. The proportion of sEVs in each subpopulation was presented using a bar graph (left panel) and a Venn diagram (Right panel), respectively.

2. Is the conclusion that PD-L1⁺ sEV are the predominate type overall in the blood of HNSCC patients whereas the PD-1/CD80⁺ immune cell sEVs are the second most common and a novel population in cancer patients that contribute to tumour pathogenesis?

Response: Based on our findings, we have observed that PD-L1⁺ sEVs accounted for approximately 40% of the total circulating sEVs in HNSCCs, while PD-1⁺/CD80⁺/PD-1⁺CD80⁺ sEVs constituted approximately 60% of it, with some overlap between these subtypes (**revised Supplementary Fig. 1b**). Thus, our conclusion is not simply that PD-L1⁺ sEVs are the most significant and PD-1⁺/CD80⁺/PD-1⁺CD80⁺ sEVs are the second most significant factors contributing to tumour development in tumour patients. What we have

demonstrated is that elevated levels of sEV PD-1/CD80 in the circulation of tumour patients are one of the main factors leading to the feedback secretion of PD-L1⁺ sEVs by tumour cells, resulting in increased levels of circulating sEV PD-L1. This process is also a crucial factor contributing to systemic immune suppression in tumour patients and is associated with poor response to immunotherapy. Thanks for the reviewer's question. We have also reorganized the manuscript to better present the highlights of our study.

Minor

1. Patients in this study are currently on anti-PD-1 therapy and chemotherapy. Are they on the same chemotherapy regimen? Do they all have a similar starting point in terms of exosome composition after washout from previous treatments? Was there a difference at baseline in sEV composition in primary versus recurrent HSCC?

Response: In this study, all the patients here were receiving the same combination chemotherapy regimen of paclitaxel plus cisplatin (TP). However, they do not have a similar starting point in terms of sEV composition before they started immunotherapy. As shown in **Figure R5**, all 23 tumour patients receiving immunotherapy did not have the same baseline levels of total circulating sEV number and protein concentration, as well as expression levels and of sEV PD-1, sEV CD80, and sEV PD-L1. Instead, these parameters exhibited a distribution similar to a normal distribution. In fact, this outcome is expected because prior to the onset of treatment, the circulating sEVs of these patients had varying levels of immune checkpoint molecules, which contributed to their diverse responses to immunotherapy. This is the key focus of our study and serves as the rationale for developing predictive treatment responses. We have also tested the baseline in sEV composition in primary and recurrent HNSCC patients, as the reviewer suggested. As shown in **Figure R6**, for each individual HNSCC patient, the composition of sEVs appears to be dynamic and not consistently maintained between the primary and recurrent stages. While

the levels are generally comparable, based on the currently available data, we have not been able to reach a definitive conclusion.

To enhance the accuracy and comprehensibility of the manuscript, we have included a detailed description of the patient information in the **Methods** section (Page 25, Line 649) and **Supplemental Table 1**.

Figure R5 Detection of circulating sEV composition in HNSCC patients who underwent immunotherapy before treatment started.

Figure R6 Detection of circulating sEV composition in primary and recurrent HNSCC patients. a, Comparison of the number of circulating sEVs between primary and recurrent HNSCC patients. b, Comparison of the protein content of circulating sEVs between primary and recurrent HNSCC patients. c-d, Comparison of the RFI of PD-L1 (a), PD-1 (b) or CD80 (c) on circulating sEVs between primary and recurrent HNSCC patients. Error bars represented mean \pm s.d., statistical analyses were performed using two-sided paired t-test (a-e).

2. Supplemental Figure 1E- do you have the flow plot for this Venn diagram?

Response: We thank the reviewer for the suggestion. **Supplementary Fig. 1** does not include “E”. If we understand correctly, the reviewer is likely referring

to **Supplementary Figure 2E**. We are more than pleased to provide the corresponding flow plot (**Figure R7**). It should be noted that when calculating the proportions in the Venn diagram, we excluded PD-1⁻CD80⁻ sEVs. We exhibited the proportion of PD-1⁺CD80⁺ sEVs among the PD-1⁺/CD80⁺/PD-1⁻CD80⁺ sEVs, aiming to demonstrate the synchronous expression of PD-1 and CD80 in circulating sEVs. To better showcase our results, based on the reviewer's suggestion, **Figure R7** has been included in the revised manuscript as **Figure 1j**.

Figure R7 Detection of circulating PD-1⁺, CD80⁺ or PD-1⁺CD80⁺ sEVs composition in HNSCC patients. a, Representative contour plots of sEVs examined for the expression of PD-1, CD80 or co-expressing PD-1 and CD80. Quantification for the percentages of PD-1⁺ sEVs, CD80⁺ sEVs and PD-1⁺CD80⁺ sEVs in HNSCC patients.

3. Figure 1E and Supplemental Figure 2F- is it possible to plot tumour burden and survival against co-expression of PD-1/CD80 instead of just one surface marker alone?

Response: Plotting survival against the coexpression of PD-1/CD80 could be provided. As shown in **Figure R8**, patients with simultaneous elevation of sEV PD-1 and sEV CD80 have worse prognosis, and the tendency is consistent with the analysis of a single surface marker alone (**revised Supplementary Fig. 1g**). However, analysing the correlation between sEV PD-1/CD80 co-

expression and tumour burden is challenging. This is because in our previous analysis, we measured the protein concentrations of PD-1 and CD80 in circulating sEVs using ELISA, and logically, it is not feasible to simply combine the concentrations of these two different proteins together for correlation analysis.

Although we mentioned in the results that there is a potential synchrony in the expression of PD-1 and CD80 in circulating sEVs, it is important to note that PD-1⁺/CD80⁺/PD-1⁺CD80⁺ sEVs all can exert downstream effects. Therefore, we do not advocate solely analysing the double-positive subset, as this may overlook the role of the single-positive subsets.

Figure R8 Overall survival rate of HNSCC patients with elevated level of PD-1⁺CD80⁺ sEVs. Error bars represented mean \pm s.d., statistical analyses were performed using log-rank test.

4. I understand that the majority of PD-1/CD80 sEV are CD45⁺ but could you provide the flow plots and gating strategy for supplemental Figure 4A? I think it is an important concept to demonstrate within the overall sEV pool what percentage are CD45⁺ and within that particular pool the prevalence of PD-1/CD80⁺ sEV? Again, this goes back to the above question regarding PD-L1 vs PD-1/CD80. Are CD45⁺ sEV PD-L1 positive? What is the relative abundance of sEV PD-L1⁺ versus CD45⁺/CDPD-1/CD80⁺ in HNSCC?

Response: We apologize for any data inadequately presented in the manuscript. As shown in **Figure R9a**, we have determined the proportion of CD45⁺ particles within the subset of PD-1⁺ or CD80⁺ sEVs by gating them from the total circulating sEVs. However, this gating strategy does not address the reviewer's question regarding the percentage of CD45⁺ particles in the overall

sEV pool and the prevalence of PD-1/CD80⁺ sEVs within that specific pool. To address this concern, we performed gating to isolate CD45⁺ sEVs from the circulating sEV pool, and subsequently analysed the presence of PD-1⁺ or CD80⁺ sEVs within the CD45⁺ subtype. Among the CD45⁺ sEVs, approximately 40-50% were PD-1⁺ and CD80⁺, respectively, while less than 20% were PD-L1⁺ (**Figure R9b**). Furthermore, as previously mentioned in **Figure R4**, PD-L1⁺ sEVs comprised around 40% of the total circulating sEVs in HNSCCs, while PD-1⁺/CD80⁺/PD-1⁺CD80⁺ sEVs constituted approximately 60% of the total, with some overlap observed between these subtypes. We further examined the composition of PD-L1⁺ sEVs and found that the EpCAM⁺ subpopulation was the predominant subtype (**Figure R9c**). Interestingly, we also observed a subpopulation of CD45⁺ sEVs within the PD-L1⁺ subtype, suggesting that immune cell-derived sEVs contribute to the presence of PD-L1⁺ sEVs in the circulation.

We appreciate the reviewer's question, as it is indeed an important one. At the current stage, we have made efforts to our best to perform multi-colour labelling on circulating sEVs to analyse their subpopulations, similar to conventional cell flow cytometry. However, considering the unique physicochemical properties of sEVs, we acknowledge that there are still some technical limitations that need to be further explored.

Figure R9 Detection of circulating sEVs composition in HNSCC patients. a, Flow plots and gating strategy for CD45⁺, CD144⁺, and EpCAM⁺ subpopulation in PD-1⁺ or CD80⁺ sEVs. b, Flow plots and gating strategy for PD-L1⁺, PD-1⁺, and CD80⁺ subpopulation in CD45⁺ sEVs (left). Quantification the data (right). c, Flow plots and gating strategy for CD45⁺, CD144⁺, and EpCAM⁺ subpopulation in PD-L1⁺ sEVs (left). Quantification the data (right). Error bars represented mean \pm s.d., statistical analyses were performed using one-way ANOVA (b, c).

5. Figure 1F- Could you include the CD3 staining in the heatmap?

Response: Thanks for the reviewer's suggestion. We appreciate the interest in including CD3 staining in the heatmap of original Fig. 1F (**revised**

Supplemental Fig. 2b). We understand that CD3 is an important marker for T cells, and its inclusion can provide additional information on the immune cell composition. As shown in **Figure R10**, both CD3⁺PD-1⁺ and CD3⁺CD80⁺ sEVs distinguished HNSCC patients from healthy donors. The importance score of CD3 has also been calculated. We have carefully considered the suggestion from the reviewer and incorporated CD3 staining in the heatmap in the revised manuscript.

Figure R10 Heat maps illustrated the levels of immunocyte markers on circulating PD-1⁺ and CD80⁺ sEVs. CD3, CD4, CD8, CD11c, CD19 and CD68 were detected in circulating PD-1⁺ and CD80⁺ sEVs from HD (n = 36) and HNSCC patients (n = 46). Bar plots showed the importance score of target proteins on immunocyte-derived sEV PD-1 (top) and CD80 (bottom) in distinguishing HNSCC patients from HDs.

6. Supplemental Figure 4E- based on the method section, it looks like the macrophages were polarized to an M2 phenotype. Literature suggests that both M1 and M2 macrophages play a role in tumour pathogenesis. What kind of sEVs do M1 macrophages secrete?

Response: We appreciate the reviewer's concern on the polarization of macrophages. We additionally activated macrophage with IFN- γ and LPS, inducing M1 phenotype. With western blotting, we were able to detect the

expression of PD-1 and CD80 in sEVs derived from M1 macrophages, similar to what we observed in M2 macrophages (**Figure R11a**). Taking advantage of analysis of MESF, the levels of sEV PD-1 and CD80 from M1 macrophage were illustrated in heat map (**Figure R11b**).

It is important to note that we only performed a preliminary analysis to investigate the potential differences in PD-1 and CD80 expression in sEVs derived from M1 and M2 macrophages. Considering the distinct roles of these two macrophage phenotypes in tumour immunity, we believe that their secreted sEVs likely exhibit significant compositional differences. However, it is important to emphasize that this was not the primary focus of our study.

Figure R11 Characteristics of sEVs secreted by M1 polarized macrophages. a, Western blot analysis of PD-1 and CD80 in whole cells and sEVs from activated M1 macrophages and control cells. b, Heat map showing the relative levels of PD-1 and CD80 in sEVs from activated M1 macrophages and control cells.

7. Supplemental Figure 4 G/H- Could you provide CD3 and/or CD8 staining for both the WB and TEM figuring to confirm these are indeed T cell derived sEV?

Response: We thank the reviewer for the beneficial suggestion. We supplemented the detection of CD3 in iodixanol density gradient fractions, proving that the sEVs we purified were T-cell derived (**Figure R12a**). With TEM, immunogold-labelled with CD3 was detected on T cell-derived sEVs, further confirming its source (**Figure R12b**). We have now included the results in **revised Supplementary Fig. 2d, e**.

Figure R12 Detection of CD3 in sEVs purified from T cells. a, Standard density gradient centrifugation analysis showing that PD-1 and CD80 co-fractionated with sEV markers Hrs, ALIX, TSG101, and CD81 and T cell-specific marker CD3. b, A representative TEM image of activated T cell-derived sEVs immunogold-labelled with anti-CD3, PD-1 or CD80 antibodies. Scale bar, 50 nm.

8. Do patients with HNSCC have a higher number of immune cells compared to HD and that is why you see increased sEV with PD-1/CD80 markers?

Response: We thank the reviewer for raising these questions. Previous research¹⁶ and our own has demonstrated that a similar level of total CD3-positive T lymphocytes in tumour patients and healthy donors, indicating the number of T cells did not significantly change before and after tumourigenesis. PD-1 was also labelled in sEVs secreted by unstimulated T cells but barely detected with TEM (**Figure R13a**). With nanoparticle flow cytometry, we confirmed the increased level of PD-1 in individual sEVs from activated T cells when compared to that from non-activated ones (**Figure R13b**). Meanwhile, as shown in **Figure R13c**, the secretion level of sEVs was obviously enhanced after T cell activation, which meant both sEV quantity and PD-1 carried by each sEV were increased after T cell activation.

As for the number of immune cells, we analysed the blood analysis results from patient admission information and found that the overall number of

immunocytes in HNSCC patients did not significantly deviate from the normal range (**Figure R14**). This suggests that changes in cell numbers are not the primary factor influencing sEV PD-1/CD80 levels, but rather the cellular state.

Figure R13 Detection of PD-1 and CD80 in T cell-derived sEVs. a, A representative TEM image of unstimulated T cell-derived sEVs immunogold-labelled with anti-PD-1 antibodies. b, Nano-FCM analysis for PD-1 on individual sEV from unstimulated and stimulated T cells. c, The number of sEVs analysed by NTA. Nano-FCM, nanoparticle flow cytometry; NTA, nanoparticle Tracking Analysis. Error bars represented mean \pm s.d., statistical analyses were performed using two-sided paired t-test (b, c).

Figure R14 Quantification of immunocytes in HNSCCs with blood analysis results from patients' admission information. Error bars represented mean \pm s.d., statistical analyses were performed using two-sided unpaired t-test.

9. Figure 1K- the legend graft colours are off and don't line up with the words
Response: We apologize for missing the legend graft colour in original Figure 1K. We have revised the figure to ensure that the lines align correctly with their respective legend graft (**revised Fig. 1g, h**).

10. Supplementary Figure 9C- why do you think the T cell derived sEV localize to the tumour? Are there specific integrins driving this phenotype?

Response: We thank the reviewer for raising this question. Indeed, we have explored the mechanism of sEV PD-1 interplaying with tumour cells. A recent study discovered that adhesion molecules like ICAM-1 were essential for interaction between sEVs from tumour cells and T cells in sEV PD-L1-mediated immune suppression¹⁷. Hereby, we asked whether the affinity was reasoned from the ligation of PD-L1 to PD-1/CD80 or ICAM-1 to LFA-1. As shown in **Figure R15a**, with western blot, we first found that PD-1, CD80 and LFA-1 enriched in immune cells while PD-L1 and ICAM-1 in tumour cells. By immunofluorescence staining, we then found co-localization of incubated aT-sEVs and membrane PD-L1 and ICAM-1 on tumours (**Figure R15b**). However, blocking PD-L1 or CD80 failed to attenuate the fluorescence intensity of CFSE-labelled aT-sEVs detected on tumour cells (**Figure R15c, d**), but eliminated downstream effects including redistribution of PD-L1 in tumour cells. We further found that blocking either ICAM-1 on tumour cells or LFA-1 on aT-sEVs significantly decreased the binding of aT-sEVs to tumour cells (**Figure R15c-d**). Additionally, we also found that the binding of aT-sEVs to tumour cells was abrogated by blocking LFA-1 on aT-sEVs *in vivo*, demonstrated by *in vivo* fluorescence imaging analysis cells (**Figure R15e**). Altogether, these findings suggest that adhesion molecules ICAM-1 and LFA-1 regulate the physical affinity between sEVs and tumour cells, increasing the chance of interaction between PD-1/CD80-PD-L1, which is a prerequisite for generating significant downstream effects of PD-1/CD80-PD-L1 signalling axis. PD-1/CD80 and PD-L1 themselves do not participate in adhesion regulation.

In summary, these results suggest a physical and efficient ligation between aT-sEVs and tumour cells in an ICAM-1-LFA-1-dependent manner.

Figure R15 PD-L1 on tumour cells ligate with PD-1/CD80 on sEVs in an ICAM-1-LFA-1-dependent manner. a, Western blot analysis of LFA-1 in whole cells and sEVs from activated T cells and control cells (top). Western blot analysis of PD-L1 and ICAM-1 in CAL27, H1264, MDA-MB-231 and A375 cells (bottom). b, Confocal microscopy analysis of PD-L1 (top) and ICAM-1 (bottom) in Cal27 cells colocalization with CFSE-labelled aT-sEVs. c, Representative histograms of Cal27 cells with or without ICAM-1 antibody blocking cocultured with aT-sEVs (left). Quantification the data (right). d, Representative histograms of Cal27 cells cocultured with aT-sEVs with or without LFA-1 antibody blocking (top). Quantification the data (bottom). e, Confocal microscopy analysis for CSFE labelled aT-sEVs binding to tumour cells in tumour environment (left).

Quantification the data (right). Error bars represented mean \pm s.d., statistical analyses were performed using one-way ANOVA (c, d, e).

11. Supplementary Figure 9E- Will the tumour cells uptake any kind of EV vesicle or is this specific to the T cell derived sEV? If you pre-treat the sEV with anti-PD-1 antibody, do you lose this association on the tumour cell surface?

Response: Previous research demonstrated that tumour cells uptook sEVs from various cell type including immunocytes, endothelial cells and fibroblast¹⁸⁻²⁰. In our study, based on the characterization of circulating sEVs from patients, we used aT-sEVs as sEV PD-1/CD80 to stimulate tumour cells. We believe that in addition to T cell-derived sEVs, sEVs secreted by other immune cells carrying LFA-1 may also interact with tumour cells according to the results in **Figure R15** mentioned above. To further verify, we treated tumour cells with sEVs from other immunocytes, revealing that sEVs derived from B cells and macrophages could interact with tumour cells as well (**Figure R16**).

Meanwhile, as we proved in **Figure R15**, blocking PD-1 or CD80 would not attenuate the binding of aT-sEVs on tumour cells. However, the binding of aT-sEVs to tumour cells was abrogated by blocking LFA-1. Thus, the affinity of aT-sEVs to tumour cells was initiated by LFA-1-ICAM-1 but not PD-L1-PD-1/CD80. Nevertheless, blocking the PD-L1-PD-1/CD80 signalling axis would still result in the elimination downstream effects triggered by aT-sEVs in tumour cells. Overall, our findings support that the signal transduction through PD-L1-PD-1/CD80 interaction relies on the binding facilitated by LFA-1-ICAM-1, which is in accordance with a prior study highlighting the role of tumour sEV ICAM-1 in interacting with T cells¹⁷.

Figure R16 Immunocyte-derived sEV bond to tumour cells. a, Confocal microscopy analysis of tumour cells after incubation with CFSE-labelled activated B cell or macrophage-derived sEVs for 6 h. b, Representative histogram of tumour cells after incubation with CFSE-labelled activated B cell or macrophage-derived sEVs for 6 h.

12. Line 425- trans-ligation is misspelled.

Response: We thank the reviewer for pointing out the spelling error. We apologize for the typo. We have corrected the spelling of "trans-ligation" in the revised manuscript.

13. Supplemental Figure 10D- those errors bars are wide, more replicates would be more convincing. It is an important point to note if PD-L1 mRNA is transferred from T cell derived EV or not.

Response: We appreciate the reviewer's comments regarding the error bars in Supplemental Figure 10D. To strengthen the data, we conducted additional experiments and increased the replicates to five. Similarly, treatment of aT-sEVs did not alter the *PD-L1* mRNA level in tumour cells (**Figure R17a**). We understand that it is important to address whether *PD-L1* mRNA is transferred from T cell-derived EVs. Thus, we evaluated the level of *PD-L1* mRNA in these aT-sEVs. As shown in **Figure R17b**, the *PD-L1* mRNA was hardly detected by PCR. Considering that the levels of PD-L1 mRNA in tumour cells treated with aT-sEVs did not show significant changes, this negative result leads us to lean towards the absence of substantial PD-L1 mRNA transfer in the aforementioned process. We have now updated the **original Supplementary Fig. 10d** as the **revised Supplementary Fig. 5d**, incorporating the results in **Figure R17a**.

Figure R17 a, Quantification of *PD-L1* mRNA level in CAL27 cells without or with aT-sEVs treatment; **b**, Quantification of PD-L1 mRNA level in CAL27 cells and T-cell-derived sEVs. Error bars represented mean \pm s.d., statistical analyses were performed using two-sided unpaired *t*-test.

14. Figure 3D- very nice experiment.

Response: We thank the reviewer for the positive feedback on Figure 3D. We appreciate your recognition of our experiment.

15. Figure 3D- I assume EEA and Rab7 are endosome associated markers, I would specify this in the manuscript.

Response: We apologize for not specifying in the manuscript that EEA1 (Early Endosome Antigen 1) and RAB7 are indeed early and late endosome markers, respectively (**Page 12, Line 296**). We have provided this clarification in the revised manuscript.

16. Figure 4B-E- How are you measuring the secreted PD-L1? Are these sEV collected from the supernatants of the tumour cells before and after stimulation? Could the T cell derived sEV have PD-L1 on them which would be increasing overall PD-L1 expression instead of just secretion from the tumour cells?

Response: In Figures 4B-E, we measured the secreted sEV PD-L1 levels using nanoparticle flow cytometry on the collected supernatants of the tumour cells after stimulation of aT-sEVs. Thus, we understand the reviewer's concern regarding the potential impact of aT-sEVs on the quantification of sEV PD-L1

secreted by tumour cells. Here, we apologize for the inadequate description of aT-sEV treatment in the **Methods** section (**Page 31, Line 794**). In fact, in the experiments evaluating the impact of aT-sEVs on tumour cell sEV PD-L1 secretion, after treating the tumour cells with aT-sEVs for 6 hours, we performed PBS washes to remove any residual aT-sEVs. The cells were then cultured in sEV-free medium for an additional 24 hours before measuring the levels of sEV PD-L1 in the supernatant. This approach was implemented to minimize any potential interference from the aT-sEVs, ensuring that the measured sEV PD-L1 levels primarily reflect the secretion by the tumour cells themselves.

We appreciate the reviewer's question, and we have provided additional details in the **Methods** section regarding this experiment.

17. Figure 4F- can you explain the results of the WCL WB? I am not sure I understand why TSG101 levels would go down when you add T cell derived sEV to the supernatant?

Response: In the **original Fig. 4F** (the **revised Fig. 4i**), the western blotting showed a decrease in TSG101 levels in the whole-cell lysate of CAL27 cells after treatment of aT-sEVs. Sorting of cargo into sEVs involves TSG101 associated with ESCRT pathway. Thus, TSG101, exclusively found within sEVs, is a classically used sEV marker^{5,21-23}. The explanation for the decrease in TSG101 levels could be the activation of sEV secretion in tumour cells after treatment of aT sEVs. Due to the increased secretion of sEV PD-L1, TSG101, which regulates cargo sorting, is also secreted with sEVs. Therefore, we can detect an increase in TSG101 levels in the sEVs secreted by tumour cells stimulated with aT-sEVs. This tendency is consistent with the changes observed in the protein level of HRS.

18. Supplemental figure 11B-Very nice western blots.

Response: We thank the reviewer again for the positive comments.

19. Supplementary figure 12 E&F- what are the control tEV? Are they from non-activated T cells?

Response: The control tEVs (cT-sEVs) used in **Supplementary Figure 12 E&F** (the **revised Supplementary Fig. 6e**) are sEVs derived from non-activated T cells. By comparing the responses induced by aT-sEVs to those of cT-sEVs, we were able to assess the specific effects of sEV PD-1/CD80 on the regulation of immunogenicity in tumour cells. To enhance clarity, the information has now been included in the **Methods** section (**Page 31, Line 799**).

20. Supplementary Figure 12G- do you have decreased phagocytosis of tumour cells with the increase in CD47 expression after sEV treatment?

Response: In the original manuscript, we did not present any data demonstrating an increase in CD47 expression in tumour cells. In **Supplementary Figure 12G** (the **Supplementary revised Fig. 6f**), we evaluated the impact of sEV treatment on the accumulation of CD47 in sEVs derived from tumour cells. We hypothesized that the elevated expression of CD47 in tumour cell-derived sEVs might lead to their reduced clearance since extensive research have demonstrated the role of CD47 as a “don’t eat me” signalling in anti-tumour immunity.

To address whether the secretion of sEV CD47 alter the phagocytosis, the expression of CD47 on tumour cell treated with aT-sEVs was evaluated. As shown in **Figure R18a, b**, we found that treatment with aT-sEVs significantly increase membrane expression of CD47 in tumour cells, suggesting a potential mechanism of resistance for tumour cells against clearance by phagocytic cells. To confirm this, we initially stimulated tumour cells with or without aT-sEVs, and subsequently coculture them with M1 macrophages in which differentiation had been pre-induced *in vitro*. After 6 h, macrophages and tumour cells were labelled with cell marker respectively, and undergone immunofluorescence staining. The overlap of the two fluorescent signals indicated that the tumor cells were engulfed by the macrophages. As we expected, treatment of aT-

sEVs led to a reduction of tumour cell phagocytosis by macrophages. (**Figure R18c**). This partially validates our hypothesis, but the underlying mechanistic details still need further investigation.

Figure R18. Detection of CD47 in tumour cells treated with aT-sEVs. a, Heat map showed the relative level of CD47 in tumour cell with or without aT-sEVs treatment. b, Representative flow cytometric histograms and quantification analysis of membrane CD47 expression level in tumour cells with or without aT-sEV treatment. c, Representative immunofluorescence staining of EpCAM⁺ (green) tumour cells and CD68⁺ (red) human macrophages (left). Corresponding relative phagocytosis, calculated as the number of macrophages with phagocytized tumour cell divided by total macrophages per five high-power field × 100%. Error bars represented mean ± s.d., statistical analyses were performed using two-sided unpaired t-test (b, c).

21. Supplemental Figure 16E- PDL-1- patients have 0% ORR score whereas PD-L1+ patients have 100% response? Not sure I understand these results.

Response: We apologize for the unclear labelling and inadequate description of **Supplementary Figure 16E** (the revised **Fig. 7e**). In fact, this figure showed the subgroup of patients from **Supplementary Fig. 16c** (the revised **Fig. 7c**)

who have high levels of sEV PD-1/CD80 but low levels of sEV PD-L1, rather than all patients. Because for this subgroup of patients, responders could not be effectively distinguished from non-responders to immunotherapy through sEV PD-1/CD80 and sEV PD-L1. To solve the problem, we attempted to incorporate additional indicators. By evaluating the importance scores of several common indicators in the **revised Fig. 7d**, we found that TPS of the tumour cell PD-L1 might help further distinguish the aforementioned subgroup of patients. Thus, based on the random forest predictive model, we performed TPS evaluation on the subgroup of patients with high sEV PD-1/CD80 but low sEV PD-L1 levels (the **revised Fig. 7e**). We found that among the six patients in this subgroup, those who were TPS-negative had a lower overall survival rate (0%), while TPS-positive patients had a higher overall survival rate (100%). This suggests that combining TPS evaluation with respect to patients with high sEV PD-1/CD80 but low sEV PD-L1 levels is a feasible predictive strategy. Therefore, we depicted this strategy as a predictive flowchart in the **revised Fig. 7f**. To enhance clarity, we have now provided detailed information in the corresponding figure legends.

22. In line 873, you claim that less than 30% of sEVs were antibody-bound in HNSCC patients treated with blocking antibody. If the sEVs do not readily bind antibody, how can you explain the effect seen in the neutralizing experiments performed throughout the paper where anti-PD-1/anti-CD80 antibody was added to the sEV prior to culture or in vivo injection?

Response: We appreciate the reviewer's question and recognize the potential discrepancy between the observed effect in the neutralizing experiments *in vitro* and the low antibody binding to sEVs *in vivo*. While it is true that we observed a relatively low percentage of antibody-bound sEVs in HNSCC patients, it is important to note that the neutralizing experiments involved adding the anti-PD-1/anti-CD80 antibody directly to the sEVs prior to culture or in vivo injection. This approach allows for a localized and concentrated exposure of the

antibodies to the sEVs, enhancing their binding. Accordingly, a great many of studies have demonstrated the effectiveness and feasibility of blocking cell or sEV surface proteins using corresponding blocking antibodies including PD-1 and CD80^{5,15,17,24-27}. In the present study, the application of blocking antibody attenuated the effects induced by aT-sEVs also suggested the effectiveness of antibody neutralizing.

However, it is important to consider that the binding kinetics and accessibility of antibodies to circulating sEVs in patients may differ from *in vitro* or controlled experimental settings. Factors such as the dynamic nature of sEVs in circulation, the metabolism-induced clearance of circulating antibodies, the low exposure of specific subpopulations of sEVs and antibodies in circulation, and other potential factors in the complex biological milieu can influence the interaction between antibodies and sEVs. Therefore, while the efficacy of antibody-mediated neutralization has been demonstrated in various experimental contexts, further investigation is needed to fully understand the binding dynamics and accessibility of antibodies to sEVs in the specific context of patients with HNSCC.

23. How much sEV were added for each of these *in vitro* experiments? Was it the same 100ug as the *in vivo* studies?

Response: The dose of 100 µg sEVs was used for *in vivo* studies because the dose was equivalent to approximately 30% of the physiological level of circulating exosomes and comparable to those from a palpable tumour in mice according to our previous research⁵.

As for sEVs for *in vitro* treatment, 25 µg/ml of aT-sEVs (carrying surface PD-1 or CD80 at a level of approximately 8 to 10 pg per µg of sEVs as determined by ELISA) were used as the circulating sEV PD-1 or CD80 levels in HNSCC patients is around 200 pg/ml (**original Supplementary Fig. 2b, c**).

We have now added this information in the **Methods** section (**Page 31, Line 793; Page 33, Line 862**).

Reviewer #3

Comments to the Authors

In the manuscript titled Extracellular vesicle PD-1/CD80 from immunocytes induce cold tumours featured with enhanced adaptive immunosuppression, the author unveiled the mechanisms of sEV PD-1/CD80 via an adaptive loop to suppress the anti-tumour immunity. Also, the author proposed a novel strategy to predict the immunotherapy response by synchronously analysing the multiple checkpoints on circulating sEVs. This study provided some interesting findings and are valuable for further evolution of the immunotherapeutic approaches. However, considering the following problems, MINOR revision has to be done before the manuscript could be accepted for publication.

Response: We appreciate the reviewer's positive comments regarding the novelty and value of our study, specifically highlighting the mechanisms of sEV PD-1/CD80 in suppressing anti-tumour immunity and the proposed strategy for predicting immunotherapy response. We are pleased that our findings have generated interest and have the potential to contribute to the evolution of immunotherapeutic approaches. We have carefully revised the manuscript to address each of the identified issues and provided a point-by-point response correspondingly.

Minor

1. In figure 1a, one of the EVs you showed is more fusiform than spherical. Please explain the reason or provide a more typical photo.

Response: We appreciate the reviewer's kind reminder regarding the morphology of sEVs shown in Fig. 1a. We have updated the representative image in the **revised Fig. 1a** to better exemplify the typical spherical morphologies of sEVs.

2. The author needs to make an accurate definition of EV PD-1/CD80. Is EV PD-1/CD80 a mixture of EV PD-1 and EV CD80, or an EV containing both PD-

1 and CD80, or a mixture of the three? This should also be reflected in your Graphical Abstract and figure 5h.

Response: Thanks for the reviewer's comments on the composition of circulating sEVs carrying PD-1 and CD80. We apologize for any confusion that may have arisen from the lack of clarity in the description of the original **Supplemental Fig. 2b-d**. Firstly, we examined the expression levels of PD-1 and CD80 on circulating sEVs in patients with HNSCC, colorectal cancer, lung cancer, and breast cancer using ELISA. We observed a common occurrence of simultaneous upregulation of PD-1 and CD80 on circulating sEVs from patients with different types of tumours. Furthermore, using nanoparticle flow cytometry, we simultaneously labelled the surface PD-1 and CD80 on circulating sEVs and identified three subpopulations: PD-1⁺ sEVs, CD80⁺ sEVs, and PD-1⁺CD80⁺ sEVs. Interestingly, we found that more than half of these sEVs carried both PD-1 and CD80 molecules (**revised Fig. 1j**). Considering their similar functional roles in mediating downstream signalling upon binding to tumour cell PD-L1, we collectively refer to them as "sEV PD-1/CD80". We have carefully revised the manuscript to accurately represent this information and have made the necessary adjustments to ensure that it is correctly reflected in the **revised Supplemental Fig. 10** and **Figure 5h**.

3. In the introduction, you should summarize the previous researches (reference 28-30) and point out your own innovation to highlight the advantages of your study.

Response: We thank the reviewer for valuable suggestion regarding the introduction of our manuscript. We had discussed these previous studies in the original **Discussion** section. Briefly, the presence of PD-1 and CD80 in sEVs in HNSCC patients was initially identified in 2017; however, their potential involvement in immune regulation remained unclear. The significant roles played by sEV PD-1 in orchestrating the immune system of cancer patients were not emphasized until recent studies brought attention to its controversial

association with immunotherapy. Without elucidating the underlying mechanism, one study proposed the clearance of harmful PD-L1⁺ sEVs, while another claimed the consumption of therapeutic anti-PD-1 blocking antibodies induced by sEV PD-1. Importantly, the comprehensive evaluation of the interaction between these sEVs and cells in the tumour microenvironment has been lacking. In our study, we provide evidence demonstrating that sEV PD-1, along with its counterpart CD80, predominantly secreted by activated immunocytes, downregulates membrane PD-L1 expression on tumour cells and promotes the release of sEV PD-L1 into circulation through an ESCRT-dependent mechanism. This process "cools down" the tumour microenvironment, resulting in systemic immunosuppression.

In the revised manuscript, following the reviewer's suggestion, the above contents have been added to the **Introduction** section (**Page 5, Line 98-104**) to provide a comprehensive background and emphasized the innovative aspects for our study. We believe that these revisions have strengthened the introduction and provided a clearer context for the significance of our work.

4. Some sentences are too wordy and obscure (e.g. line 545-550). The current manuscript can be polished by a native English speaker or a professional language editing service.

Response: We feel sorry for the wordy and obscure description in the current manuscript. To improve, we have carefully edited our manuscript. Additionally, the manuscript has undergone professional language editing provided by Springer Nature Author Services (**Figure R19**). All the changes have been highlighted in the resubmitted manuscript. Thanks again for the reviewer's suggestion.

[REDACTED]

Figure R19. The editing certificate provided by SPRINGER NATURE Author Services.

5. The discussion section contains too many repetitions of the results. Sentences need to be more concise. In addition, there was no mention of the limitation of the study. This needs to be presented in the discussion.

Response: We appreciate the reviewer's feedback regarding the discussion section. In response, we have made efforts to simplify the contents based on the reviewer's suggestion in the revised manuscript. Additionally, we have included a section in the discussion that addresses the limitations of our study (**Page 23, Line 591**). Briefly, it is important to acknowledge that we haven't directly compared the functional distinctions among single-positive PD-1 or CD80 sEVs and double-positive PD-1/CD80 sEVs in patients, although we've identified similar functions for sEV PD-1 and CD80. Thank you for bringing this to our attention, and we believe that these revisions will enhance the clarity and completeness of the discussion section.

References

1. Gardiner, C. et al. Techniques used for the isolation and characterization of extracellular vesicles: results of a worldwide survey. *J. Extracell. Vesicles* **5**, 32945 (2016).
2. Hoshino, A. et al. Extracellular Vesicle and Particle Biomarkers Define Multiple Human Cancers. *Cell* **182**, 1044-1061.e1018 (2020).
3. Wang, G. et al. Tumour extracellular vesicles and particles induce liver metabolic dysfunction. *Nature* **618**, 374-382 (2023).
4. Théry, C. et al. Minimal information for studies of extracellular vesicles 2018 (MISEV2018): a position statement of the International Society for Extracellular Vesicles and update of the MISEV2014 guidelines. *J. Extracell. Vesicles* **7**, 1535750 (2018).
5. Chen, G. et al. Exosomal PD-L1 contributes to immunosuppression and is associated with anti-PD-1 response. *Nature* **560**, 382-386 (2018).
6. Yang, Y. et al. Exosomal PD-L1 harbors active defense function to suppress T cell killing of breast cancer cells and promote tumor growth. *Cell Res.* **28**, 862-864 (2018).
7. Chen, J. et al. PDL1-positive exosomes suppress antitumor immunity by inducing tumor-specific CD8 T cell exhaustion during metastasis. *Cancer Sci.* **112**, 3437-3454 (2021).
8. Qiu, Y. et al. Activated T cell-derived exosomal PD-1 attenuates PD-L1-induced immune dysfunction in triple-negative breast cancer. *Oncogene* **40**, 4992-5001 (2021).
9. Serratì, S. et al. Circulating extracellular vesicles expressing PD1 and PD-L1 predict response and mediate resistance to checkpoint inhibitors immunotherapy in metastatic melanoma. *Mol. Cancer* **21**, 20 (2022).
10. Ludwig, S. et al. Suppression of Lymphocyte Functions by Plasma Exosomes Correlates with Disease Activity in Patients with Head and Neck Cancer. *Clin. Cancer Res.* **23**, 4843-4854 (2017).
11. Abusamra, A. et al. Tumor exosomes expressing Fas ligand mediate CD8+ T-cell apoptosis. *Blood Cell. Mol. Dis* **35**, 169-173 (2005).
12. Wan, C. et al. Exosome-related multi-pass transmembrane protein TSAP6 is a target of rhomboid protease RHBDD1-induced proteolysis. *PloS one* **7**, e37452 (2012).
13. Ludwig, S. et al. Molecular and Functional Profiles of Exosomes From HPV(+) and HPV(-) Head and Neck Cancer Cell Lines. *Front. Oncol.* **8**, 445 (2018).
14. Chen, J. et al. GOLM1 exacerbates CD8 T cell suppression in hepatocellular carcinoma by promoting exosomal PD-L1 transport into tumor-associated macrophages. *Signal Transduct. Target Ther.* **6**, 397 (2021).
15. Li, M. et al. WJMSC-derived small extracellular vesicle enhance T cell suppression through PD-L1. *J. Extracell. Vesicles* **10**, e12067 (2021).

16. Caruntu, A. et al. Persistent Changes of Peripheral Blood Lymphocyte Subsets in Patients with Oral Squamous Cell Carcinoma. *Healthcare (Basel, Switzerland)* **10** (2022).
17. Zhang, W. et al. ICAM-1-mediated adhesion is a prerequisite for exosome-induced T cell suppression. *Dev. Cell* **57**, 329-343.e327 (2022).
18. Xie, Q. et al. Exosome-Mediated Immunosuppression in Tumor Microenvironments. *Cells* **11** (2022).
19. Lin, W. et al. Molecular actions of exosomes and their theragnostics in colorectal cancer: current findings and limitations. *Cell. oncol.* **45**, 1043-1052 (2022).
20. Huo, H. et al. Brain endothelial cells-derived extracellular vesicles overexpressing ECRG4 inhibit glioma proliferation through suppressing inflammation and angiogenesis. *J. Tissue Eng. Regen.* **15**, 1162-1171 (2021).
21. Ruan, Z. et al. P2RX7 inhibitor suppresses exosome secretion and disease phenotype in P301S tau transgenic mice. *Mol. Neurodegener.* **15**, 47 (2020).
22. Bijnsdorp, I. et al. Feasibility of urinary extracellular vesicle proteome profiling using a robust and simple, clinically applicable isolation method. *J. Extracell. Vesicles* **6**, 1313091 (2017).
23. Tauro, B. et al. Two distinct populations of exosomes are released from LIM1863 colon carcinoma cell-derived organoids. *Mol. Cell. proteomics P* **12**, 587-598 (2013).
24. Kojima, Y. et al. CD47-blocking antibodies restore phagocytosis and prevent atherosclerosis. *Nature* **536**, 86-90 (2016).
25. Wang, T. et al. Blocking PD-L1-PD-1 improves senescence surveillance and ageing phenotypes. *Nature* **611**, 358-364 (2022).
26. Sugiura, D. et al. PD-1 agonism by anti-CD80 inhibits T cell activation and alleviates autoimmunity. *Nat. Immunol.* **23**, 399-410 (2022).
27. Maurer, M. et al. The engineered CD80 variant fusion therapeutic davoceticept combines checkpoint antagonism with conditional CD28 costimulation for anti-tumor immunity. *Nat. Commun.* **13**, 1790 (2022).

REVIEWER COMMENTS

Reviewer #1 (Remarks to the Author):

The revised and re-organized manuscript is greatly improved. It conveys an interesting and intriguing idea that PD-1/CD80+ sEV produced by immunocytes regulate PD-L1 transport from the surface of tumor cells to tumor-derived PD-L1+ EVs thus enhancing immune suppression and the tumor resistance to immune therapies. Additionally, PD-1/CD80+ IsEV alter adhesion and antigen-presenting molecules in the tumor, impair immune cell infiltration and convert the tumor to a "cold" phenotype. The revised text documents the molecular events driven by PD-1/CD80+ IsEV in a nicely organized fashion and demonstrates the clinical significance of PD-1/CD80 IsEV in HNSCC patients undergoing immunotherapy, where the frequency of circulating sEV carrying PD1/CD80 and PD-L1 distinguishes responders from nonresponders to PD-1 therapy. The reorganized text logically develops this scenario and provides convincing evidence for molecular mechanism underlying its various stages as they evolve in vitro and in vivo.

The authors also provide thoughtful answers to reviewers' comments, explaining in depth the experimental approaches often by presenting additional data and explaining the experimental rationale. The revised manuscript in part reflects these answers, which greatly contributes to the overall data presentation and strengthens the content of the narrative. While the authors have done a great job addressing the reviewers' concerns, there still remain a few issues to clarify. Much of the data evolve around the frequency and intensity of expression on IsEV of PD-1, CD80 or PD-L1 and on the confirmation of immunocyte origin of these vesicles. These results are based on single vesicle nanoflow cytometry. While SFigure 1 illustrates the gating strategy used and application of beads to establish the vesicle size, the description of nanoparticle flow cytometry in Methods is inadequate. In fact, it is better explained in the answer to review than in Methods. This technology is still in development, and no single instrument or procedure has been validated so far. Therefore, it is necessary to describe in a great detail the gating, sizing and staining of vesicles for nanoflow cytometry and document reproducibility and reliability before using it for routine vesicle phenotyping. While the authors appear to be comfortable and confident in the use of single vesicle flow cytometry, it would be appropriate to tell the reader exactly how they do it to allow others to reproduce the authors' results.

The second issue has to do with extensively presented comparison of effects on the tumor exerted by IFN- γ versus PD-1/CD80+ sEV derived from aTcells. These two different mechanisms regulating the PD-L1 level on tumor cells are nicely documented. While the authors suggest that aT-sEV have dominant effects, it seems more likely that the mechanisms subserve an activated T cell in balancing its immunoregulatory effects depending on the context in the microenvironment. In this way, a T cell has an option to regulate immune suppression the tumor delivers and immunotherapy induced rejuvenation of T cells would be likely to change this balance. Is it possible that sEV released by aT cells carry not only PD-1/CD80 but also IFV- γ to the tumor??-

The other remaining issue is the quality of writing, which despite extensive revisions and help from Author Services by Springer remains inadequate. For example, it would be better to use "PD-1/CD80+ IsEVs" than "IsEVs PD-1/CD80" throughout the manuscript. The title should read "PD-1/CD80+ small extracellular vesicles from immunocytes induce cold tumors with enhanced adaptive iline mmunosuppression." The abstract needs extensive revisions as follows:

Line 35 "attributed not only to crosstalk...."

Lines 36,37.... " but also to an interaction pattern of...."

Line 38..... "We demonstrate that....."

Line 39..... "sEVs (IsEVs) induces an....."

Line 40 "redistribution of PD-L1 in tumor cells, resulting in decreased"

Line 41..... "and increased secretion of PD-L1+ sEVs into the circulation."

Line 42..... "This contributes to In addition,....."

Line 43..... "PD-1/CD80+ IsEVs downregulate....."

Line 45..... "cell infiltration, thereby converting tumors to an....."

Line 49..... "our study shows that"

Line 50..... "carried by sEVs function via....."

Line 51..... "providing a rationale for"

Similar changes are needed throughout the manuscript and the text requires extensive revisions to make English acceptable, especially in the areas outlined in red. As is, the manuscript still suffers from poor phrases and inadequate use of English. Extensive editorial help is necessary.

Discussion is still too long and often repeats results, rather than discussing the impact and importance of the presented data. Please abbreviate and focus on telling the reader why your data are impactful and contribute to the better understanding of the role immunocyte-derived sEV play in HNSCC. Specifically, please revise the final paragraph of Discussion: as is, it leaves one with an impression that this study has many limitations, rather than advantages. The data presentation is very well done and figure legends amply describe the illustrated examples.

Reviewer #3 (Remarks to the Author):

The authors have addressed all my comments. I'd like to support its publication.

Reviewer #4 (Remarks to the Author):

1. R#2 had two major questions. Unfortunately, the authors did not address them well. For Q1, the reviewer asked the important question concerning whether the sEVs that express PD-L1 were the same ones that express PD-1 and CD80. To address this question, the authors performed triple labeling for these proteins, and stated that only 2.5% of the sEVs were triple positive. It is well known in nanoparticle flow cytometry that exosomes cannot be stained for 3 markers due to their small size (~100nm). Physical hinderance of antibodies on the surface of the same exosome would lead to gross underestimation. In fact, even the values for double staining are questionable for this reason. This is particularly troublesome considering that CD80, PD-L1 and PD-1 can directly interact with each other, which makes the hinderance highly likely. For Q2, the reviewer asked about the relevant abundance of PD-L1 sEVs vs PD-1/CD80 sEVs. Again, problems with multiple staining of sEVs raise the same concerns. More importantly, different antibodies have different affinities and avidities for different antigens. Therefore, the authors cannot draw concrete conclusions regarding the relative amounts of different populations of sEVs.
2. There are issues with the authors' response to some other questions raised by R#2 as well, I will not list each of them here. While some of them are of minor concern, a major issue was the amount of sEV (100 ug) used in the in vivo experiment (Q23). The authors stated that this amount accounts for "approximately 30% of the physiological levels of circulating exosomes". This is impossible as exosomes derived from platelets and many other cell types are dominant source of circulating exosomes. It is hard to believe that exosomes specifically carrying PD-1 and/or CD80 from activated T cells constitute such a major proportion--this is huge! This is an important concern because the level of exosome infusion really needs to mirror the physiology.
3. The authors used Relative Fluorescence Intensity (RFI) as their readout. RFI can vary significantly from experiment to experiment, and for different proteins. It is thus not a reliable readout for quantitative comparisons that are necessary for this study. The reviewers asked for more detailed description. Unfortunately, the authors just stated "Concentrations were determined by accounting for flow rate, measurement time, and sample dilution to correct the number of detected". This is a list of known variables in experimentation, but not actual numbers for their specific experiments.

Point-by-point response

Reviewer #1

Comments to the Authors

The revised and re-organized manuscript is greatly improved. It conveys an interesting and intriguing idea that PD-1/CD80⁺ sEV produced by immunocytes regulate PD-L1 transport from the surface of tumor cells to tumor-derived PD-L1⁺ EVs thus enhancing immune suppression and the tumor resistance to immune therapies. Additionally, PD-1/CD80⁺ IsEV alter adhesion and antigen-presenting molecules in the tumor, impair immune cell infiltration and convert the tumor to a “cold” phenotype. The revised text documents the molecular events driven by PD-1/CD80⁺ IsEV in a nicely organized fashion and demonstrates the clinical significance of PD-1/CD80 IsEV in HNSCC patients undergoing immunotherapy, where the frequency of circulating sEV carrying PD1/CD80 and PD-L1 distinguishes responders from nonresponders to PD-1 therapy. The reorganized text logically develops this scenario and provides convincing evidence for molecular mechanism underlying its various stages as they evolve in vitro and in vivo. The authors also provide thoughtful answers to reviewers' comments, explaining in depth the experimental approaches often by presenting additional data and explaining the experimental rationale. The revised manuscript in part reflects these answers, which greatly contributes to the overall data presentation and strengthens the content of the narrative.

Response: We thank the reviewer for the positive feedback on our revised manuscript and response letter.

1. While the authors have done a great job addressing the reviewers' concerns, there still remain a few issues to clarify. Much of the data evolve around the frequency and intensity of expression on IsEV of PD-1, CD80 or PD-L1 and on the confirmation of immunocyte origin of these vesicles. These results are based on single vesicle nanoflow cytometry. While SFigure 1 illustrates the gating strategy used and application of beads to establish the vesicle size, the

description of nanoparticle flow cytometry in Methods is inadequate. In fact, it is better explained in the answer to review than in Methods. This technology is still in development, and no single instrument or procedure has been validated so far. Therefore, it is necessary to describe in a great detail the gating, sizing and staining of vesicles for nanoflow cytometry and document reproducibility and reliability before using it for routine vesicle phenotyping. While the authors appear to be comfortable and confident in the use of single vesicle flow cytometry, it would be appropriate to tell the reader exactly how they do it to allow others to reproduce the authors' results.

Response: We are sorry for inadequately description of nanoparticle flow cytometry in **Methods**. To improve, detailed information on gating, sizing and staining has now been given in the updated **Methods**. Briefly, 110 nm orange FluoSpheres (standard beads with fluorescence provided by manufacturer) of known particle concentration (5000 particles/ μ l) were applied to calibrate the sample flow rate. Then, non-fluorescent standard beads with sizes of 180 nm, 240 nm, 300 nm, 590 nm, 880 nm and 1300 nm (provided by manufacturer) were employed for size gating, ensuring an appropriate working range for subsequent detection. The following reagents were used for sEV labelling in nanoparticle flow cytometry analysis: antibody against CD45, CD144, EpCAM, CD4, CD8, CD11c, CD19, CD86, PD-1, CD80, PD-L1, TIM-3, LAG-3, CTLA-4, ICAM-1, CD47, EGFR, IgG1 or IgG2b was added. Each sample (0.5-1 μ g, about $1 - 2 \times 10^7$ particles) with 0.25 μ g antibody were incubated at room temperature for 30 min and then washed twice with 1 ml PBS at 120,000g for 70 min at 4 °C (Beckman Coulter MAX-XP centrifuge). The pellet was resuspended in 300 μ L PBS for analysis under identical detection condition for standard beads (A50 micro plus Flow Cytometry, Apogee). All samples were measured for 2 min at a flow rate of 1.5 μ l/min using SSC triggering (405-nm laser, 70 mW). The detection threshold was set at 20 a.u. (small angle light scatter [SALS]) and 25 a.u. (large angle light scatter [LALS]) to eliminate optical and electronic background noise without losing particles of interest. Positive

events were defined as those exhibiting a fluorescent signal within the designated gate. Concentrations were determined by accounting for consistent flow rate, measurement time, and sample dilution to correct the number of detected for better reproducibility and reliability.

It needs to be acknowledged that there are indeed technical limitations in detecting subpopulation of sEVs. Recently, the technique with nanoparticle flow cytometry has been employed in numerous high-level studies and publications for the analysis of specific sEV subpopulations¹⁻⁵. More importantly, in this study, we compared the relative levels of PD-1/CD80 sEVs in different samples under the same detection conditions to support our research conclusions. This does not compromise the scientific rigor of our study. The aim of our present study is to propose a novel mechanism of tumour immune suppression, not to introduce an accurate quantitative technique. On the other hand, besides nanoparticle flow cytometry, we actually employed with other conventional detection techniques, including WB and ELISA. However, these results were excluded from the manuscript during the initial round of revisions due to length limitations. If required, we are prepared to furnish these additional findings.

2. The second issue has to do with extensively presented comparison of effects on the tumor exerted by IFN- γ versus PD-1/CD80⁺ sEV derived from aTcells. These two different mechanisms regulating the PD-L1 level on tumor cells are nicely documented. While the authors suggest that aT-sEV have dominant effects, it seems more likely that the mechanisms subserve an activated T cell in balancing its immunoregulatory effects depending on the context in the microenvironment. In this way, a T cell has an option to regulate immune suppression the tumor delivers and immunotherapy induced rejuvenation of T cells would be likely to change this balance. Is it possible that sEV released by aT cells carry not only PD-1/CD80 but also IFN- γ to the tumor??

Response: Thanks for the Reviewer's valuable comments and suggestions. As the Reviewer pointed out that sEVs released by activated T cells might also

carry IFN- γ , we therefore tested this with WB (**Figure R1a**) and ELISA (**Figure R1b**) but found that the level of IFN- γ in aT-sEVs was barely detectable. However, consistent with previous study⁶⁻⁹, we found that IFN- γ primarily existed in the form of soluble protein in the culture supernatant of activated T cells (**Figure R1b**).

Figure R1. Detection of IFN- γ in whole cell lysates, supernatants and sEVs from T cells. (a) Western blot analysis was performed to evaluate the level of IFN- γ in WCL and sEV from T cells without or with activation. (b) ELISA analysis of IFN- γ in Sup and sEV from T cells without or with PHA stimulation. WCL, whole cell lysate; Sup, supernatant; sEV, small extracellular vesicle.

3. The other remaining issue is the quality of writing, which despite extensive revisions and help from Author Services by Springer remains inadequate. For example, it would be better to use “PD-1/CD80+ IsEVs” than “IsEVs PD-1/CD80” throughout the manuscript. The title should read “PD-1/CD80+ small extracellular vesicles from immunocytes induce cold tumors with enhanced adaptive immunosuppression.” The abstract needs extensive revisions as follows:

Line 35”attributed not only to crosstalk.....”

Lines 36,37....” but also to an interaction pattern of.....”

Line 38.....”We demonstrate that.....”

Line 39.....”.sEVs (IsEVs) induces an.....”

Line 40 “redistribution of PD-L1 in tumor cells, resulting in decreased”

Line 41.....”and increased secretion of PD-L1+ sEVs into the circulation.”

Line 42....."This contributes to In addition,.....".

Line 43....."PD-1/CD80+ sEVs downregulate....."

Line 45....."cell infiltration, thereby converting tumors to an....."

Line 49....."our study shows that"

Line 50....."carried by sEVs function via....."

Line 51....."providing a rationale for"

Similar changes are needed throughout the manuscript and the text requires extensive revisions to make English acceptable, especially in the areas outlined in red. As is, the manuscript still suffers from poor phrases and inadequate use of English. Extensive editorial help is necessary.

Response: Thanks for the Reviewer's attention to the English usage in this manuscript. The Reviewer has provided very detailed suggestions for the language, which are highly valuable and worthy of incorporation into the manuscript editing. Therefore, we have made corresponding modifications in the manuscript based on the Reviewer's suggestions. On the other hand, we also trust that the professional editing service provided by Author Services by Springer adheres to academic standards for British English. Thus, we have also preserved certain writing aspects brought by variations in language usage habits. As for the description of sEVs terminology, "PD-1/CD80+ sEVs" emphasizes the vesicles themselves, while "sEV PD-1/CD80" focuses on the surface molecules carried by the vesicles, making a subtle distinction. Accordingly, we reexamined the usage of CD80 and PD-1 in the manuscript, still striving to employ the expression "PD-1/CD80+ sEVs" whenever possible to reduce ambiguity and enhance readability. We would like to express our gratitude once again for the insightful suggestions provided by the Reviewer.

4. Discussion is still too long and often repeats results, rather than discussing the impact and importance of the presented data. Please abbreviate and focus on telling the reader why your data are impactful and contribute to the better understanding of the role immunocyte-derived sEV play in HNSCC. Specifically,

please revise the final paragraph of Discussion: as is, it leaves one with an impression that this study has many limitations, rather than advantages.

Response: We appreciate the Reviewer's constructive feedback and suggestions for improving the manuscript. In the revised version, we have streamlined the **Discussion** to ensure a more concise and impactful narrative that highlights the significance of our findings in elucidating the role of sEV PD-1/CD80 in HNSCC. We would also reframe the closing statements to underscore the advantages and relevance of our study in the field.

5. The data presentation is very well done and figure legends amply describe the illustrated examples.

Response: Thank the Reviewer for the positive feedback regarding the data presentation and figure legends in our manuscript.

Reviewer #3

Comments to the Authors

The authors have addressed all my comments. I'd like to support its publication.

Response: We appreciate the positive comments from the Reviewer.

Reviewer #4

Comments to the Authors

1. R#2 had two major questions. Unfortunately, the authors did not address them well.

For Q1, the reviewer asked the important question concerning whether the sEVs that express PD-L1 were the same ones that express PD-1 and CD80. To address this question, the authors performed triple labelling for these proteins, and stated that only 2.5% of the sEVs were triple positive. It is well known in nanoparticle flow cytometry that exosomes cannot be stained for 3 markers due to their small size (~100nm). Physical hinderance of antibodies on the surface of the same exosome would lead to gross underestimation. In fact, even the values for double staining are questionable for this reason. This is particularly troublesome considering that CD80, PD-L1 and PD-1 can directly interact with each other, which makes the hinderance highly likely.

For Q2, the reviewer asked about the relevant abundance of PD-L1 sEVs vs PD-1/CD80 sEVs. Again, problems with multiple staining of sEVs raise the same concerns. More importantly, different antibodies have different affinities and avidities for different antigens. Therefore, the authors cannot draw concrete conclusions regarding the relative amounts of different populations of sEVs.

Response: We apologize for any confusion brought to **Reviewer 4**. However, in **Major Q1 and Q2**, **Reviewer 2** simply requested clarification on whether PD-L1/PD-1/CD80 coexist in sEVs and if PD-L1 sEVs are the predominant subset in the circulation of HNSCC patients, rather than asking for specific proportions of each sEV subpopulation with comprehensive supporting evidence.

In fact, it can be clarified that circulating sEVs triple positive for PD-L1, PD-1, and CD80 do exist, albeit in low proportions. As shown in **Supplementary Fig. S2**, we found that sEVs from activated T cells prominently contained CD80 and PD-1, while these molecules were scarcely detected in sEVs from HNSCC cells. These results were based on single antibody labelling nanoparticle flow cytometry, being standardized by Quantum™ MESF (Molecules of Equivalent

Soluble Fluorochrome) microsphere kit. The results were further confirmed with WB analysis that originally removed in last round of revision (**Figure R2**). Therefore, we draw a clear conclusion that sEVs derived from HNSCC tumour cells express PD-L1 but show minimal expression of PD-1 or CD80. As for sEVs derived from immunocytes, it is reasonable to infer that sEVs derived from immune cells carry PD-L1/CD80/PD-1, considering that sEVs inherit proteins from the parental cells and immune cells simultaneously express all three proteins PD-L1/CD80/PD-1. This was further supported by results from double antibodies labelling-based nanoparticle flow cytometry (**Figure R3**), where the ratio of PD-1⁺, CD80⁺ or PD-L1⁺ sEVs in immunocyte-derived sEVs was fairly high, suggesting a potential overlap of the three subpopulation.

Apart from abovementioned indirect evidence, to address the concerns of **Reviewer 4** more comprehensively, we here conducted additional experiments. Using magnetic beads, we performed negative selection to remove sEVs positive for one specific membrane protein from circulating sEVs, followed by the analysis of the remaining sEVs' composition. As shown in **Figure R4a**, the depletion of PD-L1⁺ sEVs in circulating sEVs dramatically decreased the number of PD-L1⁺ sEVs in the tested blood sample, as well as the number of PD-1-CD80 double-positive sEVs at that time. These results indicated a coexistence of PD-L1, CD80 and PD-1 on sEVs. With WB analysis, we also discovered that the depletion of either PD-L1⁺ resulted in simultaneous downregulation of PD-1 and CD80 in circulating sEVs from blood samples (**Figure R4b**), further supporting the abovementioned results. As shown in **Figure R4c**, after removing PD-1⁺ sEVs using antibody-specific magnetic beads, about 11% of sEVs remained CD80 positive, as observed in single antibody labelling nanoparticle flow cytometry. Likewise, after depleting CD80⁺ sEVs, around 10% of sEVs were PD-1 positive. These findings closely aligned with the results obtained from double labelling-based nanoflow cytometry presented in **Figure 1j**, reinforcing the reliability of our outcomes. Meanwhile, these results were consistent with our findings in **revised Supplementary Fig.**

1b, addressing the **Major Q2** from **Reviewer 2** that it was not simply that PD-L1⁺ sEVs are the most predominant subpopulation. In fact, both PD-L1⁺ and PD-1/CD80⁺ sEVs have a relatively high proportion in circulation.

As for technique issues raised by the Reviewers regarding multiple staining of sEVs, nanoparticle flow cytometry is a relatively recent technique for sEV detection and analysis. We recognize that, like any detection method, it has inherent limitations, including potential constraints associated with the size of sEVs in multicolour labelling. However, in numerous high-level studies^{1,4,10-12}, nanoparticle flow cytometry distinguishes itself as a more reliable and superior technique for detecting sEVs labelled with multiple antibodies compared to traditional methods, such as magnetic bead-based detection. Because magnetic bead-based detection may bring alteration to the physicochemical properties of sEVs, which could worsen interference with the detection outcome. In fact, the concerns raised by **Reviewer 4** are not simply about the limitations of the detection method itself but rather the inherent challenges associated with the binding of multiple antibodies to sEVs. This prompts a broader question: should we consider any method that involves using more than one antibody to label sEVs as problematic? Clearly, in many studies, this approach is widely accepted. As for potential differences in antibody affinity, it is essential to note that multi-antibody labelling for the detection of surface markers on diverse cells through flow cytometry is a common and widely accepted technique. Generally, variations in affinity do not significantly impact the results of flow cytometry. Most importantly, by labelling sEVs with three antibodies, we simultaneously detected PD-1, CD80 and PD-L1 on circulating sEVs via nanoparticle flow cytometry. Even considering physical hinderance and cis interaction, it can be inferred that the actual proportion of triple-positive sEVs might be higher than what we detected with nanoparticle flow cytometry. While acknowledging the technical limitations raised by **Reviewer 4** that may affect the precision of proportion detected, it is sufficient to answer **Reviewer 2**'s question regarding the simultaneous presence of the three proteins in sEVs and their relative

proportion. To further enhance the rigor of the study, we have removed the precise description of the proportion of triple-positive sEVs in the manuscript, stating only that they coexist but the proportion is not relatively high.

Indeed, elucidating the heterogeneity of circulating sEVs is an essential topic in the field of extracellular vesicles. However, it's essential to note that whether sEVs are triple-positive is not the primary focus of our study. In our research, the immunosuppressive function of triple-positive sEVs is similar to that of PD-1/CD80 double-positive sEVs. Here, we are still pleased to provide a more detailed description of the experimental procedures and detection methods for nanoparticle flow cytometry in the **Methods** section, aiming to enhance the reproducibility of the results presented in this study.

Figure R2 Western blot analysis of PD-1, CD80 or PD-L1 in whole cells and sEVs from tumour cells and immunocytes.

(a) Western blot analysis of PD-1, CD80, and PD-L1 in whole cells and sEVs from CAL27 and SCC25 cells. (b) Western blot analysis was performed to evaluate the level of PD-1 and CD80 in WCL and sEV from T cells, dendritic cells, and macrophages without or with activation.

Figure R3 Detection of circulating immunocyte sEVs composition in HNSCC patients.

Flow plots and gating strategy for PD-L1+, PD-1+, and CD80+ subpopulation in CD45+ sEVs (left). Quantification analysis of the data (right). Error bars represented mean \pm s.d., statistical analyses were performed using one-way ANOVA.

Figure R4 Detection of circulating sEV subpopulation composition in HNSCC patients.

(a) Comparison of PD-L1⁺ EV number before and after PD-L1⁺ sEV depletion (left). Comparison of PD-1⁺CD80⁺ sEV number before and after PD-L1⁺ sEV depletion (right). (b) Western blot analysis of PD-1, CD80, and PD-L1 in sEVs without or with PD-L1⁺, CD80⁺, or PD-1⁺ sEV depletion, respectively (left). Quantification analysis of the data (right) (c) Nanoflow cytometry analysis of PD-1 and CD80 on sEVs without or with PD-L1⁺, CD80⁺, or PD-1⁺sEV depletion, respectively (left). Quantification analysis of the data (right). Error bars represented mean \pm s.d., statistical analyses were performed using two-sided paired t-test.

2. There are issues with the authors' response to some other questions raised by R#2 as well, I will not list each of them here. While some of them are of minor concern, a major issue was the amount of sEV (100 μ g) used in the *in vivo* experiment (Q23). The authors stated that this amount accounts for "approximately 30% of the physiological levels of circulating exosomes". This is impossible as exosomes derived from platelets and many other cell types are dominant source of circulating exosomes. It is hard to believe that exosomes specifically carrying PD-1 and/or CD80 from activated T cells constitute such a major proportion--this is huge! This is an important concern because the level of exosome infusion really needs to mirror the physiology.

Response: Apologies for the insufficient clarity in the previous point-by-point response, which may have led to a misunderstanding by the Reviewer. Actually, **Reviewer 2** here was more concerned about the dosage used in the *in vitro* experiments with sEVs rather than the *in vivo* dosage. Thus, in our response to **Reviewer 2 (Q23)**, we primarily addressed the dosage for *in vitro* assays, while the description for *in vivo* experiments was somewhat unclear. Here, we did not imply that PD-1/CD80⁺ sEVs derived from T cells constitute 30% of the total circulating sEVs. As we proved in our manuscript, approximately 80% of PD-1⁺

or CD80⁺ sEVs were CD45 positive, suggesting immunocytes as their main cell source (**Supplementary Fig. 2**). While T cells serve as the predominant source of sEV PD-1/CD80, B cells, macrophages, dendritic cells, and other immune cells also contribute to the production of sEV PD-1/CD80 (data removed in previous revision stage). Here, we chose to use aT-sEV PD-1/CD80 to mimic the total circulating sEV PD-1/CD80 under pathological conditions. According to the previous studies and our own results, the total protein of circulating sEVs in mice is generally around 300-1000 μg ^{13,14}. Therefore, when we mentioned 30%, we were referring to the addition of sEVs, where the protein amount was approximately 30% of the total protein of circulating sEVs. This is also why many studies choose such a dosage for animal experiments¹⁵⁻²². In our setup, we assessed the *in vivo* circulating levels of sEV PD-1 and CD80 with ELISA before proceeding with further evaluations. In mice bearing HNSCC cell lines, the concentration of circulating sEV PD-1 was about 100-200 pg/mL, and for circulating sEV CD80, it was about 100-300 pg/mL (**Figure R5a**). In general, the total circulating volume of mouse body fluids is approximately 2 mL. Therefore, each mouse may carry between 200-600 pg of either CD80 or PD-1 in their circulating sEVs. We also assessed the levels of PD-1 and CD80 on mouse T cell-derived sEVs used in our animal study. As shown in **Fig R5b**, the concentrations of PD-1 and CD80 on sEVs were 2-4 pg/ μg and 4-6 pg/ μg , respectively. Therefore, 100 μg activated mouse T cell-derived sEVs carried approximately 300 pg PD-1 and 500 pg CD80, which was quite comparable to their levels in the circulating sEVs of tumour-bearing mice and mirrored their pathological conditions.

Figure R5 Detection of sEV PD-1 and CD80 concentration from blood of mice or activated mouse T (EL4) cells.

(a) ELISA of PD-1 and CD80 on circulating sEVs from control mice and tumour-bearing mice. (b) ELISA of the PD-1 and CD80 protein content of sEVs per μg from activated mouse T(EL4) cells.

3. The authors used Relative Fluorescence Intensity (RFI) as their readout. RFI can vary significantly from experiment to experiment, and for different proteins. It is thus not a reliable readout for quantitative comparisons that are necessary for this study. The reviewers asked for more detailed description. Unfortunately, the authors just stated “Concentrations were determined by accounting for flow rate, measurement time, and sample dilution to correct the number of detected”. This is a list of known variables in experimentation, but not actual numbers for their specific experiments.

Response: The Reviewer may have overlooked specific information in **Methods**. In fact, in the corresponding section, before the sentence mentioned by the Reviewer, we have provided a detailed description of the parameters used for detection, “... All samples were measured for **2 min** at a flow rate of **1.5 $\mu\text{L}/\text{min}$** using SSC triggering (405-nm laser, 70 mW). The detection threshold was set at 20 a.u. (small angle light scatter [SALS]) and 25 a.u. (large angle light scatter [LALS]) to eliminate optical and electronic background noise without losing particles of interest...”. While we use RFI for relative

quantification, it's essential to note that the detection using nanoparticle flow cytometry in our study was performed under the same conditions mentioned above. Thus, the data are comparable, supporting the conclusions of our study.

To enhance the reliability and reproducibility of the study, we have further improved the description of this method as per the suggestions. Briefly, 110 nm orange FluoSpheres (standard beads with fluorescence provided by manufacturer) of known particle concentration (5000 particles / μ l) were applied to calibrate the sample flow rate. Then, non-fluorescent standard beads with sizes of 180 nm, 240 nm, 300 nm, 590 nm, 880 nm, and 1300 nm (provided by manufacturer) were employed for size gating, ensuring an appropriate working range for subsequent detection. The following reagents were used for sEV labelling in nanoparticle flow cytometry analysis: antibody against CD45, CD144, EpCAM, CD4, CD8, CD11c, CD19, CD86, PD-1, CD80, PD-L1, TIM-3, LAG-3, CTLA-4, ICAM-1, CD47, EGFR, IgG1 or IgG2b was added. Each sample (0.5-1 μ g, about $1 - 2 \times 10^7$ particles) with 0.25 μ g antibody were incubated at room temperature for 30 min and then washed twice with 1 ml PBS at 120,000g for 70 min at 4 °C (Beckman Coulter MAX-XP centrifuge). The pellet was resuspended in 300 μ L PBS for analysis under identical detection condition for standard beads (A50 micro plus Flow Cytometry, Apogee). All samples were measured for 2 min at a flow rate of 1.5 μ l/min using SSC triggering (405-nm laser, 70 mW). The detection threshold was set at 20 a.u. (small angle light scatter [SALS]) and 25 a.u. (large angle light scatter [LALS]) to eliminate optical and electronic background noise without losing particles of interest. Positive events were defined as those exhibiting a fluorescent signal within the designated gate. Concentrations were determined by accounting for consistent flow rate, measurement time, and sample dilution to correct the number of detected for better reproducibility and reliability.

References

1. Huang, L., *et al.* Engineered exosomes as an in situ DC-primed vaccine to boost antitumor immunity in breast cancer. *Molecular cancer* **21**, 45 (2022).
2. Lam, S., *et al.* A multi-omics investigation of the composition and function of extracellular vesicles along the temporal trajectory of COVID-19. *Nature metabolism* **3**, 909-922 (2021).
3. Dabrowska, S., *et al.* Imaging of extracellular vesicles derived from human bone marrow mesenchymal stem cells using fluorescent and magnetic labels. *International journal of nanomedicine* **13**, 1653-1664 (2018).
4. Salmond, N., Khanna, K., Owen, G. & Williams, K. Nanoscale flow cytometry for immunophenotyping and quantitating extracellular vesicles in blood plasma. *Nanoscale* **13**, 2012-2025 (2021).
5. Padda, R., *et al.* Nanoscale flow cytometry to distinguish subpopulations of prostate extracellular vesicles in patient plasma. *The Prostate* **79**, 592-603 (2019).
6. Choi, J., *et al.* Combination checkpoint therapy with anti-PD-1 and anti-BTLA results in a synergistic therapeutic effect against murine glioblastoma. *Oncoimmunology* **10**, 1956142 (2021).
7. Petit, A., *et al.* A major secretory defect of tumour-infiltrating T lymphocytes due to galectin impairing LFA-1-mediated synapse completion. *Nature communications* **7**, 12242 (2016).
8. Ramana, C., *et al.* Inflammatory impact of IFN- γ in CD8+ T cell-mediated lung injury is mediated by both Stat1-dependent and -independent pathways. *American journal of physiology. Lung cellular and molecular physiology* **308**, L650-657 (2015).
9. Li, J., *et al.* Transfer of in vitro expanded T lymphocytes after activation with dendritomas prolonged survival of mice challenged with EL4 tumor cells. *International journal of oncology* **31**, 193-197 (2007).
10. Cheng, L., *et al.* Proteomic and lipidomic analysis of exosomes derived from ovarian cancer cells and ovarian surface epithelial cells. *Journal of ovarian research* **13**, 9 (2020).
11. Ponath, V., *et al.* Secreted Ligands of the NK Cell Receptor NKp30: B7-H6 Is in Contrast to BAG6 Only Marginally Released via Extracellular Vesicles. *International journal of molecular sciences* **22**(2021).
12. Yang, H. & Rhee, W. Single Step In Situ Detection of Surface Protein and MicroRNA in Clustered Extracellular Vesicles Using Flow Cytometry. *Journal of clinical medicine* **10**(2021).
13. Chen, G., *et al.* Exosomal PD-L1 contributes to immunosuppression and is associated with anti-PD-1 response. *Nature* **560**, 382-386 (2018).

14. Liu, J., *et al.* Immunosuppressive effect of small extracellular vesicle PD-L1 is restricted by co-expression of CD80. *British journal of cancer* **129**, 925-934 (2023).
15. Wang, Q., *et al.* Blood exosomes regulate the tissue distribution of grapefruit-derived nanovector via CD36 and IGFR1 pathways. *Theranostics* **8**, 4912-4924 (2018).
16. Sun, P., *et al.* Circulating Exosomes Control CD4 T Cell Immunometabolic Functions via the Transfer of miR-142 as a Novel Mediator in Myocarditis. *Molecular therapy : the journal of the American Society of Gene Therapy* **28**, 2605-2620 (2020).
17. Liu, T., *et al.* Evaluating adipose-derived stem cell exosomes as miRNA drug delivery systems for the treatment of bladder cancer. *Cancer medicine* **11**, 3687-3699 (2022).
18. Maji, S., *et al.* Exosomal Annexin II Promotes Angiogenesis and Breast Cancer Metastasis. *Molecular cancer research : MCR* **15**, 93-105 (2017).
19. Luo, Z., *et al.* Exosomal OTULIN from M2 macrophages promotes the recovery of spinal cord injuries via stimulating Wnt/ β -catenin pathway-mediated vascular regeneration. *Acta biomaterialia* **136**, 519-532 (2021).
20. Bu, T., *et al.* Il-10Exosome-mediated delivery of inflammation-responsive mRNA for controlled atherosclerosis treatment. *Theranostics* **11**, 9988-10000 (2021).
21. Yu, L., *et al.* Exosomes derived from osteogenic tumor activate osteoclast differentiation and concurrently inhibit osteogenesis by transferring COL1A1-targeting miRNA-92a-1-5p. *Journal of extracellular vesicles* **10**, e12056 (2021).
22. You, H., *et al.* Mesenchymal stem cell-derived exosomes improve motor function and attenuate neuropathology in a mouse model of Machado-Joseph disease. *Stem cell research & therapy* **11**, 222 (2020).

REVIEWERS' COMMENTS

Reviewer #1 (Remarks to the Author):

This is the third review round for Dr. Chen's manuscript. The author has responded well to my concerns, addressing all in a satisfactory manner. The addition of methodological details relevant to single vesicle flow cytometry strengthens the scientific impact of the paper. Overall, this is a strong contribution. However, the revised manuscript still requires corrections in the use of English, including a correction in the manuscript title.

Below, I have listed the corrections that are needed in the manuscript.

The title should read: "PD-1/CD80+ small extracellular vesicles from immunocytes induce cold tumors featuring enhanced adaptive immunosuppression"

Abstract: " In addition to....., PD-1/CD80+ sEV downregulated and impaired....."

Line 84, replace "rendered" with "considered"

Line 85, "PD-L1 is enriched on tumor cell-secreted.....and suppresses....."

Line 103, "PD-1+ sEVs alleviate.....while the other proposes that they reduce....."

Line 110, ".... of PD-1/CD80+sEVs, derived mainly....."

Line111, "...D1/CD80 on sEV functions via...."

Line 156, "...the peripheral circulation...." (here and elsewhere)

Line 164, "...while proportions of single positive sEVs for PD-1, CD80 and PD-L1 were comparable...."

Line 168, "...PD-1/CD80+ sEVs impair the response...."

Line 178, ...replace "tumor threat" with "tumor development"

Line 214, "...significantly altered the effects...."

Line 219, replace "revealed" with "were obtained"

Line 225, "...correlated with levels of...."

Line 227, replace "have revealed" with "showed" (here and elsewhere)levels of PD-1 and CD80 on circulating PD-L1+ sEVs are also associated associated....." Please pay attention to the structure of this sentence.

Line 238, remove "obviously"

Line 240, "...levels induced by aT-sEVs were suppressed (remove "almost")

Line 249, "...after application of aT-sEVs...."

Line 257, "...potentially contribute" and "decrease"

Line 318, "Taken together, these results suggest....."

Line 319, remove "mainly "

Line 321, remove "entirely"

Line 357, replace by "macrophages"

Line383, replace "almost" with "nearly"

Line 418, replace with "to a different extent"

Line 454"...speculated that a new strategy...."

Line 455, replace "to predict" with "predicts response to immunotherapy"

Line 505, replace "multivariable" with "multivariate"

Line 528, "... plays a greater role...."

Line 529,"Immunity: even simultaneous stimulation of tumor cells....."

Line 533, Put a period after "immunity."In contrast, IFN-γ...."

Line 547, "...in the circulation..."

Line 548, delete "in circulation".

Line 554, "...requiring a more sophisticated blockade than that provided by Abs."

Line 555, "This is supported by...."

Line 575/576, "...serves as a warning signal for"

Line 581, replace "renovates" with "reprograms"

Reviewer #4 (Remarks to the Author):

The authors provided reasonable response to my questions.

Point-by-point response

Reviewer #1

Remarks to the Authors

This is the third review round for Dr. Chen's manuscript. The author has responded well to my concerns, addressing all in a satisfactory manner. The addition of methodological details relevant to single vesicle flow cytometry strengthens the scientific impact of the paper. Overall, this is a strong contribution. However, the revised manuscript still requires corrections in the use of English, including a correction in the manuscript title. Below, I have listed the corrections that are needed in the manuscript.

The title should read: "PD-1/CD80+ small extracellular vesicles from immunocytes induce cold tumors featuring enhanced adaptive immunosuppression"

Abstract: "In addition to....., PD-1/CD80+ sEV downregulated and impaired....."

Line 84, replace "rendered" with "considered"

Line 85, "PD-L1 is enriched on tumor cell-secreted.....and suppresses....."

Line 103, "PD-1+ sEVs alleviate.....while the other proposes that they reduce....."

Line 110, ".... of PD-1/CD80+sEVs, derived mainly....."

Line 111, "...D1/CD80 on sEV functions via...."

Line 156, "...the peripheral circulation...." (here and elsewhere)

Line 164, "...while proportions of single positive sEVs for PD-1, CD80 and PD L1 were comparable....."

Line 168, "...PD-1/CD80+ sEVs impair the response...."

Line 178, ...replace "tumor threat" with "tumor development"

Line 214, "...significantly altered the effects...."

Line 219, replace "revealed" with "were obtained"

Line 225, "...correlated with levels of...."

Line 227, replace "have revealed" with "showed" (here and elsewhere) levels of PD-1 and CD80 on circulating PD-L1+ sEVs are also associated associated....." Please pay attention to the structure of this sentence.

Line 238, remove "obviously"

Line 240, "...levels induced by aT-sEVs were suppressed (remove "almost")

Line 249, "...after application of aT-sEVs...."

Line 257, "...potentially contribute" and "decrease"

Line 318, "Taken together, these results suggest....."

Line 319, remove "mainly "

Line 321, remove "entirely"

Line 357, replace by "macrophages"

Line 383, replace "almost" with "nearly"

Line 418, replace with "to a different extent"

Line 454, "...speculated that a new strategy...."

Line 455, replace "to predict" with "predicts response to immunotherapy"

Line 505, replace "multivariable" with "multivariant"

Line 528, "... plays a greater role...."

Line 529, "Immunity: even simultaneous stimulation of tumor cells....."

Line 533, Put a period after "immunity."In contrast, IFN- γ"

Line 547, "...in the circulation..."

Line 548, delete "in circulation".

Line 554, "...requiring a more sophisticated blockade than that provided by Abs."

Line 555, "This is supported by...."

Line 575/576, "...serves as a warning signal for"

Line 581, replace "renovates" with "reprograms"

Response: We thank the Reviewer very much for the detailed review of our manuscript and for providing valuable feedback to improve the quality of the paper. We greatly appreciate these thoroughness and constructive suggestions. We are also pleased to hear that the Reviewer found the revisions satisfactory and that the addition of methodological details has strengthened the scientific impact of the paper.

We have carefully reviewed all the suggested corrections and have made revisions to ensure the accuracy and clarity of the manuscript.

Reviewer #4

Remarks to the Authors

The authors provided reasonable response to my questions.

Response: We appreciate the positive response from the Reviewer.